# MisoDICE: Multi-Agent Imitation from Unlabeled Mixed-Quality Demonstrations

**The Viet Bui**
Singapore Management University, Singapore
`theviet.bui.2023@phdcs.smu.edu.sg`

**Tien Mai**
Singapore Management University, Singapore
`atmai@smu.edu.sg`

**Thanh Hong Nguyen**
University of Oregon Eugene, Oregon, United States
`thanhhng@cs.uoregon.edu`

## Abstract

We study offline imitation learning (IL) in cooperative multi-agent settings, where demonstrations have *unlabeled mixed quality* — containing both expert and suboptimal trajectories. Our proposed solution is structured in two stages: trajectory labeling and multi-agent imitation learning, designed jointly to enable effective learning from heterogeneous, unlabeled data. In the first stage, we combine advances in large language models and preference-based reinforcement learning to construct a progressive labeling pipeline that distinguishes expert-quality trajectories. In the second stage, we introduce MisoDICE, a novel multi-agent IL algorithm that leverages these labels to learn robust policies while addressing the computational complexity of large joint state-action spaces. By extending the popular single-agent DICE framework to multi-agent settings with a new value decomposition and mixing architecture, our method yields a convex policy optimization objective and ensures consistency between global and local policies. We evaluate MisoDICE on multiple standard multi-agent RL benchmarks and demonstrate superior performance, especially when expert data is scarce.

## 1   Introduction

Imitation Learning (IL) is a key subfield of Reinforcement Learning (RL) that focuses on learning effective policies solely from expert demonstration data, without requiring direct access to reward functions [60]. A broad range of IL algorithms have been developed — primarily for single-agent settings — ranging from simple supervised learning methods like Behavior Cloning (BC) [41, 45] to more advanced occupancy-matching techniques [16, 18, 23]. While these methods typically assume access to high-quality expert demonstrations, recent research has extended single-agent IL to more challenging scenarios involving limited or suboptimal data [4, 8, 28, 65].

Motivated by the success of single-agent IL, recent studies have sought to extend IL to multi-agent settings characterized by complex inter-agent dynamics. For example, leading multi-agent IL algorithms have adapted well-known single-agent approaches [16, 18, 23] by incorporating game-theoretic concepts such as Nash equilibrium for competitive scenarios [51, 59], or by carefully decomposing joint policies into local policies in cooperative tasks [36]. A key limitation of these methods is their reliance on large volumes of expert data to learn effective policies. In practice, acquiring such comprehensive datasets in complex multi-agent environments is highly impractical, due to the combinatorial explosion of joint state-action spaces as the number of agents increases.

39th Conference on Neural Information Processing Systems (NeurIPS 2025).

Our work tackles a new highly challenging problem of offline multi-agent IL using *only unlabeled demonstrations of mixed quality* (i.e., containing both expert and sub-optimal trajectories). To the best of our knowledge, this is the first effort to address such a practical yet underexplored setting in multi-agent IL. Our proposed solution is structured into two key stages: data labeling and multi-agent IL, that work together to enable effective learning from heterogeneous unlabeled data.

The first stage, named *multi-step data labeling*, combines recent advances in LLMs and preference-based RL to build a progressive trajectory labeling pipeline. We begin by leveraging LLM-generated feedback to produce initial preference annotations between pairs of trajectories from the unlabeled dataset. While LLMs can offer valuable insights, their judgments are error-prone in the complex, high-dimensional dynamics of multi-agent environments. To refine this coarse feedback, we introduce a second step using O-MAPL, a leading preference-based MARL algorithm [6]. O-MAPL trains a soft Q-function based on the initial LLM preferences, capturing a more grounded notion of demonstration quality. We then re-purpose this Q-function to extract reward signals and re-rank trajectories, enabling more accurate separation of expert and suboptimal data. This ranking lays the foundation for more effective and data-efficient multi-agent IL in our subsequent learning stage.

The second stage, the algorithmic core of our proposed framework, focuses on designing a novel multi-agent IL algorithm, MisoDICE, capable of effectively leveraging the labeled demonstrations (expert vs. non-expert) to learn robust policies. MisoDICE follows an occupancy-matching principle, minimizing the divergence between the learned and behavior policies within a mixed-quality dataset. To manage the exponential complexity in multi-agent environments, we build on the established Centralized Training with Decentralized Execution (CTDE) paradigm. However, MisoDICE significantly differs from prior CTDE-based learning methods by introducing customized learning objectives and constraints that are suited to the challenges of mixed-quality demonstrations. Furthermore, MisoDICE features a unique value decomposition strategy and is supported by a theoretical analysis establishing local-global optimality consistency. Specifically, we show that under our value factorization strategy, the global learning objective is convex, which ensures both stability and efficiency during training. Moreover, we prove that once the value function is learned, each agent's local policy can be optimized independently and in parallel, while still guaranteeing convergence to the globally optimal joint policy. We also derive a closed-form mathematical relationship between the local policies and the value function, offering deeper insight into the structure and characteristics of the learned solution.

Finally, we conduct extensive experiments with comprehensive ablation studies on challenging multi-agent environments, i.e., SMACv1 [46], and SMACv2 [12]. The results show that our MisoDICE outperforms all baselines, highlighting the benefit of its integrated approach, combining occupancy matching, value decomposition, and effective use of the multi-step labeling process.

## 2 Related Work

**Imitation Learning (IL).** The well-known behavioral cloning (BC) frames IL as a supervised learning problem by maximizing the likelihood of expert state-action pairs under the learned policy [41, 45]. However, BC disregards environment dynamics, leading to compounding errors due to distributional shift. Recent methods recover the expert's reward function (explicitly or implicitly) so that the learned policy can account for such dynamics [14, 16, 23, 29, 43]. Notable examples include distribution-matching-based GAIL and AIRL algorithms [16, 23], alternatively optimizing policy and reward functions via an adversarial learning process, which requires access to on-policy data. In contrast, ValueDice [29] introduces an off-policy divergence objective to align state-action occupancies, enabling fully offline training. IQ-Learn [18] bypasses reward recovery by directly learning the Q-function, leading to more stable policy learning than adversarial methods. Beyond these, other research addresses practical challenges in single-agent environments such as limited [65] or imperfect demonstrations [4, 8, 28], and constrained IL [47]. Additional directions include model-based IL [2, 61, 63] and the use of diffusion models to improve policy learning from demonstrations [30, 54]. While many works focus on single-agent settings, only few studies have explored IL in multi-agent environments, whether cooperative [3, 5, 31, 36, 52] or competitive [33, 44, 51, 59], primarily due to the complex nature of multi-agent interactions. Some adapt single-agent methods like GAIL and AIRL to multi-agent settings using game-theoretic concepts [51, 59], but they inherit instability issues from adversarial training. A recent algorithm, MIFQ [36], extends IQ-Learn [18] to the multi-agent setting under the CTDE paradigm, which achieves outstanding performance in multi-agent IL

scenarios. Unlike existing multi-agent IL methods that rely on high-quality expert data, we address a more challenging setting with only unlabeled demonstrations of various quality are available.

**Preference-based Reinforcement Learning (PbRL).** PbRL is an emerging subfield of RL that can train policies only using preference data, typically in the form of pairwise trajectory comparisons. Most existing PbRL methods follow a two-phase approach: first, a reward function is learned via supervised learning techniques (e.g., using the Bradley-Terry model), and then RL is applied to optimize the policy accordingly [9, 10, 17, 22, 32, 24, 27, 37, 64]. However, this separation introduces an additional learning component atop already complex RL algorithms, resulting in high variance and instability during training. To mitigate this issue, recent works proposed end-to-end methods that learn policies directly from preferences, using contrastive learning [1, 20], information matching [26], or Q-function optimization [21], resulting in greater stability and performance. Following advances in single-agent PbRL, recent works have extended it to multi-agent settings [6, 25, 62], with O-MAPL [6] achieving state-of-the-art results on challenging multi-agent benchmarks. Our work uses LLM-generated feedback to produce initial preference data, which is then used to train a soft Q-function via O-MAPL. We treat O-MAPL as a label-refinement step by extracting reward signals from its Q-function to partition the unlabeled data into expert and non-expert trajectories.

**Centralized Training with Decentralized Execution (CTDE).** CTDE is a well-known learning paradigm in MARL in which global information are leveraged for an efficient training of joint value functions [15, 34, 42, 53] while allowing agents to operate independently based on their local observations. There are various MARL algorithms exploring different value decomposition strategies under CTDE. For example, QMIX [42] offers an advanced value decomposition method based on mixing and hyper-network concepts [19]. Others such as QTRAN [50] and QPLEX [55], introduce new factorization methods which are more expressive than QMIX. Besides these value-based methods, there are policy gradient based MARL algorithms [35, 58] which are extensions of the PPO algorithm [48]. Building on the success of CTDE in MARL, this paradigm has been extended to multi-agent IL [16, 23, 36], preference-based RL [6, 25, 62], and offline MARL [38, 40, 49, 56, 57]. Our multi-agent IL algorithm adopts the DICE approach to learn effective policies from mixed-quality data within the CTDE framework. Despite operating under the same CTDE framework, existing methods — including ours — differ notably in learning objectives, constraints, value decomposition strategies, and theoretical guarantees, each tailored to their specific problem settings.

# 3 Preliminaries

Our work focuses on cooperative MARL, modeled as a multi-agent Partially Observable Markov Decision Process (POMDP): $\mathcal{M} = \langle \mathcal{S}, \mathcal{A}, P, r, \mathcal{Z}, \mathcal{O}, n, \mathcal{N}, \gamma \rangle$ where $n$ is the number of agents and $\mathcal{N} = \{1, \ldots, n\}$ is the agent set. The environment has the global state space $\mathcal{S}$, and the joint action space is $\mathcal{A} = \prod_{i \in \mathcal{N}} \mathcal{A}_i$, where $\mathcal{A}_i$ is the set of actions for agent $i$. At each time step, each agent $i$ selects an action $a_i \in \mathcal{A}_i$, forming a joint action $\mathbf{a} = (a_1, a_2, \ldots, a_n) \in \mathcal{A}$. The transition dynamics are governed by $\mathcal{T}(\mathbf{s}' \mid \mathbf{s}, \mathbf{a}) : \mathcal{S} \times \mathcal{A} \times \mathcal{S} \to [0, 1]$, which is the probability of transitioning to global state $\mathbf{s}'$ from state $\mathbf{s}$ under action $\mathbf{a}$. Finally, $\gamma \in [0, 1)$ is the discounting factor. We also define the *occupancy measure* of a joint policy $\boldsymbol{\pi}_{\text{tot}}$ as $\rho^{\boldsymbol{\pi}_{\text{tot}}}(\mathbf{s}, \mathbf{a}) = (1 - \gamma) \sum_{t=0}^{\infty} \gamma^t P(\mathbf{s}_t = \mathbf{s}, \mathbf{a}_t = \mathbf{a} | \boldsymbol{\pi}_{\text{tot}})$, which is the discounted visitation distribution over state-action pairs when following $\boldsymbol{\pi}_{\text{tot}}$.

In a partially observable setting, each agent receives a local observation $o_i \in \mathcal{O}_i$ via an observation function $\mathcal{Z}_i(\mathbf{s}) : \mathcal{S} \to \mathcal{O}_i$, and the joint observation is $\mathbf{o} = (o_1, o_2, \ldots, o_n)$. In practice, the global state $\mathbf{s}$ is not directly accessible and is instead represented by the joint observation $\mathbf{o} = \mathcal{Z}(\mathbf{s})$. For notational simplicity, we use $\mathbf{s}$ in our formulation, but it refers to $\mathbf{o}$ in implementation.

In cooperative MARL, all agents share a global reward function $r(\mathbf{s}, \mathbf{a}) : \mathcal{S} \times \mathcal{A} \to \mathbb{R}$. The objective is to learn a joint policy $\boldsymbol{\pi}_{\text{tot}} = \{\pi_1, \ldots, \pi_n\}$ that maximizes the expected discounted return:

$$\mathbb{E}_{\{\mathbf{s}_t, \mathbf{a}_t\}_t \sim \boldsymbol{\pi}_{\text{tot}}} \left[ \sum_{t=0}^{\infty} \gamma^t r(\mathbf{s}_t, \mathbf{a}_t) \right].$$

In offline MARL, learning is performed solely based on a fixed offline dataset $\mathcal{D}$, collected from a behavior policy $\mu_{\text{tot}} = \{\mu_1, \ldots, \mu_n\}$. In offline multi-agent IL, the reward function is not available. Instead, a set of expert demonstrations is provided, and the goal is to recover both the global and local expert policies accordingly. A common objective is to minimize the divergence between the stationary distribution induced by the learned policy and that of the expert demonstrations:

$$\boldsymbol{\pi}_{tot}^* = \operatorname{argmin}_{\boldsymbol{\pi}_{\text{tot}}} D_{\text{KL}} \left( \rho_{tot}^{\boldsymbol{\pi}} \,\|\, \rho_{tot}^E \right), \tag{1}$$

where $\rho_{\text{tot}}^{\boldsymbol{\pi}}$ denotes the global state-action visitation distribution under the learned policy $\boldsymbol{\pi}_{\text{tot}}$, and $\rho_{\text{tot}}^{E}$ denotes the corresponding distribution derived from expert demonstrations.

The DICE framework has been widely adopted to address training objectives involving KL divergence between stationary distributions (e.g., Eq. (1)). The core idea is to formulate IL as a constrained optimization program over occupancy measures. In a later section, we extend this approach by integrating DICE with value decomposition for efficient offline multi-agent IL.

# 4 Unlabeled Demonstrations: Multi-Step Labeling Procedure via LLMs

Addressing multi-agent IL from unlabeled demonstrations of mixed quality requires a robust procedure to identify high-quality, expert-like behaviors within the dataset. We propose a multi-step data labeling pipeline designed to progressively refine trajectory quality assessments and partition the unlabeled dataset ($\mathcal{D}_{unlabeled}$) into expert ($\mathcal{D}^{E}$) and non-expert segments ($\mathcal{D}^{Mix}$). This process serves as the first crucial stage of our MisoDICE framework for subsequent efficient IL.

The initial step in this procedure involves generating preference data. A set of trajectory pairs ($N_p$) is randomly sampled from $\mathcal{D}_{unlabeled}$. An LLM (e.g., GPT-4o) is then prompted to provide preference labels for these pairs, based on high-level semantic features and game context, such as unit health, positions, and actions in SMAC environments. This yields an initial preference dataset $\mathcal{P}$. While LLMs offer valuable insights, their judgments in complex multi-agent settings can be error-prone.

The next step is preference-based reward recovery. The preference dataset $\mathcal{P}$ is used to train an offline preference-based MARL algorithm, i.e., the leading O-MAPL [6]. O-MAPL learns a soft Q-function (and corresponding value function V) directly from these preferences, capturing a more grounded notion of demonstration quality compared to raw LLM feedback. After training O-MAPL, an underlying reward function $R(s, a)$ is recovered for all state-action pairs in $\mathcal{D}_{unlabeled}$. This recovery utilizes the relationship $R \approx Q - \gamma V$, adapted from MaxEnt RL principles.

Using the recovered rewards, we compute the total return $G(\sigma) = \sum_{(s,a) \in \sigma} R(s, a)$ for each trajectory $\sigma$ in $\mathcal{D}_{unlabeled}$ and rank them accordingly. The top-$K_{expert}$ trajectories are selected to form the expert dataset $\mathcal{D}^{E}$ and the remaining trajectories form the suboptimal dataset $\mathcal{D}^{Mix}$. These datasets are used in the subsequent multi-agent IL phase (Phase 2 of MisoDICE).

This multi-step labeling process enables MisoDICE to leverage unlabeled, mixed-quality demonstrations by inferring demonstration quality through preference learning and using it to guide imitation learning. The use of LLMs for initial preference generation, followed by refinement with O-MAPL, offers a powerful mechanism for identifying expert behavior without prior annotations.

# 5 Multi-Agent Imitation Learning from Mixed-Quality Demonstrations

## 5.1 Distribution Matching Based Formulation

Our goal is to leverage two datasets (as a result of our Phase 1): (1) a limited *expert dataset*, $\mathcal{D}^{E}$; and (2) a large *suboptimal dataset*, $\mathcal{D}^{Mix}$ to learn global and local policies that closely imitate the expert behavior. We first formulate the DICE-based learning objective as a function of the global joint policy $\boldsymbol{\pi}_{\text{tot}}$ for multi-agent IL in a similar fashion as prior work [28] for single-agent IL:

$$\min_{\boldsymbol{\pi}_{\text{tot}}} \ D_{\text{KL}} \left( \rho_{\text{tot}}^{\boldsymbol{\pi}} \,\|\, \rho_{\text{tot}}^{E} \right) + \alpha \, D_{\text{KL}} \left( \rho_{\text{tot}}^{\boldsymbol{\pi}} \,\|\, \rho_{\text{tot}}^{U} \right), \tag{2}$$

where $\rho_{\text{tot}}^{E}$ and $\rho_{\text{tot}}^{U}$ denote the global state-action stationary distributions induced by the expert dataset $\mathcal{D}^{E}$ and the union dataset $\mathcal{D}^{U} = \mathcal{D}^{E} \cup \mathcal{D}^{Mix}$, respectively. The hyperparameter $\alpha \geq 0$ controls the influence of the suboptimal data in the learning objective.

We can reformulate (2) as a constrained optimization problem over the occupancy measure $\rho_{\text{tot}}^{\boldsymbol{\pi}}$:

$$\min_{\{\rho_{\text{tot}}^{\boldsymbol{\pi}}(\mathbf{s},\mathbf{a}) \geq 0\}} \quad \sum_{\mathbf{s},\mathbf{a}} \rho_{\text{tot}}^{\boldsymbol{\pi}}(\mathbf{s}, \mathbf{a}) \log \frac{\rho_{\text{tot}}^{\boldsymbol{\pi}}(\mathbf{s}, \mathbf{a})}{\rho_{\text{tot}}^{E}(\mathbf{s}, \mathbf{a})} + \alpha \, \rho_{\text{tot}}^{\boldsymbol{\pi}}(\mathbf{s}, \mathbf{a}) \log \frac{\rho_{\text{tot}}^{\boldsymbol{\pi}}(\mathbf{s}, \mathbf{a})}{\rho_{\text{tot}}^{U}(\mathbf{s}, \mathbf{a})} \tag{3}$$

$$\text{s.t.} \quad \sum_{\mathbf{a}} \rho_{\text{tot}}^{\boldsymbol{\pi}}(\mathbf{s}, \mathbf{a}) = (1 - \gamma) P_0(\mathbf{s}) + \gamma \sum_{\mathbf{s}', \mathbf{a}'} \mathcal{T}(\mathbf{s} \mid \mathbf{s}', \mathbf{a}') \rho_{\text{tot}}^{\boldsymbol{\pi}}(\mathbf{s}', \mathbf{a}'), \quad \forall \mathbf{s} \in \mathcal{S} \tag{4}$$

Constraint (4) enforces the Bellman flow condition, ensuring that $\rho_{\text{tot}}^{\boldsymbol{\pi}}$ corresponds to a valid state-action visitation distribution. The objective in (3) is convex in $\rho_{\text{tot}}^{\boldsymbol{\pi}}(\mathbf{s}, \mathbf{a})$. Thus we can incorporate the

flow constraints into (3) using Lagrangian duality, resulting in the following saddle-point formulation:

$$\max_{\nu^{\text{tot}}} \min_{\rho^{\boldsymbol{\pi}}_{\text{tot}} \geq 0} \left\{ \mathcal{L}(\nu^{\text{tot}}, \rho^{\boldsymbol{\pi}}_{\text{tot}}) \right\},$$

where $\nu^{\text{tot}}$ are Lagrange multipliers corresponding to the Bellman flow constraints, and the objective

$$\mathcal{L}(\nu^{\text{tot}}, \rho^{\boldsymbol{\pi}}_{\text{tot}}) = \sum_{\mathbf{s},\mathbf{a}} \rho^{\boldsymbol{\pi}}_{\text{tot}}(\mathbf{s},\mathbf{a}) \log \frac{\rho^{\boldsymbol{\pi}}_{\text{tot}}(\mathbf{s},\mathbf{a})}{\rho^{E}_{\text{tot}}(\mathbf{s},\mathbf{a})} + \alpha\, \rho^{\boldsymbol{\pi}}_{\text{tot}}(\mathbf{s},\mathbf{a}) \log \frac{\rho^{\boldsymbol{\pi}}_{\text{tot}}(\mathbf{s},\mathbf{a})}{\rho^{U}_{\text{tot}}(\mathbf{s},\mathbf{a})}$$
$$+ \sum_{\mathbf{s}} \nu^{\text{tot}}(\mathbf{s}) \Big( (1-\gamma) P_0(\mathbf{s}) + \gamma \sum_{\mathbf{s}',\mathbf{a}'} \mathcal{T}(\mathbf{s}|\mathbf{s}',\mathbf{a}') \rho^{\boldsymbol{\pi}}_{\text{tot}}(\mathbf{s}',\mathbf{a}') - \sum_{\mathbf{a}} \rho^{\boldsymbol{\pi}}_{\text{tot}}(\mathbf{s},\mathbf{a}) \Big) \quad (5)$$

which can be simplified to a function of $(\nu^{\text{tot}}, w^{\text{tot}})$ as follows:

$$\mathcal{L}(\nu^{\text{tot}}, w^{\text{tot}}) = (1-\gamma) \mathbb{E}_{\mathbf{s} \sim P_0}[\nu^{\text{tot}}(\mathbf{s})] + \mathbb{E}_{(\mathbf{s},\mathbf{a}) \sim \rho^{U}_{\text{tot}}}[w^{\text{tot}}(\mathbf{s},\mathbf{a})(A^{\text{tot}}_{\nu}(\mathbf{s},\mathbf{a}) - (1+\alpha) \log(w^{\text{tot}}(\mathbf{s},\mathbf{a}))]$$

where $A^{\text{tot}}_{\nu}(\mathbf{s},\mathbf{a}) = \log \frac{\rho^{E}_{\text{tot}}(\mathbf{s},\mathbf{a})}{\rho^{U}_{\text{tot}}(\mathbf{s},\mathbf{a})} + \gamma \sum_{\mathbf{s}'} \mathcal{T}(\mathbf{s}'|\mathbf{s},\mathbf{a}) \nu^{\text{tot}}(\mathbf{s}') - \nu^{\text{tot}}(\mathbf{s})$ and $w^{\text{tot}}(\mathbf{s},\mathbf{a}) = \frac{\rho^{\boldsymbol{\pi}}_{\text{tot}}(\mathbf{s},\mathbf{a})}{\rho^{U}_{\text{tot}}(\mathbf{s},\mathbf{a})}$

Details of this derivation, adapted to the multi-agent setting from the single-agent IL in [28], can be found in our appendix. Here, $\nu^{\text{tot}}(\mathbf{s})$ can be interpreted as the value function at the global state $\mathbf{s}$. The reformulation above enables us to express the training objective as an optimization problem over $\nu^{\text{tot}}$ and $w^{\text{tot}}$ in which the objective $\mathcal{L}(\nu^{\text{tot}}, w^{\text{tot}})$ is linear in $\nu^{\text{tot}}$ and convex in $w^{\text{tot}}$. Moreover, the inner problem $\max_{w^{\text{tot}}} \mathcal{L}(\nu^{\text{tot}}, w^{\text{tot}})$ admits a closed-form solution, allowing the original minimax objective to be reduced to the following non-adversarial minimization problem over $\nu^{\text{tot}}$:

$$\max_{w^{\text{tot}}} \mathcal{L}(\nu^{\text{tot}}, w^{\text{tot}}) = \mathcal{L}(\nu^{\text{tot}}, w^{*}_{\nu}) \text{ where } w^{*}_{\nu} = \exp \Big( \frac{A^{\text{tot}}_{\nu}(\mathbf{s},\mathbf{a})}{1+\alpha} - 1 \Big)$$

## 5.2 Value Factorization

The objective $\mathcal{L}(\nu^{\text{tot}}, w^{*}_{\nu})$ is generally intractable in the multi-agent setting due to the high dimensionality of the joint state and action space. To address this, we adopt a value factorization technique in which the global $\nu^{\text{tot}}$ is decomposed into local value functions using a mixing network. Specifically, let $\boldsymbol{\nu}(\mathbf{s}) = \{\nu_i(s_i)\}_{i \in \mathcal{N}}$ be the set of local value functions. We introduce a mixing network $\mathcal{M}_{\phi}$, parameterized by $\phi$, which aggregates the local values to form the global value function:

$$\nu^{\text{tot}}(\mathbf{s}) = \mathcal{M}_{\phi}[\boldsymbol{\nu}(\mathbf{s})].$$

To implement $\mathcal{M}_{\phi}$, we adopt a linear structure (e.g., a single-layer network) due to: (i) it preserves convexity in the learning objective within the value function space, leading to stable and consistent training; and (ii) nonlinear mixing networks (e.g., a two-layer network with ReLU activations), particularly in offline settings with limited data, are prone to overfitting and have unstable performance [7].

Under the mixing architecture described above, the training objective becomes:

$$\mathcal{L}(\phi, \boldsymbol{\nu}) = (1-\gamma)\, \mathbb{E}_{\mathbf{s} \sim P_0} \left[ \mathcal{M}_{\phi}[\boldsymbol{\nu}(\mathbf{s})] \right] + \mathbb{E}_{(\mathbf{s},\mathbf{a}) \sim \rho^{U}_{\text{tot}}} \left[ w^{*}_{\boldsymbol{\nu}}(\mathbf{s},\mathbf{a}) \left( A^{\text{tot}}_{\boldsymbol{\nu}}(\mathbf{s},\mathbf{a}) - (1+\alpha) \log w^{*}_{\boldsymbol{\nu}}(\mathbf{s},\mathbf{a}) \right) \right], \quad (6)$$

where the total advantage function $A^{\text{tot}}_{\boldsymbol{\nu}}$ and the optimal weighting function $w^{*}_{\boldsymbol{\nu}}$ are defined as:

$$A^{\text{tot}}_{\boldsymbol{\nu}}(\mathbf{s},\mathbf{a}) = \log \frac{\rho^{E}_{\text{tot}}(\mathbf{s},\mathbf{a})}{\rho^{U}_{\text{tot}}(\mathbf{s},\mathbf{a})} + \gamma \sum_{\mathbf{s}'} \mathcal{T}(\mathbf{s}' \mid \mathbf{s},\mathbf{a}) \mathcal{M}_{\phi}[\boldsymbol{\nu}(\mathbf{s}')] - \mathcal{M}_{\phi}[\boldsymbol{\nu}(\mathbf{s})], \quad (7)$$

$$w^{*}_{\boldsymbol{\nu}}(\mathbf{s},\mathbf{a}) = \exp \Big( \frac{A^{\text{tot}}_{\boldsymbol{\nu}}(\mathbf{s},\mathbf{a})}{1+\alpha} - 1 \Big). \quad (8)$$

**Proposition 5.1.** *If the mixing network $\mathcal{M}_{\phi}[\boldsymbol{\nu}(\mathbf{s})]$ is a linear function of both $\boldsymbol{\nu}(\mathbf{s})$ and $\phi$, then the training objective $\mathcal{L}(\phi, \boldsymbol{\nu})$ is convex in both $\phi$ and $\boldsymbol{\nu}$.*

As a result, if the mixing network is constructed as a linear combination of local value functions, i.e., $\mathcal{M}_{\phi}[\boldsymbol{\nu}(\mathbf{s})] = \sum_{i \in \mathcal{N}} \phi_i \nu_i(s_i) + \phi_0$, where $\phi = \{\phi_0, \phi_1, \ldots, \phi_n\}$ are learnable parameters, then the global value function is linear in both the parameters $\phi_i$ and the local value outputs $\nu_i(s_i)$. This structural property is critical for stable and efficient optimization, as it guarantees that the learning objective preserves convexity w.r.t both the policy parameters and the mixing network parameters.

Conversely, this objective convexity does not holds if we employ multi-layer neural network structures for the mixing network $\mathcal{M}_{\phi}[\boldsymbol{\nu}]$, leading to unstable training dynamics and convergence to poor local minima, especially in offline learning settings where data is limited and overfitting is a concern.

**Proposition 5.2.** *If the mixing network $\mathcal{M}_{\phi}[\boldsymbol{\nu}]$ is a multi-layer feedforward network with $\boldsymbol{\nu}$ as input (even with convex activation functions), the objective $\mathcal{L}(\phi, \boldsymbol{\nu})$ becomes non-convex in $\boldsymbol{\nu}$.*

## 5.3 Occupancy Ratio Estimation

The training objective described in Section 5.2 involves the computation of the total advantage function $A_{\boldsymbol{\nu}}^{\text{tot}}(\mathbf{s}, \mathbf{a})$, which in turn requires the occupancy ratio term $\log(\rho_{\text{tot}}^E(\mathbf{s}, \mathbf{a})/\rho_{\text{tot}}^U(\mathbf{s}, \mathbf{a}))$. However, this ratio is not directly observable from data. We now describe a practical method to estimate this ratio under the CTDE framework. Following extensions from the single-agent setting, the occupancy ratio can be estimated by solving the following discriminator-based classification problem:

$$\max_{c \in (0,1)^{|\mathcal{S}| \times |\mathcal{A}|}} \ J(c) = \mathbb{E}_{\rho_{\text{tot}}^E}[\log c(\mathbf{s}, \mathbf{a})] + \mathbb{E}_{\rho_{\text{tot}}^U}[\log(1 - c(\mathbf{s}, \mathbf{a}))]. \tag{9}$$

This objective is concave in $c$ and admits a unique optimal solution: $c^*(\mathbf{s}, \mathbf{a}) = \frac{\rho_{\text{tot}}^E(\mathbf{s},\mathbf{a})}{\rho_{\text{tot}}^E(\mathbf{s},\mathbf{a}) + \rho_{\text{tot}}^U(\mathbf{s},\mathbf{a})}$, from which we can recover the occupancy ratio via: $\frac{\rho_{\text{tot}}^E(\mathbf{s},\mathbf{a})}{\rho_{\text{tot}}^U(\mathbf{s},\mathbf{a})} = \frac{c^*(\mathbf{s},\mathbf{a})}{1 - c^*(\mathbf{s},\mathbf{a})}$.

In the multi-agent context, directly optimizing $c(\mathbf{s}, \mathbf{a})$ is impractical due to the exponential joint state-action space. We thus leverage the CTDE framework to approximate $c(\mathbf{s}, \mathbf{a})$ in a decentralized manner. We define local discriminators $\mathbf{c}(\mathbf{s}, \mathbf{a}) = \{c_i(s_i, a_i)\}_{i \in \mathcal{N}}$, where each $c_i$ corresponding to agent $i$. A mixing network $\mathcal{M}_\eta$, parameterized by $\eta$, is then used to aggregate these local outputs: $c(\mathbf{s}, \mathbf{a}) = \mathcal{M}_\eta[\mathbf{c}(\mathbf{s}, \mathbf{a})]$. The resulting training objective for estimating the occupancy ratio becomes:

$$J(\mathbf{c}, \eta) = \mathbb{E}_{\rho_{\text{tot}}^E}[\log \mathcal{M}_\eta[\mathbf{c}(\mathbf{s}, \mathbf{a})]] + \mathbb{E}_{\rho_{\text{tot}}^U}[\log(1 - \mathcal{M}_\eta[\mathbf{c}(\mathbf{s}, \mathbf{a})])].$$

It can be shown that this objective is concave in $\mathbf{c}$ if the mixing network $\mathcal{M}_\eta$ is linear in its inputs, which contributes to a stable training process. We state this formally below:

**Proposition 5.3.** *If the mixing network $\mathcal{M}_\eta[\boldsymbol{c}(\boldsymbol{s}, \boldsymbol{a})]$ is linear in $\boldsymbol{c}(\boldsymbol{s}, \boldsymbol{a})$, then the training objective $J(\boldsymbol{c}, \eta)$ for occupancy ratio estimation is concave in both $\boldsymbol{c}$ and $\eta$.*

## 5.4 Policy Extraction

We now discuss how to extract the global and local policies from the output of the training objective in Section 5.2. Once the value function $\boldsymbol{\nu}^*$ has been optimized, we can compute the corresponding optimal occupancy ratio $w_{\boldsymbol{\nu}^*}^{\text{tot}}$, which reflects the ratio between the visitation distributions of the optimal policy $\boldsymbol{\pi}^*$ and the mixture distribution $\rho_{\text{tot}}^U$:

$$w_{\boldsymbol{\nu}^*}^{\text{tot}}(\mathbf{s}, \mathbf{a}) = \exp\left(\frac{A_{\boldsymbol{\nu}^*}^{\text{tot}}(\mathbf{s}, \mathbf{a})}{1 + \alpha} - 1\right) = \frac{\rho_{\text{tot}}^{\boldsymbol{\pi}^*}(\mathbf{s}, \mathbf{a})}{\rho_{\text{tot}}^U(\mathbf{s}, \mathbf{a})}.$$

The global optimal policy can be then recovered by solving a weighted BC objective:

$$\max_{\boldsymbol{\pi}_{\text{tot}} \in \Pi} \sum_{(\mathbf{s}, \mathbf{a}) \sim \rho_{\text{tot}}^U} w_{\boldsymbol{\nu}^*}^{\text{tot}}(\mathbf{s}, \mathbf{a}) \log \boldsymbol{\pi}_{\text{tot}}(\mathbf{s}, \mathbf{a}), \tag{10}$$

where $\Pi$ denotes the feasible set of joint policies. In the multi-agent setting, especially under the CTDE framework, it is more practical and desirable to recover decentralized local policies $\{\pi_i\}_{i \in \mathcal{N}}$. Therefore, we adopt a local weighted behavior cloning approach to recover each agent's policy:

$$\max_{\pi_i} \sum_{(s_i, a_i) \sim \rho_{\text{tot}}^U} w_{\boldsymbol{\nu}^*}^{\text{tot}}(\mathbf{s}, \mathbf{a}) \log \pi_i(a_i \mid s_i), \tag{11}$$

where all agents share the same global weight $w_{\boldsymbol{\nu}^*}^{\text{tot}}(\mathbf{s}, \mathbf{a})$. This enables each local policy to be optimized using globally-informed signals, ensuring alignment with the credit assignment in the multi-agent system. Furthermore, as shown in Proposition 5.4, the optimization of local policies via this decentralized weighted BC is consistent with the global weighted BC formulation in Equation (10).

**Proposition 5.4** (Global–Local Consistency)**.** *Assume the global policy space $\Pi$ is factorizable, i.e., $\Pi = \left\{\boldsymbol{\pi}_{tot} \mid \exists \{\pi_i\}_{i \in \mathcal{N}} \text{ such that } \boldsymbol{\pi}_{tot}(\boldsymbol{s}, \boldsymbol{a}) = \prod_{i \in \mathcal{N}} \pi_i(a_i \mid s_i)\right\}$. Let $\{\pi_i^*\}_{i \in \mathcal{N}}$ and $\boldsymbol{\pi}_{tot}^*$ be the solutions to the local and global weighted BCs, respectively. Then for any $(\boldsymbol{s}, \boldsymbol{a})$, it holds that:*

$$\boldsymbol{\pi}_{tot}^*(\boldsymbol{s}, \boldsymbol{a}) = \prod_{i \in \mathcal{N}} \pi_i^*(a_i \mid s_i).$$

The local weighted BC objective also reveals an interesting connection between the optimal local policies and the local value functions obtained from optimizing the objective in Section 5.2. Recall that the total advantage function can be written as: $A_{\boldsymbol{\nu}^*}^{\text{tot}}(\mathbf{s}, \mathbf{a}) = r(\mathbf{s}, \mathbf{a}) + \gamma \mathbb{E}_{\mathbf{s}'}[\nu^{\text{tot}*}(\mathbf{s}')] - \nu^{\text{tot}*}(\mathbf{s})$,

**Algorithm 1: MisoDICE** – Multi-Agent Imitation Policy Learning

---

1 Initialize local networks $(\theta_c, \theta_\nu, \theta_\pi)$, and mixing networks $(\eta, \phi)$;
2 **for** *a number of discriminator training steps* **do**
3     Train $(\theta_c, \eta)$ from Eq 9 with $c(\mathbf{s}, \mathbf{a})$ is computed in Eq. 13;
4 Compute occupancy ratio: $\frac{\rho_{\text{tot}}^E(\mathbf{s}, \mathbf{a})}{\rho_{\text{tot}}^U(\mathbf{s}, \mathbf{a})} = \frac{c(\mathbf{s}, \mathbf{a})}{1 - c(\mathbf{s}, \mathbf{a})}$ based on updated $(\theta_c, \eta)$;
5 **for** *a number of value function training steps* **do**
6     Train $(\theta_\nu, \phi)$ from Eq. 6 using the occupancy ratio estimated above;
7 Compute the weight $w_{\boldsymbol{\nu}^*}^{\text{tot}}(\mathbf{s}, \mathbf{a})$ using Eq. 8 w.r.t updated $(\theta_\nu, \phi)$;
8 **for** *a number of policy training steps* **do**
9     Update $\theta_\pi$ based on weighted BC in Eq. 11 and the weight $w_{\boldsymbol{\nu}^*}^{\text{tot}}(\mathbf{s}, \mathbf{a})$;
10 **return** *local policies $\pi_i$ for $i = 1, \ldots, n$*

---

where $\nu^{\text{tot}*}(\mathbf{s}) = \mathcal{M}_{\phi^*}[\boldsymbol{\nu}^*(\mathbf{s})]$ and the term $r(\mathbf{s}, \mathbf{a}) = \log \frac{\rho_{\text{tot}}^E(\mathbf{s}, \mathbf{a})}{\rho_{\text{tot}}^U(\mathbf{s}, \mathbf{a})}$ can be interpreted as a global reward function. We can now define the global $Q$-function as follows: $q^{\text{tot}*}(\mathbf{s}, \mathbf{a}) = r(\mathbf{s}, \mathbf{a}) + \gamma \mathbb{E}_{\mathbf{s}'}[\nu^{\text{tot}*}(\mathbf{s}')]$.

Next, let's assume that the optimal linear mixing network takes the form: $\mathcal{M}_{\phi^*}[\boldsymbol{\nu}^*(\mathbf{s})] = \sum_{i \in \mathcal{N}} \phi_i^* \nu_i^*(s_i) + \phi_0^*$, where $\phi^*$ and $\boldsymbol{\nu}^*$ are the solutions obtained from optimizing the training objective in Eq. 6; and the reward function admits a linear decomposition: $r(\mathbf{s}, \mathbf{a}) = \sum_{i \in \mathcal{N}} \phi_i^* r_i(s_i, a_i) + \phi_0^*$, noting that such decompositions are often feasible or can be approximated in practice. Substituting the decomposition into the advantage function yields: $A_{\boldsymbol{\nu}^*}^{\text{tot}}(\mathbf{s}, \mathbf{a}) = \sum_{i \in \mathcal{N}} \phi_i^* (q_i^*(s_i, a_i) - \nu_i^*(s_i)) + \gamma \phi_0^*$, where the local $Q$-function is defined as: $q_i^*(s_i, a_i) = r_i(s_i, a_i) + \gamma \mathbb{E}_{s_i'|s_i, a_i}[\nu_i^*(s_i')]$. This reformulation allows the occupancy ratio to be written in a decomposed, local form:

$$w_{\boldsymbol{\nu}^*}^{\text{tot}}(\mathbf{s}, \mathbf{a}) = e^{-1} \exp \Big( \frac{1}{1 + \alpha} \Big( \sum_{i \in \mathcal{N}} \phi_i^* (q_i^*(s_i, a_i) - \nu_i^*(s_i)) + \gamma \phi_0^* \Big) \Big). \tag{12}$$

This expression enables us to derive the following result, which shows how the optimal local policy can be explicitly expressed as a function of the local $Q$-function and value function:

**Proposition 5.5** (Local Policy as a Softmax over Local Functions). *Assume that the joint behavior policy of the union dataset $\mu^U(\mathbf{a}|\mathbf{s})$ is decomposable into local behavior policies $\mu_i^U(a_i \mid s_i)$. Let $\pi_i^*$ be the optimal solution to the local weighted BC objective. Then the optimal local policy can be expressed as:*

$$\pi_i^*(a_i \mid s_i) = \frac{1}{\Delta(s_i)} \exp \Big( \frac{\phi_i^*}{1 + \alpha} q_i^*(s_i, a_i) + \log \mu_i^U(a_i \mid s_i) \Big),$$

*where $\Delta(s_i) = \sum_{a_i \in \mathcal{A}_i} \exp \big( \frac{\phi_i^*}{1+\alpha} q_i^*(s_i, a_i) + \log \mu_i^U(a_i \mid s_i) \big)$, is the normalization constant, ensuring that $\pi_i^*(a_i \mid s_i)$ is a valid probability distribution over the local action space.*

## 6 Practical Algorithm

In implementation, we construct three types of local networks for every agent $i \in \mathcal{N}$: (i) local discriminator networks $\{c_i(s_i, a_i; \theta_c)\}$ (parameterized by $\theta_c$); (ii) local value networks $\{\nu_i(s_i; \theta_\nu)\}$ (parameterized by $\theta_\nu$); and (iii) local policy networks $\{\pi_i(s_i; \theta_\pi)\}$ (parameterized by $\theta_\pi$). In addition, there are two different mixing networks: $\mathcal{M}_\eta$ (with parameters $\eta$) and $\mathcal{M}_\nu$ (with parameters $\nu$) which are used to aggregate local discriminators and value functions, i.e.,

$$c(\mathbf{s}, \mathbf{a}) = \mathcal{M}_\eta[\{c_i(s_i, a_i; \theta_c)\}_{i \in \mathcal{N}}] \qquad \nu^{\text{tot}}(\mathbf{s}) = \mathcal{M}_\phi[\{\nu_i(s_i; \theta_\nu)\}_{i \in \mathcal{N}}] \tag{13}$$

Note that we use the same local network architectures with shared parameters $(\theta_c, \theta_\nu, \theta_\pi)$ for all agents for efficient training. The overview of our algorithm MisoDICE is illustrated in Algorithm 1.

## 7 Experiments

We run various experiments on challenging MARL benchmarks, including the StarCraft Multi-Agent Challenge version 1 (SMACv1) [46], and its successor, SMACv2 [13]. Both offer discrete action

Table 1: Final average returns for MisoDICE and baseline methods on SMACv2 tasks.

|  | BC | | | OMAPL | INDD | MARL-SL | VDN | MisoDICE (ours) |
|---|---|---|---|---|---|---|---|---|
|  | ($\beta=0.0$) | ($\beta=0.5$) | ($\beta=1.0$) | | | | | |
| 2c_vs_64zg | 8.5 ± 0.1 | 9.7 ± 0.3 | 12.6 ± 0.3 | 12.2 ± 0.4 | 14.6 ± 1.0 | 14.0 ± 1.6 | 12.7 ± 0.6 | **16.4 ± 1.3** |
| 5m_vs_6m | 5.0 ± 1.1 | 6.7 ± 0.0 | 6.1 ± 0.1 | 5.7 ± 0.2 | 6.7 ± 0.1 | 6.8 ± 0.1 | 6.2 ± 1.4 | **7.3 ± 0.1** |
| 6h_vs_8z | 7.0 ± 0.0 | 7.4 ± 0.0 | 7.2 ± 0.1 | 6.6 ± 0.2 | 7.5 ± 0.2 | 7.8 ± 0.1 | 8.2 ± 0.2 | **8.7 ± 0.2** |
| corridor | 1.5 ± 0.1 | 1.5 ± 0.2 | 4.3 ± 0.7 | 2.2 ± 1.3 | 4.4 ± 1.2 | 1.8 ± 0.2 | 4.7 ± 0.6 | **5.8 ± 0.8** |
| *Protoss* 5_vs_5 | 9.2 ± 0.1 | 11.7 ± 0.5 | 10.2 ± 0.5 | 9.6 ± 1.1 | 10.9 ± 0.1 | 11.6 ± 0.3 | 11.5 ± 0.2 | **12.4 ± 0.5** |
| 10_vs_10 | 10.3 ± 0.6 | 11.8 ± 0.5 | 10.6 ± 0.2 | 10.1 ± 0.9 | 11.0 ± 0.7 | 11.9 ± 0.4 | 12.4 ± 0.2 | **12.9 ± 0.2** |
| 10_vs_11 | 8.2 ± 0.4 | 9.6 ± 0.4 | 8.7 ± 0.3 | 8.5 ± 1.2 | 9.4 ± 0.4 | 9.9 ± 0.3 | 10.4 ± 0.1 | **10.7 ± 0.4** |
| 20_vs_20 | 10.1 ± 0.2 | 10.4 ± 0.5 | 10.5 ± 0.3 | 9.4 ± 0.4 | 11.4 ± 0.5 | 13.1 ± 0.4 | 12.1 ± 0.5 | **13.5 ± 0.5** |
| 20_vs_23 | 8.1 ± 0.2 | 8.6 ± 0.3 | 8.3 ± 0.2 | 7.9 ± 0.3 | 9.6 ± 0.3 | 9.6 ± 0.3 | 10.3 ± 0.4 | **10.6 ± 0.2** |
| *Terran* 5_vs_5 | 6.5 ± 0.8 | 8.1 ± 0.5 | 7.1 ± 0.6 | 6.2 ± 0.6 | 7.9 ± 0.5 | 8.1 ± 0.4 | 8.3 ± 1.0 | **9.1 ± 0.3** |
| 10_vs_10 | 6.6 ± 0.3 | 7.4 ± 0.4 | 6.7 ± 0.6 | 6.9 ± 1.1 | 7.6 ± 0.4 | 7.7 ± 0.2 | 8.0 ± 0.4 | **9.1 ± 1.3** |
| 10_vs_11 | 4.7 ± 0.2 | 5.7 ± 0.3 | 5.2 ± 0.3 | 4.2 ± 0.6 | 5.7 ± 0.5 | 5.7 ± 0.4 | 6.0 ± 0.2 | **6.4 ± 0.5** |
| 20_vs_20 | 6.9 ± 0.4 | 7.9 ± 0.8 | 6.7 ± 0.2 | 6.9 ± 0.5 | 8.0 ± 0.5 | 8.6 ± 0.3 | 8.2 ± 0.5 | **9.2 ± 0.6** |
| 20_vs_23 | 4.0 ± 0.3 | 5.1 ± 0.4 | 4.3 ± 0.3 | 4.3 ± 0.4 | 5.1 ± 0.4 | 5.1 ± 0.6 | **5.6 ± 0.3** | 5.6 ± 0.4 |
| *Zerg* 5_vs_5 | 5.7 ± 0.5 | 6.6 ± 0.4 | 5.9 ± 0.3 | 6.1 ± 0.5 | 6.4 ± 0.2 | 7.1 ± 0.5 | 7.1 ± 0.9 | **7.5 ± 0.1** |
| 10_vs_10 | 7.3 ± 0.1 | 8.7 ± 0.6 | 7.4 ± 0.7 | 6.8 ± 0.6 | 8.2 ± 0.2 | 9.0 ± 0.4 | 9.7 ± 0.5 | **10.2 ± 0.6** |
| 10_vs_11 | 7.3 ± 0.2 | 8.3 ± 0.4 | 7.3 ± 0.5 | 7.2 ± 0.4 | 8.0 ± 0.2 | 8.8 ± 0.4 | 9.1 ± 0.2 | **9.4 ± 0.3** |
| 20_vs_20 | 7.4 ± 0.6 | 9.0 ± 0.5 | 7.7 ± 0.2 | 6.9 ± 0.5 | 8.3 ± 0.4 | 8.8 ± 0.6 | 9.0 ± 0.5 | **10.2 ± 0.6** |
| 20_vs_23 | 7.1 ± 0.3 | 7.9 ± 0.3 | 7.0 ± 0.2 | 7.1 ± 0.4 | 8.2 ± 0.4 | 8.8 ± 0.2 | 8.7 ± 0.5 | **9.5 ± 0.2** |

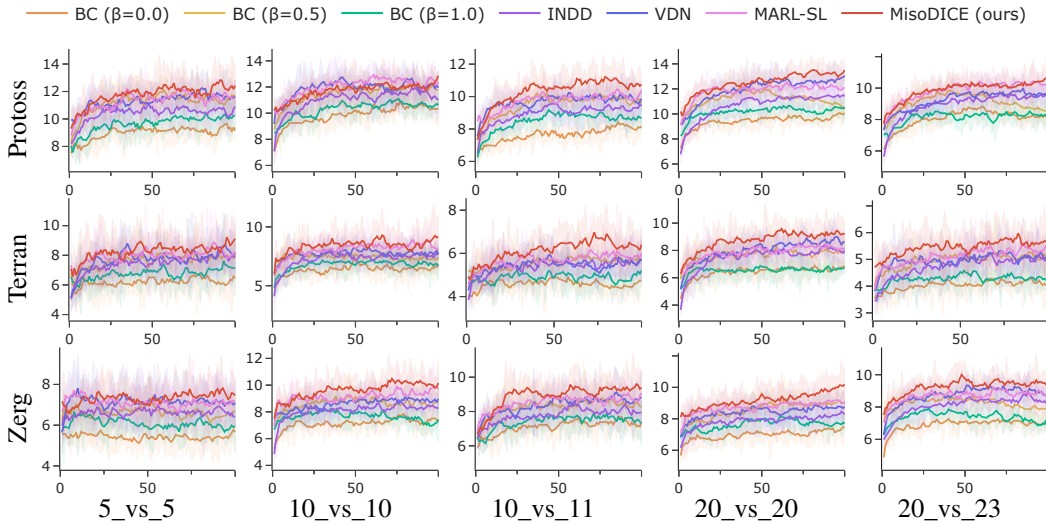

Figure 1: Learning curves of the average return for MisoDICE and baseline methods on SMACv2.

spaces and focus on decentralized micromanagement scenarios derived from StarCraft II [13, 46]. In these environments, each combat unit is controlled by an individual agent, and the team must learn cooperative strategies to defeat built-in AI opponents. SMACv2 introduces increased stochasticity and diversity by incorporating features like randomized start positions and unit types. In the appendix, we also include experiments on MAMuJoCo [11], another standard MARL benchmark. We note that LLMs struggle to provide meaningful preference feedback in MAMuJoCo due to a lack of contextual understanding. In this setting, we instead assume access to a rule-based oracle capable of providing preference feedback.

**Dataset** We use the offline data generated from [7, 56], providing trajectories categorized by quality (expert, medium, and poor) across various tasks in SMACv1 and SMACv2. For our experiments focused on learning from unlabeled, mixed-quality demonstrations, we curated a suboptimal dataset for each task by sampling 200 expert and 1000 poor trajectories from the datasets in [7, 56]. These were combined and shuffled to form an unlabeled dataset, $\mathcal{D}^U$. This construction reflects realistic scenarios where demonstration quality is heterogeneous and unknown a priori. The expert dataset $\mathcal{D}^E$, used in Phase 2 of MisoDICE, is derived from $\mathcal{D}^U$ based on our labeling procedure in Section 4.

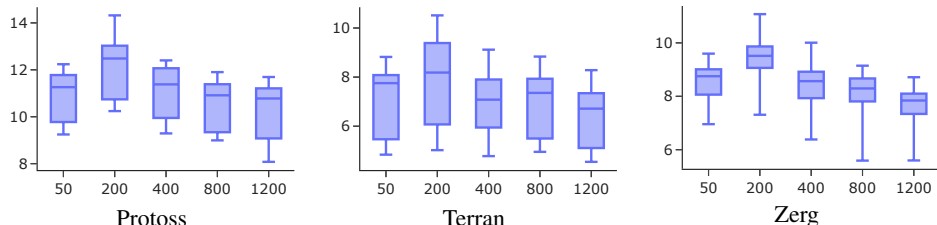

Figure 2: Box plots of final returns on SMACv2 by varying the number of top-k expert trajectories.

**Baselines** We compare MisoDICE performance against several key baselines. Multi-Agent BC (MA-BC) serves as a foundational baseline, adapted for mixed data via a coefficient $\beta$ that balances learning between the expert dataset ($\mathcal{D}^E$) and the union dataset ($\mathcal{D}^U$). We evaluate MA-BC under different $\beta$ values: $\beta = 1.0$ (expert-only), $\beta = 0.5$ (equal weighting), and $\beta = 0.0$ (union-only). The end-to-end preference-based OMAPL trains a policy directly from preference data generated by LLMs. Individual DemoDICE (INDD) applies the single-agent DemoDICE algorithm independently to each agent, using individual marginal distributions from $\mathcal{D}^E$ and $\mathcal{D}^U$. MARL with Supervised Reward (MARL-SL) follows a two-stage setup: a reward is learned from preference data via supervised learning, followed by MARL-based policy optimization. Finally, MisoDICE-VDN is an ablation using a standard Value Decomposition Network mixer (sum of local values) in place of MisoDICE's learnable mixing, isolating the contribution of the advanced value mixing architecture.

**Results** Table 1 provides a detailed quantitative comparison of the final average returns achieved by MisoDICE and the baselines across various SMACv2 maps. MisoDICE consistently achieves the highest mean return in nearly every scenario tested, significantly outperforming all baseline methods. The INDD baseline consistently underperforms compared to MisoDICE, highlighting the deficiency of independent learning and the necessity of coordinated value decomposition in multi-agent settings. While MARL-SL and MisoDICE-VDN represent stronger baselines by incorporating MARL optimization and value decomposition, their performance is still generally lower than MisoDICE. This suggests that MisoDICE's formulation for handling mixed-quality data, combined with its specific mixing architecture, provides a distinct advantage over learning a reward via supervised methods first or using a simpler additive decomposition like VDN. Furthermore, OMAPL's performance, derived directly from Phase 1 preferences, while sometimes competitive, does not match MisoDICE's final results, emphasizing the benefit of the subsequent IL phase (Phase 2) employed by MisoDICE.

Figure 1 plots the average return curve during training for MisoDICE and the baselines on the same set of SMACv2 tasks. These plots further reinforce the findings from Table 1. The learning curve for MisoDICE (shown in red) consistently rises above those of the other methods, demonstrating not only higher final performance but often faster and more stable learning across different game tasks.

**Ablation Studies** Figure 2 presents box plots of the final returns achieved on SMACv2 maps when varying the number of top trajectories ($k$) selected as the expert dataset ($\mathcal{D}^E$). The x-axis is the value of $k$, ranging from 50 to 1200, while the y-axis shows the distribution of returns achieved across multiple runs. For all three races depicted (Protoss, Terran, and Zerg), the plots generally show that performance tends to peak when a moderate number of top trajectories are selected as experts (around $k = 200$). Using too few ($k = 50$) or too many ($k = 800, 1200$) expert trajectories seems to result in lower median returns and potentially higher variance, although performance remains relatively robust across different $k$ values. This suggests that our data-labeling procedure successfully isolates high-quality trajectories, but the optimal amount of expert data required for the IL phase is neither the absolute minimum nor the maximum available, highlighting a trade-off in selecting the size of $\mathcal{D}^E$. In addition, we conduct several ablation studies which include investigating the effect of the hyperparameter $\alpha$ controlling the influence of suboptimal data, and the performance difference when using LLMs of varying capabilities (GPT-4o vs. GPT-4o-mini) and when employing a simple rule-based method for initial preference generation. For details, we refer to our appendix.

## 8    Conclusion

We address offline multi-agent IL with unlabeled, mixed-quality demonstrations. Our proposed two-stage framework, comprising a trajectory labeling pipeline and the MisoDICE algorithm, enables

robust learning from such restricted data. By leveraging LLMs and preference-based learning for effective labeling, and introducing a scalable, value-decomposed extension of DICE for multi-agent settings, our approach achieved both computational efficiency and global-local policy consistency. Empirical results on multi-agent benchmarks show that MisoDICE outperforms all baselines.

**Limitations and Future Work.** Our work is limited to cooperative settings. Extending this approach to competitive environments remains an open challenge for future investigation. Additionally, the reliance on LLMs for trajectory labeling has limitations in domains like MaMuJoCo, where abstract task semantics are difficult for LLMs to interpret accurately. Addressing these challenges opens up exciting directions for future research in scalable, generalizable imitation learning.

## Acknowledgments and Disclosure of Funding

This work is supported by the Lee Kong Chian Fellowship awarded to Tien Mai.

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

# APPENDIX

## Contents

# A Missing Proofs

## A.1 Proof of Proposition 5.1

*Proposition.* If the mixing network $\mathcal{M}_\phi[\boldsymbol{\nu}(\mathbf{s})]$ is a linear function of both $\boldsymbol{\nu}(\mathbf{s})$ and $\phi$, then the training objective $\mathcal{L}(\phi, \boldsymbol{\nu})$ is convex in both $\phi$ and $\boldsymbol{\nu}$.

*Proof.* We write the training objective as:

$$\mathcal{L}(\phi, \boldsymbol{\nu}) = (1-\gamma)\,\mathbb{E}_{\mathbf{s} \sim P_0}\left[\mathcal{M}_\phi[\boldsymbol{\nu}(\mathbf{s})]\right] + \mathbb{E}_{(\mathbf{s},\mathbf{a}) \sim \rho_{\text{tot}}^U}\left[w_{\boldsymbol{\nu}}^*(\mathbf{s},\mathbf{a})\left(A_{\boldsymbol{\nu}}^{\text{tot}}(\mathbf{s},\mathbf{a}) - (1+\alpha)\log w_{\boldsymbol{\nu}}^*(\mathbf{s},\mathbf{a})\right)\right],$$

where the total advantage function $A_{\boldsymbol{\nu}}^{\text{tot}}$ and the optimal weighting function $w_{\boldsymbol{\nu}}^*$ are defined as:

$$A_{\boldsymbol{\nu}}^{\text{tot}}(\mathbf{s},\mathbf{a}) = \log\frac{\rho_{\text{tot}}^E(\mathbf{s},\mathbf{a})}{\rho_{\text{tot}}^U(\mathbf{s},\mathbf{a})} + \gamma\sum_{\mathbf{s}'}\mathcal{T}(\mathbf{s}'\mid\mathbf{s},\mathbf{a})\mathcal{M}_\phi[\boldsymbol{\nu}(\mathbf{s}')] - \mathcal{M}_\phi[\boldsymbol{\nu}(\mathbf{s})], \tag{14}$$

$$w_{\boldsymbol{\nu}}^*(\mathbf{s},\mathbf{a}) = \exp\left(\frac{A_{\boldsymbol{\nu}}^{\text{tot}}(\mathbf{s},\mathbf{a})}{1+\alpha} - 1\right). \tag{15}$$

We can simplify the objective as:

$$\mathcal{L}(\phi, \boldsymbol{\nu}) = (1-\gamma)\,\mathbb{E}_{\mathbf{s} \sim P_0}\left[\mathcal{M}_\phi[\boldsymbol{\nu}(\mathbf{s})]\right] + (1+\alpha)\mathbb{E}_{(\mathbf{s},\mathbf{a}) \sim \rho_{\text{tot}}^U}\left[\exp\left(\frac{A_{\boldsymbol{\nu}}^{\text{tot}}(\mathbf{s},\mathbf{a})}{1+\alpha} - 1\right)\right].$$

We now observe that if $\mathcal{M}_\phi[\boldsymbol{\nu}(\mathbf{s})]$ is linear in both $\boldsymbol{\nu}(\mathbf{s})$ and $\phi$, then each term in $A_{\boldsymbol{\nu}}^{\text{tot}}(\mathbf{s},\mathbf{a})$ is also linear in $\boldsymbol{\nu}(\mathbf{s})$ and $\phi$, since:

- The log-ratio term $\log\frac{\rho_{\text{tot}}^E}{\rho_{\text{tot}}^U}$ is constant with respect to both $\phi$ and $\boldsymbol{\nu}$,

- The remaining terms are composed of expectations of linear functions (due to the linearity of $\mathcal{M}_\phi[\cdot]$) and thus remain linear.

As a result, $A_{\boldsymbol{\nu}}^{\text{tot}}(\mathbf{s},\mathbf{a})$ is linear, and the exponential of a linear function is convex. Since the outer expectation is a linear operator, and the sum of convex and linear functions remains convex, the entire objective $\mathcal{L}(\phi, \boldsymbol{\nu})$ is convex in both $\phi$ and $\boldsymbol{\nu}(\mathbf{s})$. This concludes the convexity analysis.

$\square$

## A.2 Proof of Proposition 5.2

*Proposition.* If the mixing network $\mathcal{M}_\phi[\boldsymbol{\nu}]$ is a multi-layer feedforward network with $\boldsymbol{\nu}$ as input (even with convex activation functions), the objective $\mathcal{L}(\phi, \boldsymbol{\nu})$ becomes non-convex in $\boldsymbol{\nu}$.

*Proof.* Now assume that $\mathcal{M}_\phi[\boldsymbol{\nu}(\mathbf{s})]$ is constructed using a multi-layer feedforward neural network with convex activation functions (e.g., ReLU, softplus). This implies that $\mathcal{M}_\phi[\boldsymbol{\nu}(\mathbf{s})]$ is a convex function of both $\phi$ and $\boldsymbol{\nu}(\mathbf{s})$, but it is not linear. We aim to show that, under this assumption, the objective $\mathcal{L}(\phi, \boldsymbol{\nu})$ is no longer convex in the joint parameters $(\phi, \boldsymbol{\nu})$.

Recall that the simplified objective can be written as:

$$\mathcal{L}(\phi, \boldsymbol{\nu}) = (1-\gamma)\,\mathbb{E}_{\mathbf{s} \sim P_0}\left[\mathcal{M}_\phi[\boldsymbol{\nu}(\mathbf{s})]\right] + (1+\alpha)\,\mathbb{E}_{(\mathbf{s},\mathbf{a}) \sim \rho_{\text{tot}}^U}\left[\exp\left(\frac{A_{\boldsymbol{\nu}}^{\text{tot}}(\mathbf{s},\mathbf{a})}{1+\alpha} - 1\right)\right],$$

where $A_{\boldsymbol{\nu}}^{\text{tot}}(\mathbf{s},\mathbf{a})$ is defined as:

$$A_{\boldsymbol{\nu}}^{\text{tot}}(\mathbf{s},\mathbf{a}) = \log\frac{\rho_{\text{tot}}^E(\mathbf{s},\mathbf{a})}{\rho_{\text{tot}}^U(\mathbf{s},\mathbf{a})} + \gamma\sum_{\mathbf{s}'}\mathcal{T}(\mathbf{s}'\mid\mathbf{s},\mathbf{a})\mathcal{M}_\phi[\boldsymbol{\nu}(\mathbf{s}')] - \mathcal{M}_\phi[\boldsymbol{\nu}(\mathbf{s})].$$

The log-ratio term is constant with respect to $(\phi, \boldsymbol{\nu})$, so the dependence of $\mathcal{L}$ on these variables arises entirely through $\mathcal{M}_\phi[\boldsymbol{\nu}(\cdot)]$. Now observe:

- By assumption, the mapping $(\phi, \boldsymbol{\nu}) \mapsto \mathcal{M}_\phi[\boldsymbol{\nu}(\mathbf{s})]$ is convex.

- Consequently, the term $\gamma \sum_{\mathbf{s}'} \mathcal{T}(\mathbf{s}' \mid \mathbf{s}, \mathbf{a}) \mathcal{M}_\phi[\boldsymbol{\nu}(\mathbf{s}')]$ is convex in $(\phi, \boldsymbol{\nu})$, as it is a non-negative linear combination of convex functions.

- However, the term $-\mathcal{M}_\phi[\boldsymbol{\nu}(\mathbf{s})]$ is concave in $(\phi, \boldsymbol{\nu})$, and hence $A_{\boldsymbol{\nu}}^{\text{tot}}(\mathbf{s}, \mathbf{a})$ is generally the difference of convex functions, which is not necessarily convex.

- Since the exponential function is convex and monotonically increasing, applying it to a non-convex function (like $A_{\boldsymbol{\nu}}^{\text{tot}}/(1 + \alpha) - 1$) does not preserve convexity in general.

Hence, we conclude that the objective $\mathcal{L}(\phi, \boldsymbol{\nu})$ is not convex in general when $\mathcal{M}_\phi[\boldsymbol{\nu}(\mathbf{s})]$ is convex but not linear.

$\square$

## A.3 Proof of Proposition 5.3

*Proposition.* If the mixing network $\mathcal{M}_\eta[\mathbf{c}(\mathbf{s}, \mathbf{a})]$ is linear in $\mathbf{c}(\mathbf{s}, \mathbf{a})$, then the training objective $J(\mathbf{c}, \eta)$ for occupancy ratio estimation is concave in both $\mathbf{c}$ and $\eta$.

*Proof.* We write the objective as:

$$J(\mathbf{c}, \eta) = \mathbb{E}_{\rho_{\text{tot}}^E} \left[ \log \mathcal{M}_\eta[\mathbf{c}(\mathbf{s}, \mathbf{a})] \right] + \mathbb{E}_{\rho_{\text{tot}}^U} \left[ \log \left( 1 - \mathcal{M}_\eta[\mathbf{c}(\mathbf{s}, \mathbf{a})] \right) \right],$$

where $\mathcal{M}_\eta[\cdot]$ is linear in $\eta$ and $\mathbf{c}(\mathbf{s}, \mathbf{a})$. Now observe:

- The composition $\log \mathcal{M}_\eta[\mathbf{c}(\mathbf{s}, \mathbf{a})]$ is a concave function in $\mathcal{M}_\eta[\mathbf{c}(\mathbf{s}, \mathbf{a})]$, and hence concave in both $\eta$ and $\mathbf{c}$.

- Similarly, $\log(1 - \mathcal{M}_\eta[\mathbf{c}(\mathbf{s}, \mathbf{a})])$ is also concave in $1 - \mathcal{M}_\eta[\mathbf{c}(\mathbf{s}, \mathbf{a})]$, and thus in $\eta$ and $\mathbf{c}$.

- Since expectations preserve concavity, both terms in $J(\mathbf{c}, \eta)$ are concave in $(\mathbf{c}, \eta)$.

Here we note that if $\mathcal{M}_\eta[\mathbf{c}(\mathbf{s}, \mathbf{a})]$ is a convex (but non-linear) function of $\eta$ and/or $\mathbf{c}$, such as when using a neural network with convex activations (e.g., ReLU, softplus). In this case:

- The function $\log \mathcal{M}_\eta[\mathbf{c}(\mathbf{s}, \mathbf{a})]$ is the composition of a concave function ($\log$) with a convex function. This composition is in general *not concave*.

- Likewise, $\log(1 - \mathcal{M}_\eta[\mathbf{c}(\mathbf{s}, \mathbf{a})])$ is the composition of a concave function with a concave function, which again does not preserve concavity.

Therefore, $J(\mathbf{c}, \eta)$ is no longer guaranteed to be concave in $(\mathbf{c}, \eta)$ when $\mathcal{M}_\eta[\cdot]$ is convex but not linear. $\square$

## A.4 Proof of Proposition 5.4

*Proposition.* Assume the global policy space $\Pi$ is factorizable, i.e., $\Pi = \left\{ \boldsymbol{\pi}_{\text{tot}} \mid \exists \{\pi_i\}_{i \in \mathcal{N}} \text{ such that } \boldsymbol{\pi}_{\text{tot}}(\mathbf{s}, \mathbf{a}) = \prod_{i \in \mathcal{N}} \pi_i(a_i \mid s_i) \right\}$. Let $\{\pi_i^*\}_{i \in \mathcal{N}}$ and $\boldsymbol{\pi}_{\text{tot}}^*$ be the solutions to the local and global weighted BCs, respectively. Then for any $(\mathbf{s}, \mathbf{a})$, it holds that:

$$\boldsymbol{\pi}_{\text{tot}}^*(\mathbf{s}, \mathbf{a}) = \prod_{i \in \mathcal{N}} \pi_i^*(a_i \mid s_i).$$

*Proof.* For notational simplicity, let $\Phi(\boldsymbol{\pi}_{\text{tot}})$ be the objective function of the global WBC problem:

$$\Phi(\boldsymbol{\pi}_{\text{tot}}) = \mathbb{E}_{\mathbf{s}, \mathbf{a} \sim \rho_{\text{tot}}^U} \left[ w_{\boldsymbol{\nu}^*}^{\text{tot}}(\mathbf{s}, \mathbf{a}) \log \boldsymbol{\pi}_{\text{tot}}(\mathbf{a}|\mathbf{s}) \right].$$

Since we are seeking a decomposable policy $\boldsymbol{\pi}_{\text{tot}}(\mathbf{a}|\mathbf{s}) = \prod_{i \in \mathcal{N}} \pi_i(a_i|s_i)$, we have, for any $\boldsymbol{\pi}_{\text{tot}} \in \Pi$ such that $\boldsymbol{\pi}_{\text{tot}} = \prod_i \pi_i$:

$$\Phi(\boldsymbol{\pi}_{\text{tot}}) = \mathbb{E}_{\mathbf{s},\mathbf{a} \sim \rho_{\text{tot}}^U} \left[ w_{\boldsymbol{\nu}^*}^{\text{tot}}(\mathbf{s}, \mathbf{a}) \log \prod_i \pi_i(a_i|s_i) \right]$$

$$= \sum_{i \in \mathcal{N}} \left\{ \mathbb{E}_{\mathbf{s},\mathbf{a} \sim \rho_{\text{tot}}^U} \left[ w_{\boldsymbol{\nu}^*}^{\text{tot}}(\mathbf{s}, \mathbf{a}) \log \pi_i(a_i|s_i) \right] \right\}$$

$$\overset{(a)}{\leq} \sum_{i \in \mathcal{N}} \left\{ \mathbb{E}_{\mathbf{s},\mathbf{a} \sim \rho_{\text{tot}}^U} \left[ w_{\boldsymbol{\nu}^*}^{\text{tot}}(\mathbf{s}, \mathbf{a}) \log \pi_i^*(a_i|s_i) \right] \right\}$$

$$= \mathbb{E}_{\mathbf{s},\mathbf{a} \sim \rho_{\text{tot}}^U} \left[ w_{\boldsymbol{\nu}^*}^{\text{tot}}(\mathbf{s}, \mathbf{a}) \log \widetilde{\pi}_{\text{tot}}(\mathbf{a}|\mathbf{s}) \right],$$

where $\widetilde{\pi}_{\text{tot}} = \prod_i \pi_i^*$, and $(a)$ holds because each $\pi_i^*$ is optimal for the corresponding local WBC problem. Thus, we have $\Phi(\boldsymbol{\pi}_{\text{tot}}) \leq \Phi(\widetilde{\pi}_{\text{tot}})$ for any $\boldsymbol{\pi}_{\text{tot}} \in \Pi$, implying that $\widetilde{\pi}_{\text{tot}}$ is also optimal for the global WBC. This establishes the global-local-consistency, as desired.

$\square$

## A.5 Proof of Proposition 5.5

*Proposition.* Assume that the joint behavior policy of the union dataset $\mu^U(\mathbf{a}|\mathbf{s})$ is decomposable into local behavior policies $\mu_i^U(a_i \mid s_i)$. Let $\pi_i^*$ be the optimal solution to the local weighted BC objective. Then the optimal local policy can be expressed as:

$$\pi_i^*(a_i \mid s_i) = \frac{1}{\Delta(s_i)} \exp \left( \frac{\phi_i^*}{1+\alpha} q_i^*(s_i, a_i) + \log \mu_i^U(a_i \mid s_i) \right),$$

where $\Delta(s_i) = \sum_{a_i \in \mathcal{A}_i} \exp \left( \frac{\phi_i^*}{1+\alpha} q_i^*(s_i, a_i) + \log \mu_i^U(a_i \mid s_i) \right)$, is the normalization constant, ensuring that $\pi_i^*(a_i \mid s_i)$ is a valid probability distribution over the local action space.

*Proof.* We first recall that the total weighting function can be written as:

$$w_{\boldsymbol{\nu}^*}^{\text{tot}}(\mathbf{s}, \mathbf{a}) = e^{-1} \exp \left( \frac{1}{1+\alpha} \left( \sum_{i \in \mathcal{N}} \phi_i^* \left( q_i^*(s_i, a_i) - \nu_i^*(s_i) \right) + \gamma \phi_0^* \right) \right).$$

This allows us to express the local weighted behavior cloning (BC) objective for each agent $i$ as:

$$\max_{\pi_i} \left\{ \mathbb{E}_{\mathbf{s},\mathbf{a} \sim \rho_{\text{tot}}^U} \left[ w_{\boldsymbol{\nu}^*}^{\text{tot}}(\mathbf{s}, \mathbf{a}) \log \pi_i(a_i \mid s_i) \right] \right\}.$$

For each state $\mathbf{s}$, the corresponding sub-training objective is:

$$\sum_{\mathbf{a}} \mu^U(\mathbf{a} \mid \mathbf{s}) \left[ w_{\boldsymbol{\nu}^*}^{\text{tot}}(\mathbf{s}, \mathbf{a}) \log \pi_i(a_i \mid s_i) \right]$$

$$= \sum_{\mathbf{a}} \prod_i \mu_i^U(a_i \mid s_i) \left[ \exp \left( \frac{1}{1+\alpha} \left( \sum_{i \in \mathcal{N}} \phi_i^* \left( q_i^*(s_i, a_i) - \nu_i^*(s_i) \right) + \gamma \phi_0^* \right) \right) \log \pi_i(a_i \mid s_i) \right]$$

$$= \sum_{\mathbf{a}} \left[ \exp \left( \frac{1}{1+\alpha} \left( \sum_{i \in \mathcal{N}} \phi_i^* q_i^*(s_i, a_i) + (1+\alpha) \log \mu_i^U(a_i \mid s_i) + \text{const} \right) \right) \log \pi_i(a_i \mid s_i) \right],$$

where the constant term (including $\nu_i^*(s_i)$ and $\phi_0^*$) does not depend on $a_i$, and thus can be ignored for optimizing $\pi_i$.

Since only terms involving $a_i$ affect the optimality of the local policy $\pi_i(a_i \mid s_i)$, we isolate the relevant part and simplify the training objective for each agent $i$ as:

$$\sum_{a_i} \left[ \exp \left( \frac{1}{1+\alpha} \left( \phi_i^* q_i^*(s_i, a_i) + (1+\alpha) \log \mu_i^U(a_i \mid s_i) \right) \right) \log \pi_i(a_i \mid s_i) \right].$$

This is a weighted log-likelihood objective, where the weights are shaped by the exponentiated advantage-like term. The maximization of this expression yields the following closed-form solution:

$$\pi_i^*(a_i \mid s_i) = \frac{\mu_i^U(a_i \mid s_i) \exp \left( \frac{\phi_i^*}{1+\alpha} q_i^*(s_i, a_i) \right)}{\sum_{a_i'} \mu_i^U(a_i' \mid s_i) \exp \left( \frac{\phi_i^*}{1+\alpha} q_i^*(s_i, a_i') \right)} = \frac{1}{\Delta(s_i)} \mu_i^U(a_i \mid s_i) \exp \left( \frac{\phi_i^*}{1+\alpha} q_i^*(s_i, a_i) \right).$$

where

$$\Delta(s_i) = \sum\nolimits_{a_i} \exp\big(\frac{\phi_i^*}{1+\alpha}\, q_i^*(s_i, a_i) + \log \mu_i^U(a_i \mid s_i)\big),$$

as desired.

$\square$

# B  Additional Details

## B.1  Unlabeled Imperfect Dataset

A significant challenge in advancing offline multi-agent imitation learning (MAIL) from mixed-quality data is the lack of standardized benchmarks. Currently, there are no publicly available datasets that specifically provide suboptimal or imperfect demonstrations tailored for multi-agent reinforcement learning (MARL) settings. To address this gap and facilitate research in this crucial area, we constructed our own datasets.

Our approach leverages the offline datasets generated by O-MAPL [6]. The O-MAPL framework contributes a valuable resource by providing distinct datasets categorized by quality - expert, medium, and poor - for a range of MARL tasks. Specifically, for each task within the MaMujoco, SMACv1, and SMACv2 environments, O-MAPL offers 1000 trajectories for each quality level (expert, medium, and poor).

For the experiments, we specifically curate a suboptimal dataset designed to test the algorithm's ability to learn from mixed-quality demonstrations where the quality is not explicitly labeled. To form this dataset, we sample 200 expert trajectories and 1000 poor trajectories from the O-MAPL datasets for each considered task. These selected trajectories are then combined and shuffled thoroughly to create a single, unlabeled suboptimal dataset. This process ensures that we operate on a dataset that realistically mirrors scenarios where trajectory quality is heterogeneous and unknown beforehand, compelling the agent to discern and learn from the more effective behaviors embedded within the mixed data.

We further detail the general framework of using an expert dataset ($\mathcal{D}^E$) and a mixed-quality dataset ($\mathcal{D}^{Mix}$), with our constructed dataset serving as $\mathcal{D}^U$ in a setting where expert demonstrations within this mix are not pre-annotated. The specific MaMujoco and SMAC environments used in our evaluations are detailed in Table 2.

## B.2  Rule-based vs. LLM-based Trajectory Ranking Techniques

To effectively utilize unlabeled datasets in our multi-agent imitation learning framework, a critical step involves ranking trajectories to identify and select high-quality examples, which then form the expert dataset ($\mathcal{D}^E$). This ranking is achieved by first learning an O-MAPL (Offline Multi-agent Preference Learning) model to recover the underlying reward function from pairwise preferences. Subsequently, these recovered rewards enable the ranking of all trajectories in the unlabeled dataset.

The training of O-MAPL necessitates a preference dataset ($\mathcal{P}$) for each task. Each instance in this dataset consists of a pair of trajectories ($\sigma_1, \sigma_2$) and a label indicating which trajectory is preferred. Echoing the data generation methodologies in O-MAPL, we construct these crucial preference datasets using two distinct approaches: a rule-based method and an LLM-based method. For this purpose, we randomly sample 2,000 trajectory pairs from the unlabeled dataset for each task, and then apply one of the following methods to assign preference labels.

### B.2.1  Rule-based Preference Labeling

In the rule-based approach, we operate under the assumption that the cumulative return for each trajectory within a selected pair is known or can be accurately estimated. The preference assignment is straightforward: the trajectory yielding a higher return is labeled as preferred. This method leverages the intuition that higher returns often correlate with more successful task completion.

However, this rule-based method is not without its limitations, particularly in cooperative multi-agent settings. The accuracy can be compromised because, in certain scenarios, individual agents might adopt strategies that maximize their individual rewards without contributing to, or even at

the detriment of, the team's overall objective and cooperative success. For instance, in complex environments like StarCraft Multi-Agent Challenge (SMAC), which involves cooperative multi-agent gameplay to defeat enemies, agents might excessively focus on attacking enemies to boost their individual reward metrics. Such behavior can lead to a loss of the game due to neglecting crucial strategic elements, such as maintaining adequate health points, armor regeneration, managing weapon cooldowns, or effective unit positioning and coordination. Consequently, a trajectory with a higher summed individual return might not always represent superior cooperative behavior.

### B.2.2 LLM-based Preference Labeling

To introduce more nuanced and potentially more accurate preference signals, we also employ an LLM-based method. For this, we leverage the capabilities of a large language model, specifically GPT-4o, to act as an expert labeler. The goal is to achieve better generalization in preference assessment and thereby increase the accuracy of the subsequent preference learning phase.

Following the methodology described in studies such as O-MAPL for SMAC environments, we construct prompts for the LLM that include detailed information extracted from the global state of each trajectory in a pair. This information typically encompasses key metrics like the remaining health points and shields of allied and enemy units, the number of allied and enemy deaths, relative positions of units, weapon cooldown times, agent types, and the semantics of their actions. The LLM is then tasked to evaluate which trajectory demonstrates superior gameplay or strategy based on this comprehensive contextual information. This approach has shown potential in generating rich and cost-effective preference data, which can significantly enhance environment understanding and policy learning in complex multi-agent tasks. As noted in the MisoDICE framework, LLMs can be instrumental in inferring expert-like segments even from entirely unlabeled trajectory datasets.

### B.2.3 Reward Recovery and Trajectory Ranking

Once the preference datasets are established using either the rule-based or LLM-based method, they are used to train the O-MAPL model. After training O-MAPL, we utilize an inverse preference learning technique, specifically leveraging the relationship $R = Q - \gamma V$ (an adaptation of the inverse soft Bellman operator, $r(s,a) = Q_{tot}(s,a) - \gamma \mathbb{E}_{s'} V_{tot}(s')$ as seen in MaxEnt RL frameworks), to

Table 2: Overview of our datasets.

| Instances | | | Trajectories | Samples | Agents | State dim | Obs dim | Action dim |
|---|---|---|---|---|---|---|---|---|
| MaMujoco | | Hopper-v2 | 1.2K | 459.2K | 3 | 126 | 14 | 1 |
| | | Ant-v2 | 1.2K | 1200K | 2 | 452 | 113 | 4 |
| | | HalfCheetah-v2 | 1.2K | 1200K | 6 | 828 | 23 | 1 |
| SMACv1 | | 2c_vs_64zg | 1.2K | 43.2K | 2 | 1350 | 478 | 70 |
| | | 5m_vs_6m | 1.2K | 28.3K | 5 | 780 | 124 | 12 |
| | | 6h_vs_8z | 1.2K | 31.8K | 6 | 1278 | 172 | 14 |
| | | corridor | 1.2K | 71.8K | 6 | 2610 | 346 | 30 |
| SMACv2 | protoss | 5_vs_5 | 1.2K | 76.9K | 5 | 130 | 92 | 11 |
| | | 10_vs_10 | 1.2K | 80.2K | 10 | 310 | 182 | 16 |
| | | 10_vs_11 | 1.2K | 73.3K | 10 | 327 | 191 | 17 |
| | | 20_vs_20 | 1.2K | 78.3K | 20 | 820 | 362 | 26 |
| | | 20_vs_23 | 1.2K | 70.4K | 20 | 901 | 389 | 29 |
| | terran | 5_vs_5 | 1.2K | 60.6K | 5 | 120 | 82 | 11 |
| | | 10_vs_10 | 1.2K | 61.7K | 10 | 290 | 162 | 16 |
| | | 10_vs_11 | 1.2K | 58.2K | 10 | 306 | 170 | 17 |
| | | 20_vs_20 | 1.2K | 64.7K | 20 | 780 | 322 | 26 |
| | | 20_vs_23 | 1.2K | 57.1K | 20 | 858 | 346 | 29 |
| | zerg | 5_vs_5 | 1.2K | 34.3K | 5 | 120 | 82 | 11 |
| | | 10_vs_10 | 1.2K | 38.1K | 10 | 290 | 162 | 16 |
| | | 10_vs_11 | 1.2K | 37.3K | 10 | 306 | 170 | 17 |
| | | 20_vs_20 | 1.2K | 40.5K | 20 | 780 | 322 | 26 |
| | | 20_vs_23 | 1.2K | 38.5K | 20 | 858 | 346 | 29 |

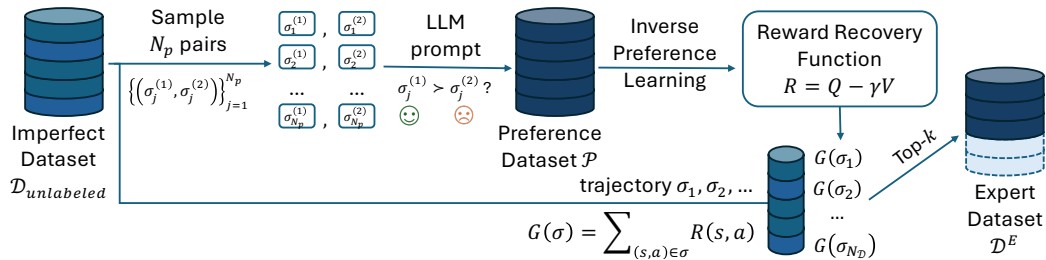

Figure 3: Generalized Expert Dataset Identification via Preference Learning

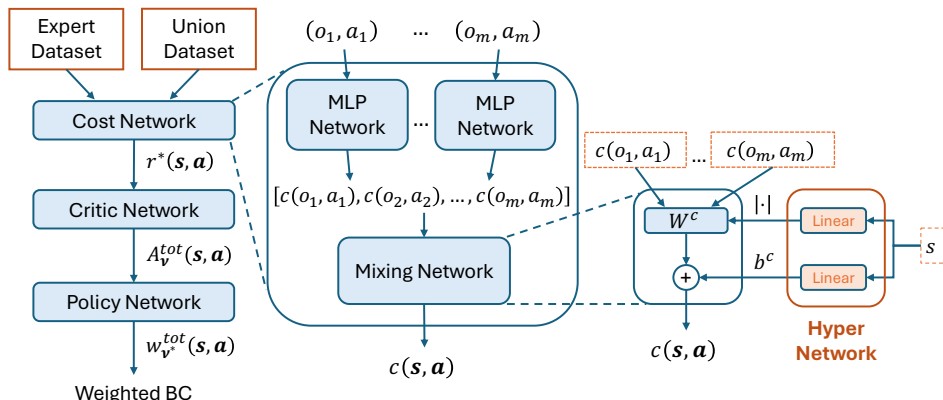

Figure 4: Multi-Agent Imitation Policy Learning

recover the reward for each state-action pair within every trajectory in the unlabeled dataset. With these recovered rewards, we can then compute the total return for each trajectory and subsequently rank all trajectories.

Finally, from the ranked lists generated by each of the two preference labeling approaches (rule-based and LLM-based), we select the top-ranked trajectories to form the respective expert datasets, denoted as $\mathcal{D}^E_{\text{rule}}$ and $\mathcal{D}^E_{\text{LLM}}$. These datasets are then used for the downstream imitation learning process in MisoDICE.

### B.3 Baselines

To rigorously evaluate MisoDICE, we compare its performance against several key baselines. These are selected to assess MisoDICE's efficacy in multi-agent imitation learning from mixed-quality demonstrations and to ablate the contributions of its core components, particularly its learnable mixing architecture.

### B.3.1 Behavioral Cloning (BC) for Mixed-Quality Data ($\beta = 1.0, 0.5, 0.0$)

Multi-Agent Behavioral Cloning (MA-BC) serves as a fundamental imitation learning baseline, where individual agent policies $\pi_i(a_i|o_i)$ are trained via supervised learning to maximize the log-likelihood of demonstrated actions. Following the approach for handling mixed-quality data in DemoDICE, we adapt MA-BC using a coefficient $\beta$. This coefficient balances learning between a limited expert dataset ($D^E$) and a larger union dataset ($D^U = D^E \cup D^{Mix}$), where $D^{Mix}$ contains uncurated, potentially suboptimal trajectories. The objective for the set of agent policies is:

$$J_{BC}(\beta) = -\beta \mathbb{E}_{(o,a)\sim D^E}\left[\sum_{i=1}^{N} \log \pi_i(a_i|o_i)\right] - (1-\beta)\mathbb{E}_{(o,a)\sim D^U}\left[\sum_{i=1}^{N} \log \pi_i(a_i|o_i)\right]$$

We evaluate three variants:

---

**Algorithm 2: MisoDICE**: Phase 1 – Expert Dataset Identification via Preference Learning

---

**Input:** Unlabeled dataset $\mathcal{D}_{\text{unlabeled}}$; O-MAPL parameters $(\psi_q^{\text{OM}}, \psi_v^{\text{OM}}, \theta^{\text{OM}})$; Preference method
      `PrefMethod`; Parameters for `PrefMethod`; Number of preference pairs $N_p$; Number of
      expert trajectories $K_{\text{expert}}$

**Output:** Expert dataset $\mathcal{D}^E$

  /* 1.   Generate Preference Dataset                                             */

1 Sample $N_p$ trajectory pairs $\mathcal{S}_{\text{pairs}}$ from $\mathcal{D}_{\text{unlabeled}}$;

2 Generate preference dataset $\mathcal{P}$ using $\mathcal{S}_{\text{pairs}}$ and the specified `PrefMethod`;

  /* 2.   Train O-MAPL for Reward Recovery                             */

3 **for** *a number of training steps* **do**

4     Update O-MAPL parameters $(\psi_q^{\text{OM}}, \psi_v^{\text{OM}}, \theta^{\text{OM}})$ using $\mathcal{P}$;

5     Maximize preference likelihood $\mathcal{L}_{\text{O-MAPL}}$ and minimize Extreme-V loss $\mathcal{J}_{\text{O-MAPL}}$;

  /* 3.   Recover Rewards and Select Expert Dataset                  */

6 Estimate $R(s, a)$ for each $(s, a) \in \mathcal{D}_{\text{unlabeled}}$ using $Q^{\text{OM}} - \gamma V^{\text{OM}}$;

7 **foreach** *trajectory $\sigma \in \mathcal{D}_{unlabeled}$* **do**

8     Compute return $G(\sigma) = \sum_{(s,a) \in \sigma} R(s, a)$;

9 Rank all trajectories by $G(\sigma)$ in descending order;

10 Select top-$K_{\text{expert}}$ trajectories as $\mathcal{D}^E$;

11 **return** $\mathcal{D}^E$

---

- **BC ($\beta = 1.0$):** This variant imitates expert data ($D^E$) exclusively. It learns by maximizing the log-likelihood of actions in the expert dataset.

- **BC ($\beta = 0.5$):** This provides an equal weighting between expert and mixed data imitation. It balances learning from the expert dataset and the larger, mixed-quality union dataset.

- **BC ($\beta = 0.0$):** This variant imitates the entire mixed dataset ($D^U$) without differentiation. It learns by maximizing the log-likelihood of actions in the combined expert and suboptimal dataset.

These variants establish a performance benchmark against direct imitation, highlighting how effectively MisoDICE utilizes mixed-quality data compared to standard BC approaches that merely re-weight datasets.

### B.3.2 Offline Multi-Agent Preference Learning (OMAPL)

This baseline refers to the O-MAPL algorithm, which is an end-to-end preference-based learning framework for cooperative MARL. O-MAPL directly learns policies from preference data (Phase 1 in MisoDICE context) without explicitly modeling a reward function. It leverages the relationship between reward functions and soft Q-functions within the MaxEnt RL framework. The learning process uses a multi-agent value decomposition strategy under the Centralized Training with Decentralized Execution (CTDE) paradigm. For this baseline, the policy is learned directly from the preference dataset generated in Phase 1. Comparing MisoDICE to OMAPL is intended to demonstrate the necessity and benefit of MisoDICE's Phase 2, which involves reward recovery and policy learning using the DICE framework.

### B.3.3 Individual DemoDICE (INDD)

This baseline applies the single-agent DemoDICE algorithm independently to each agent in the multi-agent system. DemoDICE is an offline imitation learning algorithm designed to learn from both expert and imperfect demonstrations by optimizing a policy within the space of stationary distributions. It achieves this by reducing a potentially unstable minimax optimization problem to a direct convex optimization. In the INDD baseline, each agent $i$ independently solves its own DemoDICE objective. This involves learning a policy $\pi_i$ by focusing on its individual marginal stationary distribution derived from $D_i^E$ (its portion of expert trajectories) and $D_i^U$ (its portion of the union dataset), without explicit joint state-action modeling or coordinated value/policy learning across agents. INDD contrasts MisoDICE's centralized training with value decomposition against a naive

multi-agent extension of the single-agent DemoDICE. This comparison is crucial for demonstrating the benefits of MisoDICE's joint learning approach and coordinated credit assignment.

### B.3.4 MARL with Supervised Reward (MARL-SL)

This baseline represents MisoDICE using Supervised Learning to learn reward recovery from Phase 1. In this approach, a reward function is first learned via supervised learning techniques from the preference data generated in Phase 1. Then, a MARL algorithm is applied to optimize the policy with respect to this learned reward. This can be seen as a two-phase approach, similar to some existing preference-based RL methods. This baseline helps to evaluate the effectiveness of MisoDICE's end-to-end DICE-based reward and policy learning in Phase 2 compared to a more decoupled supervised reward learning followed by MARL.

### B.3.5 MisoDICE-VDN

This is an ablated version of our MisoDICE algorithm, where the proposed learnable mixer (with a hypernetwork) is replaced by a standard Value Decomposition Network (VDN) mixer. MisoDICE-VDN retains the core framework of MisoDICE, including its adaptation of DICE principles for multi-agent imitation learning from mixed-quality data. However, for value decomposition, it uses a simple VDN approach where the joint value function (or its DICE-equivalent, such as the global Lagrange multipliers $\nu^{tot}$ or advantage function $A_{\nu}^{tot}$ as seen in DICE literature like ComaDICE) is computed as a direct summation of individual agent values: $\mathcal{M}(\{\nu_i(o_i)\}) = \sum_i \nu_i(o_i)$. This is a simpler alternative to MisoDICE's architecture, which employs a more expressive learnable mixer using a hypernetwork to generate state-dependent mixing weights, similar in spirit to advanced mixers like QMIX or those described in O-MAPL. By comparing MisoDICE against MisoDICE-VDN, we can directly assess the performance impact and importance of MisoDICE's sophisticated hypernetwork-based mixing architecture.

### B.4 Hyperparameters

Table 3: Hyperparameters used in MisoDICE experiments.

| Hyperparameter | Value |
|---|---|
| Optimizer | Adam |
| Learning rate (actor) | 3e-4 (for SMAC) |
| | 1e-5 (for MaMujoco) |
| Learning rate (critic) | 3e-4 |
| Tau ($\tau$, soft update target rate) | 0.005 |
| Alpha ($\alpha$) | 0.05 |
| Gamma ($\gamma$, discount factor) | 0.99 |
| Number of minibatch | 512 |
| Agent hidden dimension | 256 |
| Mixer hidden dimension | 64 |
| Number of seeds | 4 |
| Number of episodes for each evaluation step | 32 |
| Number of epochs | 100 |

All experiments are implemented using PyTorch and run in parallel on a single NVIDIA® H100 NVL Tensor Core GPU. Given the large size of the offline datasets for each instance, we compress all datasets into the H5 format using the `h5py` library.

We develop two versions of our MisoDICE agent to accommodate different domain characteristics: one for continuous action spaces (MaMujoco) and another for discrete action spaces (SMACv1 & SMACv2). The primary distinction lies in the output distributions of the policy networks. For continuous control tasks, we employ a Gaussian distribution (`torch.distributions.Normal`) to model the actions. For discrete action environments, we utilize a Categorical distribution (`torch.distributions.Categorical`). Notably, in the discrete version, the probability for each action of an agent is computed by applying a softmax function over the set of currently available actions for that agent, rather than over the entire action space. Consequently, actions not available to

an agent at a given state are assigned a probability of zero. This approach ensures a more accurate calculation of the log-likelihood by the actor.

The hyperparameters used in our experiments are detailed in Table 3.

## B.5 Algorithm

The MisoDICE framework is implemented in two main phases, detailed in Algorithm 2 and Algorithm 3 respectively. The first phase focuses on identifying expert trajectories from an unlabeled dataset, while the second phase performs multi-agent imitation learning using these identified trajectories.

---

**Algorithm 3: MisoDICE**: Phase 2 – Multi-Agent Imitation Policy Learning

---

**Input:** Expert dataset $\mathcal{D}^E$, unlabeled/mixed dataset $\mathcal{D}_{\text{unlabeled}}$ (or $\mathcal{D}^{\text{Mix}}$), discriminator networks $c_k, \eta_{\text{disc}}$, local value networks $\nu_k$, value mixing network $\phi_{\text{val}}$, local policy networks $\eta_k$, regularization coefficient $\alpha_{\text{Miso}}$, learning rates

**Output:** Optimized local policies $\pi_k(a_k|o_k; \eta_k)$

1 Let $\mathcal{D}^U = \mathcal{D}^E \cup \mathcal{D}_{\text{unlabeled}}$;

    /* 1.  Train Occupancy Ratio Estimator (Discriminator)         */

2 **for** *a number of discriminator training steps* **do**

3     | Update $c_k, \eta_{\text{disc}}$ by optimizing discriminator loss $J(c, \eta_{\text{disc}})$ using samples from $\mathcal{D}^E$ and $\mathcal{D}^U$;
                                                     // Ref:  MisoDICE Sec 5.3 / Eq. 9

4 Compute log occupancy ratio: $r^*(s, a) = \log \frac{c^*(s,a)}{1 - c^*(s,a)}$;

    /* 2.  Train MisoDICE Value Functions                              */

5 **for** *a number of value function training steps* **do**

6     | Update $\nu_k$ and $\phi_{\text{val}}$ by minimizing $\mathcal{L}(\phi_{\text{val}}, \{\nu_k\})$;
                                          // Ref:  MisoDICE Sec 5.2 / Eq. 6

7 Compute $w^*_{\nu^*}(s, a) = \exp\left(\frac{A^*_{\nu^*}(s,a)}{1 + \alpha_{\text{Miso}}} - 1\right)$;
                                          // Ref:  MisoDICE Sec 5.2 / Eq. 8

    /* 3.  Train MisoDICE Local Policies via Weighted Behavior Cloning     */

8 **for** *a number of policy training steps* **do**

9     | **foreach** *agent* $k \in \mathcal{N}$ **do**

10         | Update $\eta_k$ by maximizing weighted BC loss $\mathcal{L}_{\text{WBC}}(\eta_k \mid w^*_{\nu^*}(s, a))$ using $\mathcal{D}^U$;
                                        // Ref:  MisoDICE Sec 5.4 / Eq. 11

11 **return** $\pi_k(a_k|o_k; \eta_k)$ *for* $k = 1, \ldots, n$

---

## B.6 Experiment Setup

This section details the experimental environments and evaluation metrics used to assess the performance of MisoDICE.

### B.6.1 Environments

To comprehensively evaluate MisoDICE, we utilize standard and challenging benchmarks widely adopted in cooperative multi-agent reinforcement learning (MARL) research. These environments are sourced from the Multi-Agent MuJoCo (MaMujoco) suite [11], the StarCraft Multi-Agent Challenge version 1 (SMACv1) [46], and its successor, the StarCraft Multi-Agent Challenge version 2 (SMACv2) [13]. MaMujoco provides continuous control tasks where multiple agents, typically different parts of a single robot, must learn to coordinate their actions to achieve a common goal [11]. These tasks are known for their complex physics-based dynamics. In contrast, SMACv1 and SMACv2 offer discrete action spaces and focus on decentralized micromanagement scenarios derived from the real-time strategy game StarCraft II [46]. In these environments, each combat unit is controlled by an individual agent, and the team must learn cooperative strategies to defeat opponent units controlled by built-in game AI. SMACv2, in particular, introduces increased stochasticity and diversity compared to SMACv1 by incorporating features like randomized start positions and unit types, presenting a more demanding test for MARL algorithms [13]. The selection of these diverse

benchmarks allows for testing MisoDICE's capabilities across different action spaces (continuous for MaMujoco, discrete for SMAC) and varied cooperative challenges.

### B.6.2 Evaluation Metrics

To rigorously evaluate the performance of MisoDICE and compare it against baseline methods, we use standard evaluation metrics common in MARL research. The primary metrics reported are:

- **Returns**: This metric measures the average cumulative rewards achieved by the agents across multiple evaluation episodes. Higher returns generally indicate better policy performance in maximizing the task-specific objectives. We report the mean and standard deviation of returns.

- **Win Rates**: Specifically for competitive or mixed cooperative-competitive environments like the StarCraft Multi-Agent Challenge (SMAC) tasks, this metric evaluates the percentage of episodes where the team of agents achieves a win condition against opponents. We report the mean and standard deviation of win rates.

Performance for each method is typically averaged over multiple random seeds to ensure statistical robustness, and evaluation curves often depict performance trends throughout the training process. For MisoDICE, evaluations were run for 32 episodes for each evaluation step across 4 seeds.

## C   Evaluation

### C.1   Additional Experimental Results

In this section, we present the main experimental results evaluating the performance of MisoDICE. Our evaluation aims to demonstrate MisoDICE's effectiveness in learning robust multi-agent policies from mixed-quality demonstrations, particularly focusing on its ability to handle scenarios with limited or unlabeled expert data. We conduct comprehensive experiments across a range of challenging cooperative multi-agent benchmarks, including tasks from the StarCraft Multi-Agent Challenge (SMACv1 and SMACv2) and Multi-Agent MuJoCo (MaMujoco).

The results will show that MisoDICE consistently outperforms various strong baselines across these diverse environments. A key aspect of our evaluation is the comparison of MisoDICE's performance when expert trajectories are identified using two distinct approaches: a traditional rule-based ranking method and, more notably, our proposed LLM-based technique for inferring expert-like segments from entirely unlabeled datasets. This latter approach showcases a novel application of LLMs to significantly enhance multi-agent imitation learning in practical settings where explicit expert annotations are unavailable.

We compare MisoDICE against several relevant baselines. These include standard Behavioral Cloning (BC) variants that imitate different compositions of the available data, an Independent DemoDICE (INDD) baseline applying single-agent imitation learning to each agent, and a supervised MARL approach (MARL-SL) that learns a reward function from preferences before policy optimization. Furthermore, we evaluate against OMAPL, representing the policy learned directly from the preference data generated in the first phase of our framework, to highlight the benefits of MisoDICE's second phase DICE-based policy learning. An ablated version, MisoDICE-VDN, which uses a simpler value decomposition network, is also included to demonstrate the efficacy of our proposed learnable mixing architecture.

The tables 4 5 and figures 5 6 7 8 provide detailed quantitative results in terms of average returns and win rates, alongside learning curves. These results will substantiate MisoDICE's superior performance, its robustness to the quality and quantity of expert data, and the significant potential of leveraging LLMs for expert data identification in the challenging domain of multi-agent imitation learning from suboptimal demonstrations.

### C.2   Ablation Study 1: Impact of the Number of Identified Expert Trajectories (Top-k)

In this ablation study, we investigate the sensitivity of MisoDICE to the number of trajectories selected to form the expert dataset, $\mathcal{D}^E$. As outlined in Algorithm 2, the first phase of MisoDICE

involves ranking all trajectories in the unlabeled dataset $\mathcal{D}_{unlabeled}$ (or $\mathcal{D}^{Mix}$) using a preference-based model (O-MAPL) trained on LLM-generated (or rule-based) preferences. From this ranked

Table 4: Comparison of final average returns for MisoDICE and baseline methods on SMACv1 and SMACv2 tasks when using an LLM-based preference labeling approach.

|  |  | BC | | OMAPL | INDD | MARL-SL | VDN | MisoDICE |
|  | ($\beta = 0.0$) | ($\beta = 0.5$) | ($\beta = 1.0$) | | | | | (ours) |
|---|---|---|---|---|---|---|---|---|
| 2c_vs_64zg | 8.5 ± 0.1 | 9.7 ± 0.3 | 12.6 ± 0.3 | 12.2 ± 0.4 | 14.6 ± 1.0 | 14.0 ± 1.6 | 12.7 ± 0.6 | 16.4 ± 1.3 |
| 5m_vs_6m | 5.0 ± 1.1 | 6.7 ± 0.0 | 6.1 ± 0.1 | 5.7 ± 0.2 | 6.7 ± 0.1 | 6.8 ± 0.1 | 6.2 ± 1.4 | 7.3 ± 0.1 |
| 6h_vs_8z | 7.0 ± 0.0 | 7.4 ± 0.0 | 7.2 ± 0.1 | 6.6 ± 0.2 | 7.5 ± 0.2 | 7.8 ± 0.1 | 8.2 ± 0.2 | 8.7 ± 0.2 |
| corridor | 1.5 ± 0.1 | 1.5 ± 0.2 | 4.3 ± 0.7 | 2.2 ± 1.3 | 4.4 ± 1.2 | 1.8 ± 0.2 | 4.7 ± 0.6 | 5.8 ± 0.8 |
| **Protoss** 5_vs_5 | 9.2 ± 0.1 | 11.7 ± 0.5 | 10.2 ± 0.5 | 9.6 ± 1.1 | 10.9 ± 0.1 | 11.6 ± 0.3 | 11.5 ± 0.2 | 12.4 ± 0.5 |
| 10_vs_10 | 10.3 ± 0.6 | 11.8 ± 0.5 | 10.6 ± 0.2 | 10.1 ± 0.9 | 11.0 ± 0.7 | 11.9 ± 0.4 | 12.4 ± 0.2 | 12.9 ± 0.2 |
| 10_vs_11 | 8.2 ± 0.4 | 9.6 ± 0.4 | 8.7 ± 0.3 | 8.5 ± 1.2 | 9.4 ± 0.4 | 9.9 ± 0.3 | 10.4 ± 0.1 | 10.7 ± 0.4 |
| 20_vs_20 | 10.1 ± 0.2 | 10.4 ± 0.5 | 10.5 ± 0.3 | 9.4 ± 0.4 | 11.4 ± 0.5 | 13.1 ± 0.4 | 12.1 ± 0.5 | 13.5 ± 0.5 |
| 20_vs_23 | 8.1 ± 0.2 | 8.6 ± 0.3 | 8.3 ± 0.2 | 7.9 ± 0.3 | 9.6 ± 0.3 | 9.6 ± 0.3 | 10.3 ± 0.4 | 10.6 ± 0.2 |
| **Terran** 5_vs_5 | 6.5 ± 0.8 | 8.1 ± 0.5 | 7.1 ± 0.6 | 6.2 ± 0.6 | 7.9 ± 0.5 | 8.1 ± 0.4 | 8.3 ± 1.0 | 9.1 ± 0.3 |
| 10_vs_10 | 6.6 ± 0.3 | 7.4 ± 0.4 | 6.7 ± 0.6 | 6.9 ± 1.1 | 7.6 ± 0.4 | 7.7 ± 0.2 | 8.0 ± 0.4 | 9.1 ± 1.3 |
| 10_vs_11 | 4.7 ± 0.2 | 5.7 ± 0.3 | 5.2 ± 0.3 | 4.2 ± 0.6 | 5.7 ± 0.5 | 5.7 ± 0.4 | 6.0 ± 0.2 | 6.4 ± 0.5 |
| 20_vs_20 | 6.9 ± 0.4 | 7.9 ± 0.8 | 6.7 ± 0.2 | 6.9 ± 0.5 | 8.0 ± 0.5 | 8.6 ± 0.3 | 8.2 ± 0.5 | 9.2 ± 0.6 |
| 20_vs_23 | 4.0 ± 0.3 | 5.1 ± 0.4 | 4.3 ± 0.3 | 4.3 ± 0.4 | 5.1 ± 0.4 | 5.1 ± 0.6 | 5.6 ± 0.3 | 5.6 ± 0.4 |
| **Zerg** 5_vs_5 | 5.7 ± 0.5 | 6.6 ± 0.4 | 5.9 ± 0.3 | 6.1 ± 0.5 | 6.4 ± 0.2 | 7.1 ± 0.5 | 7.1 ± 0.9 | 7.5 ± 0.1 |
| 10_vs_10 | 7.3 ± 0.1 | 8.7 ± 0.6 | 7.4 ± 0.7 | 6.8 ± 0.6 | 8.2 ± 0.2 | 9.0 ± 0.4 | 9.7 ± 0.5 | 10.2 ± 0.6 |
| 10_vs_11 | 7.3 ± 0.2 | 8.3 ± 0.4 | 7.3 ± 0.5 | 7.2 ± 0.4 | 8.0 ± 0.2 | 8.8 ± 0.4 | 9.1 ± 0.2 | 9.4 ± 0.3 |
| 20_vs_20 | 7.4 ± 0.6 | 9.0 ± 0.5 | 7.7 ± 0.2 | 6.9 ± 0.5 | 8.3 ± 0.4 | 8.8 ± 0.6 | 9.0 ± 0.5 | 10.2 ± 0.6 |
| 20_vs_23 | 7.1 ± 0.3 | 7.9 ± 0.3 | 7.0 ± 0.2 | 7.1 ± 0.4 | 8.2 ± 0.4 | 8.8 ± 0.2 | 8.7 ± 0.5 | 9.5 ± 0.2 |

Table 5: Comparison of final average win rates for MisoDICE and baseline methods on SMACv1 and SMACv2 tasks when using an LLM-based preference labeling approach.

|  |  | BC | | OMAPL | INDD | MARL-SL | VDN | MisoDICE |
|  | ($\beta = 0.0$) | ($\beta = 0.5$) | ($\beta = 1.0$) | | | | | (ours) |
|---|---|---|---|---|---|---|---|---|
| 2c_vs_64zg | 0.2 ± 0.2 | 0.5 ± 0.3 | 8.9 ± 2.9 | 3.9 ± 3.4 | 11.7 ± 5.5 | 10.6 ± 6.0 | 2.7 ± 1.5 | 13.0 ± 9.0 |
| 5m_vs_6m | 0.2 ± 0.4 | 0.9 ± 0.6 | 0.1 ± 0.1 | 0.0 ± 0.0 | 0.2 ± 0.2 | 1.1 ± 0.8 | 0.9 ± 0.9 | 1.2 ± 0.5 |
| 6h_vs_8z | 0.2 ± 0.2 | 0.0 ± 0.0 | 0.2 ± 0.3 | 0.0 ± 0.0 | 0.1 ± 0.1 | 1.0 ± 0.6 | 1.2 ± 0.1 | 1.1 ± 0.8 |
| corridor | 0.1 ± 0.1 | 0.6 ± 0.7 | 0.3 ± 0.4 | 0.0 ± 0.0 | 0.1 ± 0.1 | 0.9 ± 0.6 | 0.7 ± 0.7 | 1.4 ± 0.6 |
| **Protoss** 5_vs_5 | 13.8 ± 2.7 | 17.5 ± 3.8 | 14.2 ± 2.4 | 14.1 ± 10.2 | 12.4 ± 3.1 | 15.6 ± 4.5 | 10.8 ± 1.6 | 20.7 ± 0.9 |
| 10_vs_10 | 12.1 ± 2.3 | 12.7 ± 0.9 | 11.3 ± 2.9 | 11.7 ± 3.4 | 8.9 ± 1.4 | 11.8 ± 4.3 | 9.5 ± 1.7 | 14.1 ± 2.1 |
| 10_vs_11 | 2.1 ± 0.9 | 3.5 ± 2.0 | 1.8 ± 0.6 | 0.8 ± 1.4 | 2.0 ± 0.4 | 3.5 ± 0.3 | 2.9 ± 0.9 | 4.7 ± 0.3 |
| 20_vs_20 | 4.5 ± 1.8 | 3.4 ± 1.9 | 7.0 ± 2.5 | 3.1 ± 3.8 | 5.2 ± 1.3 | 8.6 ± 2.0 | 5.2 ± 1.9 | 11.0 ± 3.2 |
| 20_vs_23 | 1.5 ± 0.8 | 0.8 ± 0.4 | 1.8 ± 0.8 | 0.8 ± 1.4 | 2.4 ± 0.9 | 2.0 ± 0.9 | 2.6 ± 1.0 | 3.8 ± 2.0 |
| **Terran** 5_vs_5 | 10.1 ± 2.3 | 12.8 ± 1.3 | 13.7 ± 3.5 | 10.2 ± 4.6 | 10.6 ± 1.3 | 9.7 ± 1.9 | 10.4 ± 3.2 | 14.2 ± 3.1 |
| 10_vs_10 | 7.3 ± 2.0 | 7.7 ± 2.2 | 7.9 ± 2.4 | 9.4 ± 3.8 | 8.0 ± 2.4 | 8.2 ± 1.6 | 7.0 ± 1.9 | 12.0 ± 1.7 |
| 10_vs_11 | 1.9 ± 0.8 | 3.1 ± 1.4 | 2.5 ± 1.1 | 0.8 ± 1.4 | 1.8 ± 0.5 | 2.6 ± 1.2 | 2.4 ± 0.8 | 4.2 ± 1.6 |
| 20_vs_20 | 4.5 ± 1.3 | 7.4 ± 2.5 | 4.4 ± 0.6 | 4.7 ± 3.5 | 5.8 ± 0.7 | 7.1 ± 1.4 | 4.5 ± 1.5 | 8.8 ± 1.5 |
| 20_vs_23 | 0.6 ± 1.0 | 1.2 ± 0.9 | 0.7 ± 0.7 | 0.0 ± 0.0 | 0.8 ± 0.4 | 1.3 ± 0.7 | 1.3 ± 1.2 | 1.8 ± 1.4 |
| **Zerg** 5_vs_5 | 6.2 ± 0.8 | 5.6 ± 1.1 | 5.9 ± 0.5 | 6.2 ± 5.8 | 6.0 ± 1.4 | 6.7 ± 1.1 | 7.0 ± 1.8 | 7.9 ± 1.0 |
| 10_vs_10 | 4.5 ± 1.4 | 6.9 ± 1.8 | 5.0 ± 1.8 | 3.9 ± 3.4 | 4.6 ± 1.0 | 5.7 ± 1.2 | 5.2 ± 0.6 | 8.9 ± 2.1 |
| 10_vs_11 | 5.7 ± 1.9 | 5.8 ± 1.9 | 5.2 ± 1.3 | 2.3 ± 1.4 | 4.7 ± 1.6 | 6.0 ± 1.9 | 3.4 ± 0.7 | 6.8 ± 1.1 |
| 20_vs_20 | 0.8 ± 0.9 | 1.5 ± 0.6 | 1.3 ± 0.8 | 0.0 ± 0.0 | 0.2 ± 0.2 | 1.2 ± 0.5 | 0.6 ± 0.2 | 2.4 ± 0.6 |
| 20_vs_23 | 1.2 ± 0.9 | 0.7 ± 0.3 | 1.2 ± 0.6 | 1.6 ± 1.6 | 1.4 ± 0.5 | 1.9 ± 0.9 | 1.8 ± 1.1 | 2.7 ± 0.4 |

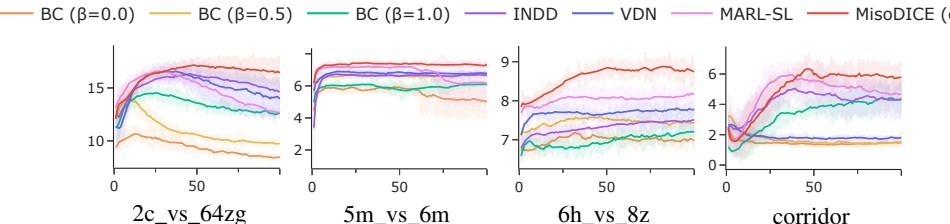

Figure 5: Learning curves of the average return for MisoDICE and baseline methods on SMACv1 tasks when using an LLM-based preference labeling approach.

list, the top-$K_{expert}$ trajectories are selected to constitute the expert dataset $\mathcal{D}^E$, which is then used alongside the full mixed dataset $\mathcal{D}^U$ in the second phase for policy learning.

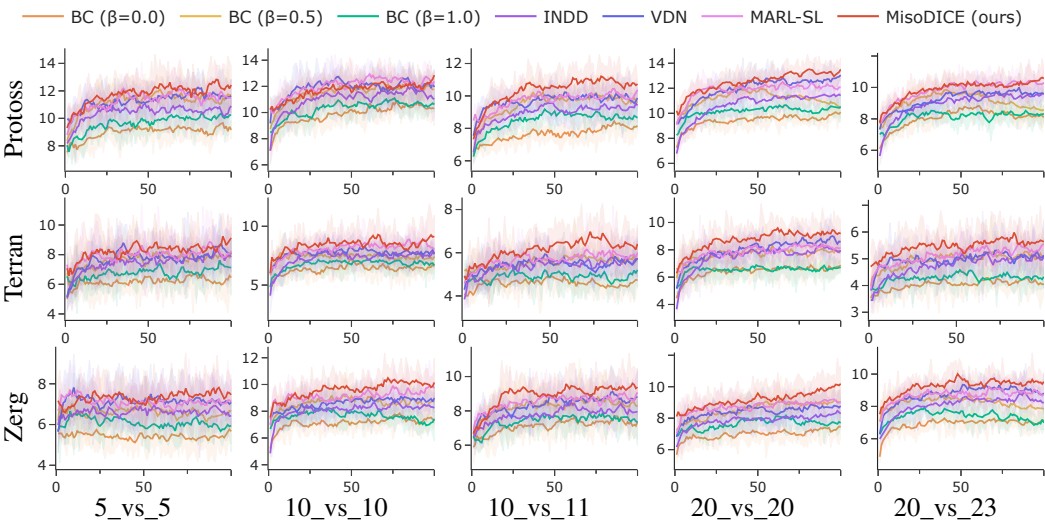

Figure 6: Learning curves of the average return for MisoDICE and baseline methods on SMACv2 tasks when using an LLM-based preference labeling approach.

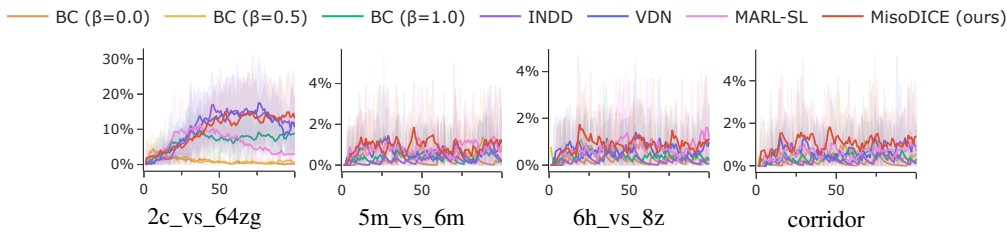

Figure 7: Learning curves of the average win rates for MisoDICE and baseline methods on SMACv1 tasks when using an LLM-based preference labeling approach.

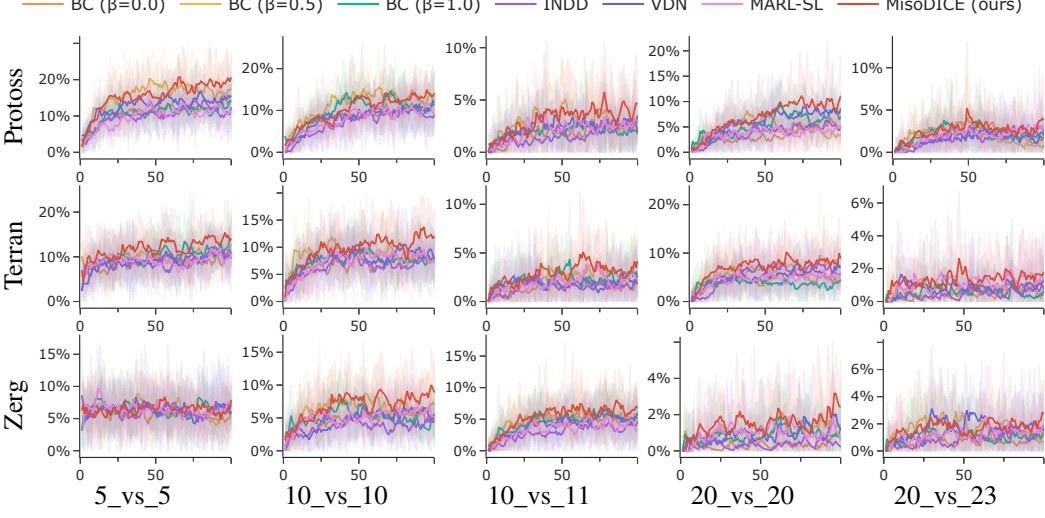

Figure 8: Learning curves of the average win rates for MisoDICE and baseline methods on SMACv2 tasks when using an LLM-based preference labeling approach.

The choice of $K_{expert}$ (referred to as "top-k") is critical. A smaller $K_{expert}$ might yield a higher-purity expert dataset but could be too limited, potentially missing some useful expert behaviors. Conversely, a larger $K_{expert}$ might include a more diverse set of expert behaviors but risks incorporating more suboptimal trajectories, thereby degrading the quality of $\mathcal{D}^E$. This study examines how varying $K_{expert}$ affects the final performance of MisoDICE. We evaluate MisoDICE's performance (in terms of returns and win rates) across a range of $K_{expert}$ values, specifically $K_{expert} \in \{50, 200, 400, 800, 1200\}$, using the LLM-based trajectory ranking method. The results, detailed in Table 6 and Table 7 (and visualized in Figure 9 11 and Figure 10 12), provide insights into the robustness of MisoDICE to the size and composition of the automatically curated expert dataset. This helps understand the trade-offs involved in selecting $K_{expert}$ and its impact on learning effective policies from mixed-quality demonstrations.

Table 6: Ablation Study 1: Final average returns for MisoDICE on SMACv1 and SMACv2 tasks, varying the number of top-k expert trajectories ($K_{expert}$) selected via LLM-based preference labeling.

| | | topk=50 | topk=200 | topk=400 | topk=800 | topk=1200 |
|---|---|---|---|---|---|---|
| | 2c_vs_64zg | 11.8 ± 1.4 | 16.4 ± 1.3 | 9.7 ± 0.1 | 10.8 ± 0.7 | 10.4 ± 0.5 |
| | 5m_vs_6m | 6.9 ± 0.4 | 7.3 ± 0.1 | 6.8 ± 0.2 | 6.5 ± 0.1 | 6.4 ± 0.2 |
| | 6h_vs_8z | 7.9 ± 0.1 | 8.7 ± 0.2 | 7.8 ± 0.2 | 7.5 ± 0.1 | 7.3 ± 0.1 |
| | corridor | 3.1 ± 1.0 | 5.8 ± 0.8 | 1.9 ± 0.1 | 1.7 ± 0.2 | 1.7 ± 0.1 |
| Protoss | 5_vs_5 | 11.9 ± 0.2 | 12.4 ± 0.5 | 11.3 ± 0.2 | 10.9 ± 0.2 | 10.9 ± 0.5 |
| | 10_vs_10 | 11.8 ± 0.1 | 12.9 ± 0.2 | 12.1 ± 0.3 | 11.5 ± 0.3 | 11.2 ± 0.4 |
| | 10_vs_11 | 9.6 ± 0.1 | 10.7 ± 0.4 | 9.5 ± 0.3 | 9.3 ± 0.5 | 9.2 ± 0.3 |
| | 20_vs_20 | 11.4 ± 0.5 | 13.5 ± 0.5 | 12.1 ± 0.1 | 11.4 ± 0.3 | 11.2 ± 0.3 |
| | 20_vs_23 | 9.8 ± 0.3 | 10.6 ± 0.2 | 10.0 ± 0.2 | 9.3 ± 0.2 | 8.7 ± 0.4 |
| Terran | 5_vs_5 | 8.0 ± 0.3 | 9.1 ± 0.3 | 7.3 ± 0.4 | 7.7 ± 0.4 | 7.0 ± 0.5 |
| | 10_vs_10 | 8.4 ± 0.4 | 9.1 ± 1.3 | 8.1 ± 0.5 | 7.6 ± 0.7 | 7.0 ± 0.4 |
| | 10_vs_11 | 5.5 ± 0.1 | 6.4 ± 0.5 | 6.1 ± 0.2 | 5.7 ± 0.4 | 5.4 ± 0.5 |
| | 20_vs_20 | 8.0 ± 0.4 | 9.2 ± 0.6 | 8.2 ± 0.6 | 8.0 ± 0.5 | 7.9 ± 0.5 |
| | 20_vs_23 | 5.1 ± 0.2 | 5.6 ± 0.4 | 5.1 ± 0.2 | 5.2 ± 0.2 | 4.8 ± 0.2 |
| Zerg | 5_vs_5 | 7.4 ± 0.4 | 7.5 ± 0.1 | 6.9 ± 0.5 | 6.4 ± 0.5 | 6.2 ± 0.5 |
| | 10_vs_10 | 9.2 ± 0.3 | 10.2 ± 0.6 | 9.0 ± 0.3 | 8.8 ± 0.2 | 8.3 ± 0.2 |
| | 10_vs_11 | 8.5 ± 0.4 | 9.4 ± 0.3 | 8.5 ± 0.0 | 8.2 ± 0.2 | 7.6 ± 0.3 |
| | 20_vs_20 | 8.9 ± 0.1 | 10.2 ± 0.6 | 8.8 ± 0.9 | 8.3 ± 0.5 | 8.1 ± 0.6 |
| | 20_vs_23 | 8.4 ± 0.4 | 9.5 ± 0.2 | 8.8 ± 0.2 | 8.5 ± 0.2 | 8.0 ± 0.1 |

Figure 9: Ablation Study 1: Box plots of final average returns for MisoDICE on SMACv2 tasks, showing the impact of varying the number of top-k expert trajectories ($K_{expert}$) selected via LLM-based preference labeling. Results are presented for Protoss, Terran, and Zerg game races.

## C.3 Ablation Study 2: Impact of the Suboptimal Data Influence Hyperparameter

In this ablation study, we examine the influence of the hyperparameter $\alpha$ on the performance of MisoDICE. As defined in our learning objective (Equation 2), $\alpha$ plays a crucial role by controlling the relative importance of the union dataset $\mathcal{D}^U$ (composed of both expert $\mathcal{D}^E$ and mixed-quality $\mathcal{D}^{Mix}$ trajectories) compared to the purely expert dataset $\mathcal{D}^E$. Specifically, the term $\alpha D_{KL}(\rho^\pi_{tot}||\rho^U_{tot})$ encourages the learned policy $\pi_{tot}$ to remain close to the overall distribution of the union dataset, while the $D_{KL}(\rho^\pi_{tot}||\rho^E_{tot})$ term pushes the policy to mimic the identified expert behaviors.

The choice of $\alpha$ therefore dictates how MisoDICE balances learning from the potentially limited but higher-quality expert demonstrations against leveraging the broader, though more suboptimal,

data distribution of the union dataset. A small $\alpha$ would emphasize strict imitation of $\mathcal{D}^E$, potentially suffering if $\mathcal{D}^E$ is very scarce or narrowly focused. Conversely, a large $\alpha$ would regularize the policy more strongly towards the mixed-quality data in $\mathcal{D}^U$, which could be beneficial for broader state-action coverage but risks learning suboptimal behaviors if the quality of $\mathcal{D}^U$ is low overall.

This study investigates MisoDICE's sensitivity to different values of $\alpha \in \{0.00, 0.05, 0.10, 0.50, 1.00, 10.0\}$. By evaluating the algorithm's performance (returns and win rates) across this range, as detailed in Table 8 and Table 9, we aim to understand the optimal balance and the robustness of MisoDICE to this critical hyperparameter when learning from demonstrations of varying quality. Our findings also help to validate the choice of $\alpha = 0.05$ used in our main experiments (see Table 3).

### C.4 Ablation Study 3: Comparison of LLM Capabilities for Preference Labeling

A core component of MisoDICE, particularly when expert demonstrations are not pre-annotated, is the utilization of Large Language Models (LLMs) to infer expert-like segments from unlabeled trajectories. This is achieved in the first phase by using an LLM to generate preference labels for pairs of trajectories, which then informs the O-MAPL model to rank demonstrations and identify the expert dataset $\mathcal{D}^E$.

In this ablation study, we investigate the impact of the chosen LLM's capability on the overall performance of MisoDICE. Specifically, we compare the results when using two different versions of OpenAI's models for the preference labeling task: the more powerful GPT-4o and its smaller, potentially more efficient counterpart, GPT-4o-mini [39]. The motivation is to assess whether the

Table 7: Ablation Study 1: Final average win rates for MisoDICE on SMACv1 and SMACv2 tasks, varying the number of top-k expert trajectories ($K_{\text{expert}}$) selected via LLM-based preference labeling.

| | | topk=50 | topk=200 | topk=400 | topk=800 | topk=1200 |
|---|---|---|---|---|---|---|
| | 2c_vs_64zg | 5.6 ± 4.9 | 13.0 ± 9.0 | 1.6 ± 0.7 | 1.7 ± 0.8 | 0.6 ± 0.7 |
| | 5m_vs_6m | 0.0 ± 0.0 | 1.2 ± 0.5 | 0.0 ± 0.0 | 0.0 ± 0.0 | 0.0 ± 0.0 |
| | 6h_vs_8z | 0.0 ± 0.0 | 1.1 ± 0.8 | 0.0 ± 0.0 | 0.0 ± 0.0 | 0.0 ± 0.0 |
| | corridor | 0.0 ± 0.0 | 1.4 ± 0.6 | 0.0 ± 0.0 | 0.0 ± 0.0 | 0.0 ± 0.0 |
| Protoss | 5_vs_5 | 8.9 ± 1.9 | 20.7 ± 0.9 | 10.1 ± 2.8 | 12.5 ± 1.7 | 9.4 ± 1.7 |
| | 10_vs_10 | 5.3 ± 1.8 | 14.1 ± 2.1 | 11.1 ± 3.3 | 10.0 ± 3.0 | 5.7 ± 1.6 |
| | 10_vs_11 | 0.3 ± 0.4 | 4.7 ± 0.3 | 2.2 ± 1.2 | 2.0 ± 1.0 | 1.6 ± 0.6 |
| | 20_vs_20 | 2.8 ± 0.8 | 11.0 ± 3.2 | 3.6 ± 1.4 | 6.1 ± 1.9 | 3.4 ± 0.9 |
| | 20_vs_23 | 0.6 ± 0.4 | 3.8 ± 2.0 | 2.2 ± 0.8 | 1.8 ± 1.2 | 1.0 ± 0.4 |
| Terran | 5_vs_5 | 8.5 ± 1.9 | 14.2 ± 3.1 | 9.8 ± 2.1 | 10.4 ± 2.1 | 5.1 ± 1.1 |
| | 10_vs_10 | 7.7 ± 0.6 | 12.0 ± 1.7 | 7.9 ± 2.1 | 8.1 ± 3.8 | 5.2 ± 0.3 |
| | 10_vs_11 | 0.8 ± 0.6 | 4.2 ± 1.6 | 2.2 ± 1.2 | 1.4 ± 0.9 | 0.6 ± 0.4 |
| | 20_vs_20 | 3.7 ± 0.4 | 8.8 ± 1.5 | 4.3 ± 0.6 | 6.0 ± 1.6 | 5.2 ± 1.2 |
| | 20_vs_23 | 0.5 ± 0.3 | 1.8 ± 1.4 | 0.6 ± 0.6 | 0.3 ± 0.4 | 0.1 ± 0.1 |
| Zerg | 5_vs_5 | 5.6 ± 1.3 | 7.9 ± 1.0 | 5.3 ± 1.1 | 4.7 ± 1.6 | 3.8 ± 0.6 |
| | 10_vs_10 | 3.4 ± 1.7 | 8.9 ± 2.1 | 5.6 ± 1.5 | 6.7 ± 1.1 | 3.5 ± 1.0 |
| | 10_vs_11 | 2.5 ± 1.0 | 6.8 ± 1.1 | 4.6 ± 1.6 | 4.5 ± 1.3 | 2.8 ± 0.6 |
| | 20_vs_20 | 0.3 ± 0.3 | 2.4 ± 0.6 | 0.3 ± 0.3 | 0.4 ± 0.5 | 0.3 ± 0.3 |
| | 20_vs_23 | 0.6 ± 0.3 | 2.7 ± 0.4 | 1.4 ± 0.7 | 1.1 ± 1.0 | 1.2 ± 0.7 |

Figure 10: Ablation Study 1: Box plots of final average win rates for MisoDICE on SMACv2 tasks, showing the impact of varying the number of top-k expert trajectories ($K_{\text{expert}}$) selected via LLM-based preference labeling. Results are presented for Protoss, Terran, and Zerg game races.

enhanced reasoning capabilities of a larger model like GPT-4o translate into significantly better preference data and, consequently, improved downstream imitation learning performance for MisoDICE, or if a more compact model like GPT-4o-mini can provide comparable results with potentially lower computational and financial costs.

We evaluate MisoDICE's final performance, in terms of returns and win rates, on our benchmark tasks when the initial preference labeling (Phase 1) is conducted by GPT-4o versus GPT-4o-mini. The comparative results, presented in Table 10, will shed light on the trade-offs associated with LLM selection for data annotation in the MisoDICE framework and provide insights into the level of LLM sophistication required for effective multi-agent imitation learning from unlabeled demonstrations.

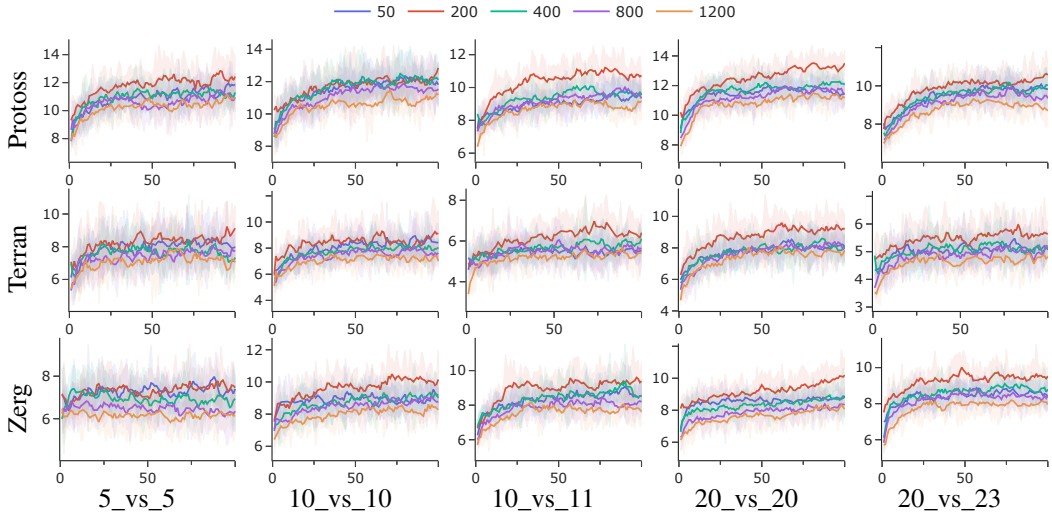

Figure 11: Ablation Study 1: Learning curves of average returns for MisoDICE on SMACv2 tasks, illustrating the impact of varying the number of top-k expert trajectories ($K_{expert}$) selected via LLM-based preference labeling. Each curve corresponds to a different $K_{expert}$ value from the set {50, 200, 400, 800, 1200}.

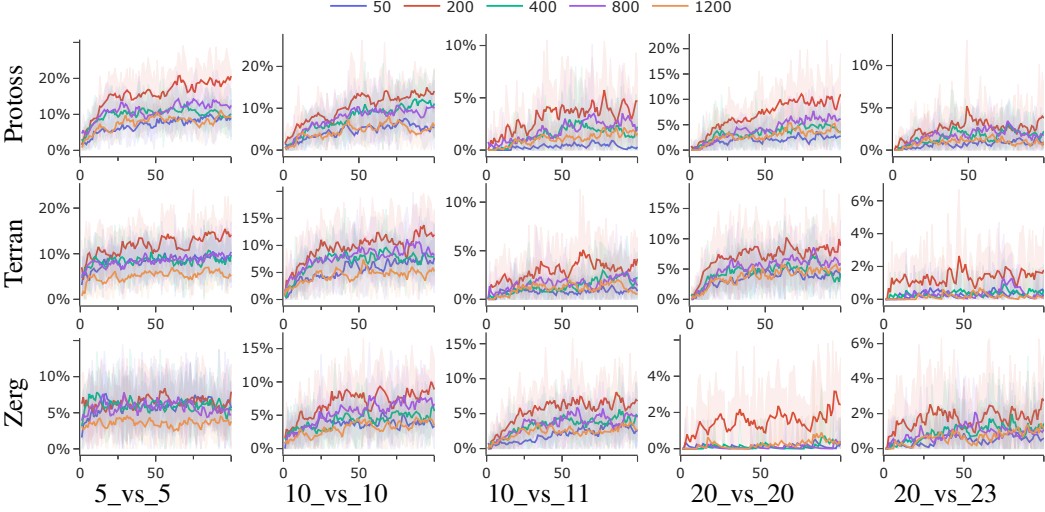

Figure 12: Ablation Study 1: Learning curves of average win rates for MisoDICE on SMACv2 tasks, illustrating the impact of varying the number of top-k expert trajectories ($K_{expert}$) selected via LLM-based preference labeling. Each curve corresponds to a different $K_{expert}$ value from the set {50, 200, 400, 800, 1200}.

## C.5 Ablation Study 4: Performance with Rule-Based Preference Labeling

While a key contribution of our work involves leveraging Large Language Models (LLMs) for inferring expert-like trajectories from unlabeled demonstrations, it is also important to understand MisoDICE's performance when relying on more traditional or simpler methods for generating initial preference data. This ablation study evaluates MisoDICE using a rule-based approach for preference labeling in its first phase.

In this rule-based setting, preference labels for pairs of trajectories $(\sigma_1, \sigma_2)$ are generated based on a predefined heuristic, typically by assuming that the cumulative return for each trajectory is known or

Table 8: Ablation Study 2: Final average returns for MisoDICE on SMACv1 and SMACv2 tasks, evaluating the impact of different values for the suboptimal data influence hyperparameter ($\alpha$) when using LLM-based preference labeling. The hyperparameter $\alpha$ controls the influence of the union dataset in the learning objective.

| | $\alpha = 0.00$ | $\alpha = 0.05$ | $\alpha = 0.10$ | $\alpha = 0.50$ | $\alpha = 1.00$ | $\alpha = 10.0$ |
|---|---|---|---|---|---|---|
| 2c_vs_64zg | $15.9 \pm 1.0$ | $16.4 \pm 1.3$ | $15.3 \pm 1.1$ | $14.9 \pm 0.7$ | $13.3 \pm 0.9$ | $10.9 \pm 0.3$ |
| 5m_vs_6m | $6.6 \pm 1.4$ | $7.3 \pm 0.1$ | $6.3 \pm 1.0$ | $6.5 \pm 0.5$ | $6.9 \pm 0.2$ | $6.7 \pm 0.1$ |
| 6h_vs_8z | $8.3 \pm 0.2$ | $8.7 \pm 0.2$ | $8.0 \pm 0.2$ | $8.0 \pm 0.2$ | $7.7 \pm 0.2$ | $7.6 \pm 0.1$ |
| corridor | $5.3 \pm 0.6$ | $5.8 \pm 0.8$ | $5.3 \pm 0.6$ | $5.1 \pm 0.2$ | $3.8 \pm 0.9$ | $1.7 \pm 0.1$ |
| Protoss 5_vs_5 | $12.4 \pm 0.4$ | $12.4 \pm 0.5$ | $11.7 \pm 0.4$ | $11.8 \pm 0.4$ | $11.9 \pm 0.3$ | $11.7 \pm 0.4$ |
| Protoss 10_vs_10 | $12.1 \pm 0.2$ | $12.9 \pm 0.2$ | $12.2 \pm 0.6$ | $12.1 \pm 0.3$ | $11.6 \pm 0.5$ | $12.1 \pm 0.4$ |
| Protoss 10_vs_11 | $10.7 \pm 0.5$ | $10.7 \pm 0.4$ | $10.1 \pm 0.5$ | $10.1 \pm 0.4$ | $9.8 \pm 0.3$ | $9.9 \pm 0.1$ |
| Protoss 20_vs_20 | $12.4 \pm 0.2$ | $13.5 \pm 0.5$ | $12.3 \pm 0.4$ | $12.2 \pm 0.5$ | $11.8 \pm 0.2$ | $12.1 \pm 0.4$ |
| Protoss 20_vs_23 | $10.4 \pm 0.4$ | $10.6 \pm 0.2$ | $10.1 \pm 0.3$ | $10.1 \pm 0.2$ | $10.0 \pm 0.2$ | $9.6 \pm 0.5$ |
| Terran 5_vs_5 | $8.8 \pm 0.7$ | $9.1 \pm 0.3$ | $8.4 \pm 0.8$ | $8.5 \pm 0.2$ | $7.9 \pm 0.4$ | $8.2 \pm 0.3$ |
| Terran 10_vs_10 | $8.9 \pm 0.8$ | $9.1 \pm 1.3$ | $8.5 \pm 0.6$ | $8.6 \pm 0.9$ | $7.8 \pm 0.7$ | $8.1 \pm 0.7$ |
| Terran 10_vs_11 | $5.9 \pm 0.4$ | $6.4 \pm 0.5$ | $5.8 \pm 0.3$ | $5.9 \pm 0.4$ | $5.7 \pm 0.5$ | $5.8 \pm 0.6$ |
| Terran 20_vs_20 | $9.0 \pm 0.6$ | $9.2 \pm 0.6$ | $8.5 \pm 0.7$ | $8.4 \pm 0.6$ | $8.2 \pm 0.6$ | $8.4 \pm 0.7$ |
| Terran 20_vs_23 | $5.6 \pm 0.2$ | $5.6 \pm 0.4$ | $5.3 \pm 0.3$ | $5.1 \pm 0.3$ | $5.1 \pm 0.3$ | $5.1 \pm 0.1$ |
| Zerg 5_vs_5 | $7.3 \pm 0.5$ | $7.5 \pm 0.1$ | $7.2 \pm 0.6$ | $6.8 \pm 0.2$ | $6.4 \pm 0.3$ | $6.4 \pm 0.3$ |
| Zerg 10_vs_10 | $9.7 \pm 0.4$ | $10.2 \pm 0.6$ | $9.2 \pm 0.3$ | $9.2 \pm 0.4$ | $8.7 \pm 0.2$ | $8.9 \pm 0.4$ |
| Zerg 10_vs_11 | $9.0 \pm 0.5$ | $9.4 \pm 0.3$ | $8.5 \pm 0.2$ | $8.8 \pm 0.3$ | $8.7 \pm 0.4$ | $8.3 \pm 0.5$ |
| Zerg 20_vs_20 | $9.0 \pm 0.4$ | $10.2 \pm 0.6$ | $9.0 \pm 0.3$ | $9.1 \pm 0.7$ | $8.8 \pm 0.4$ | $8.9 \pm 0.4$ |
| Zerg 20_vs_23 | $8.9 \pm 0.2$ | $9.5 \pm 0.2$ | $8.8 \pm 0.4$ | $8.9 \pm 0.3$ | $8.5 \pm 0.4$ | $8.6 \pm 0.4$ |

Table 9: Ablation Study 2: Final average win rates for MisoDICE on SMACv1 and SMACv2 tasks, evaluating the impact of different values for the suboptimal data influence hyperparameter ($\alpha$) when using LLM-based preference labeling. The hyperparameter $\alpha$ controls the influence of the union dataset in the learning objective.

| | $\alpha = 0.00$ | $\alpha = 0.05$ | $\alpha = 0.10$ | $\alpha = 0.50$ | $\alpha = 1.00$ | $\alpha = 10.0$ |
|---|---|---|---|---|---|---|
| 2c_vs_64zg | $11.0 \pm 5.8$ | $13.0 \pm 9.0$ | $10.0 \pm 5.0$ | $9.3 \pm 2.9$ | $7.1 \pm 4.1$ | $1.7 \pm 0.9$ |
| 5m_vs_6m | $1.1 \pm 1.0$ | $1.2 \pm 0.5$ | $0.6 \pm 0.4$ | $0.7 \pm 0.4$ | $0.9 \pm 0.7$ | $0.7 \pm 0.5$ |
| 6h_vs_8z | $1.0 \pm 0.8$ | $1.1 \pm 0.8$ | $1.1 \pm 0.8$ | $0.8 \pm 0.5$ | $0.4 \pm 0.3$ | $1.0 \pm 0.5$ |
| corridor | $0.4 \pm 0.2$ | $1.4 \pm 0.6$ | $1.2 \pm 0.5$ | $0.5 \pm 0.4$ | $1.4 \pm 0.7$ | $0.9 \pm 0.4$ |
| Protoss 5_vs_5 | $12.1 \pm 1.6$ | $20.7 \pm 0.9$ | $14.4 \pm 2.1$ | $12.9 \pm 2.0$ | $14.1 \pm 2.2$ | $17.2 \pm 2.5$ |
| Protoss 10_vs_10 | $8.4 \pm 3.3$ | $14.1 \pm 2.1$ | $9.9 \pm 2.2$ | $9.5 \pm 2.6$ | $9.0 \pm 1.7$ | $11.1 \pm 2.3$ |
| Protoss 10_vs_11 | $3.3 \pm 1.1$ | $4.7 \pm 0.3$ | $4.3 \pm 0.4$ | $3.5 \pm 1.3$ | $3.5 \pm 1.1$ | $4.3 \pm 1.2$ |
| Protoss 20_vs_20 | $5.4 \pm 0.4$ | $11.0 \pm 3.2$ | $5.5 \pm 1.1$ | $6.4 \pm 1.0$ | $6.4 \pm 0.7$ | $7.8 \pm 2.4$ |
| Protoss 20_vs_23 | $4.3 \pm 0.6$ | $3.8 \pm 2.0$ | $2.3 \pm 1.4$ | $3.5 \pm 1.6$ | $3.3 \pm 0.8$ | $2.5 \pm 1.3$ |
| Terran 5_vs_5 | $12.4 \pm 2.9$ | $14.2 \pm 3.1$ | $11.2 \pm 0.8$ | $12.1 \pm 1.6$ | $12.0 \pm 3.2$ | $12.2 \pm 3.2$ |
| Terran 10_vs_10 | $9.3 \pm 1.2$ | $12.0 \pm 1.7$ | $8.6 \pm 1.9$ | $9.2 \pm 3.6$ | $8.8 \pm 1.9$ | $9.3 \pm 3.2$ |
| Terran 10_vs_11 | $2.3 \pm 0.4$ | $4.2 \pm 1.6$ | $3.5 \pm 1.2$ | $2.3 \pm 0.7$ | $2.2 \pm 0.8$ | $3.6 \pm 1.5$ |
| Terran 20_vs_20 | $4.8 \pm 1.6$ | $8.8 \pm 1.5$ | $4.9 \pm 1.5$ | $5.9 \pm 2.5$ | $4.9 \pm 1.9$ | $9.3 \pm 2.1$ |
| Terran 20_vs_23 | $1.5 \pm 0.7$ | $1.8 \pm 1.4$ | $1.2 \pm 0.7$ | $1.8 \pm 0.7$ | $1.2 \pm 0.9$ | $1.7 \pm 1.0$ |
| Zerg 5_vs_5 | $7.1 \pm 1.6$ | $7.9 \pm 1.0$ | $7.4 \pm 1.0$ | $5.9 \pm 0.7$ | $6.4 \pm 2.7$ | $7.5 \pm 2.1$ |
| Zerg 10_vs_10 | $6.7 \pm 1.2$ | $8.9 \pm 2.1$ | $7.2 \pm 2.6$ | $7.0 \pm 0.8$ | $7.1 \pm 0.9$ | $6.9 \pm 1.0$ |
| Zerg 10_vs_11 | $5.0 \pm 1.4$ | $6.8 \pm 1.1$ | $5.0 \pm 1.5$ | $4.1 \pm 1.9$ | $5.0 \pm 1.2$ | $4.5 \pm 1.4$ |
| Zerg 20_vs_20 | $1.3 \pm 0.5$ | $2.4 \pm 0.6$ | $1.2 \pm 0.4$ | $1.7 \pm 2.1$ | $1.6 \pm 0.3$ | $2.6 \pm 0.6$ |
| Zerg 20_vs_23 | $1.4 \pm 0.5$ | $2.7 \pm 0.4$ | $1.0 \pm 0.4$ | $2.5 \pm 0.7$ | $2.7 \pm 1.7$ | $2.1 \pm 0.7$ |

can be accurately estimated, with the trajectory yielding a higher return being labeled as preferred. Although straightforward, such rule-based methods can have limitations in complex cooperative multi-agent settings, as individual rewards or simple cumulative returns may not always accurately reflect true team-level cooperativeness or overall strategic success.

The primary motivation for this study is twofold: first, to assess the inherent capability of the MisoDICE algorithm (Phase 2) to learn effective policies even when the expert dataset $\mathcal{D}^E_{rule}$ is identified using these simpler, potentially less nuanced, preference signals. Second, it provides a crucial benchmark to quantify the additional benefits and performance improvements gained by employing the more sophisticated LLM-based preference labeling approach presented as our main method.

We present the performance of MisoDICE and relevant baselines in terms of returns and win rates when using rule-based preference data, as detailed in Table 11 and Table 12. These results will illustrate MisoDICE's adaptability and provide a clearer understanding of the impact of different preference generation techniques on final policy quality.

### C.6 Ablation Study 5: Performance with Ground Truth Data Labels

In this ablation study, we aim to establish an upper-bound performance benchmark for our MisoDICE framework by utilizing ground truth labels for the initial data segmentation in Phase 1. This means that instead of relying on LLM-generated preferences or rule-based heuristics to identify expert trajectories, we directly use the original quality labels (i.e., 'expert' and 'poor') available from the O-MAPL datasets from which our mixed-quality dataset $\mathcal{D}^U$ was constructed.

Specifically, the expert dataset $\mathcal{D}^E$ is formed by selecting trajectories that were originally labeled as 'expert' in the O-MAPL dataset, and the suboptimal dataset $\mathcal{D}^{Mix}$ comprises those originally labeled as 'poor'. This process effectively bypasses the entirety of our proposed multi-step labeling pipeline (Section 4, Algorithm 2), including LLM prompting, preference learning with O-MAPL for reward recovery, and subsequent trajectory ranking. By providing MisoDICE's Phase 2 with this perfectly partitioned data, we can evaluate the core imitation learning algorithm's effectiveness under idealized conditions.

Table 10: Ablation Study 3: Comparison of MisoDICE performance (final average returns and win rates) on SMACv1 and SMACv2 tasks when using GPT-4o versus GPT-4o-mini for LLM-based preference labeling in the initial data annotation phase (Phase 1).

| | | Returns - MisoDICE | | Winrates - MisoDICE | |
| --- | --- | --- | --- | --- | --- |
| | | (gpt-4o-mini) | (gpt-4o) | (gpt-4o-mini) | (gpt-4o) |
| | 2c_vs_64zg | 12.2 ± 0.6 | 16.4 ± 1.3 | 2.1 ± 1.1 | 13.0 ± 9.0 |
| | 5m_vs_6m | 6.7 ± 0.9 | 7.3 ± 0.1 | 1.2 ± 0.4 | 1.2 ± 0.5 |
| | 6h_vs_8z | 8.1 ± 0.2 | 8.7 ± 0.2 | 0.1 ± 0.1 | 1.1 ± 0.8 |
| | corridor | 3.8 ± 0.8 | 5.8 ± 0.8 | 0.6 ± 0.7 | 1.4 ± 0.6 |
| Protoss | 5_vs_5 | 12.0 ± 0.3 | 12.4 ± 0.5 | 12.6 ± 2.6 | 20.7 ± 0.9 |
| | 10_vs_10 | 12.0 ± 0.5 | 12.9 ± 0.2 | 10.5 ± 3.5 | 14.1 ± 2.1 |
| | 10_vs_11 | 10.6 ± 0.3 | 10.7 ± 0.4 | 4.0 ± 1.8 | 4.7 ± 0.3 |
| | 20_vs_20 | 12.4 ± 0.5 | 13.5 ± 0.5 | 6.0 ± 0.9 | 11.0 ± 3.2 |
| | 20_vs_23 | 10.3 ± 0.1 | 10.6 ± 0.2 | 2.3 ± 1.4 | 3.8 ± 2.0 |
| Terran | 5_vs_5 | 8.1 ± 0.5 | 9.1 ± 0.3 | 10.0 ± 1.4 | 14.2 ± 3.1 |
| | 10_vs_10 | 8.6 ± 0.9 | 9.1 ± 1.3 | 9.2 ± 2.1 | 12.0 ± 1.7 |
| | 10_vs_11 | 6.0 ± 0.3 | 6.4 ± 0.5 | 2.2 ± 1.1 | 4.2 ± 1.6 |
| | 20_vs_20 | 8.1 ± 0.5 | 9.2 ± 0.6 | 6.1 ± 2.1 | 8.8 ± 1.5 |
| | 20_vs_23 | 5.3 ± 0.3 | 5.6 ± 0.4 | 0.9 ± 0.6 | 1.8 ± 1.4 |
| Zerg | 5_vs_5 | 7.0 ± 0.4 | 7.5 ± 0.1 | 5.3 ± 1.1 | 7.9 ± 1.0 |
| | 10_vs_10 | 9.6 ± 0.2 | 10.2 ± 0.6 | 7.1 ± 0.9 | 8.9 ± 2.1 |
| | 10_vs_11 | 9.2 ± 0.6 | 9.4 ± 0.3 | 5.0 ± 0.9 | 6.8 ± 1.1 |
| | 20_vs_20 | 8.9 ± 0.4 | 10.2 ± 0.6 | 0.8 ± 0.8 | 2.4 ± 0.6 |
| | 20_vs_23 | 9.0 ± 0.2 | 9.5 ± 0.2 | 1.7 ± 0.8 | 2.7 ± 0.4 |

The primary motivation for this study is to isolate and assess the performance of the MisoDICE multi-agent IL algorithm (Phase 2) when the complexities and potential inaccuracies of the trajectory labeling stage are removed. The results obtained from this setup will serve as a crucial reference point, allowing us to quantify how closely the performance achieved with our LLM-based or rule-based labeling approaches (as explored in previous ablation studies and main experiments) approximates this scenario with ground truth data separation. This will provide insights into the efficacy of the labeling

Table 11: Ablation Study 4: Final average returns for MisoDICE and baseline methods on MaMujoco, SMACv1, and SMACv2 tasks. For MisoDICE, the expert dataset ($\mathcal{D}^E_{\text{rule}}$) was identified using a rule-based preference labeling approach in Phase 1.

| | BC | | | INDD | VDN | MisoDICE |
| | ($\beta = 0.0$) | ($\beta = 0.5$) | ($\beta = 1.0$) | | | (ours) |
|---|---|---|---|---|---|---|
| Hopper-v2 | 123.0 ± 23.1 | 154.1 ± 44.6 | 158.1 ± 46.4 | 135.4 ± 32.8 | 171.4 ± 19.2 | 206.5 ± 22.5 |
| Ant-v2 | 1026.5 ± 77.7 | 1631.1 ± 51.8 | 1910.8 ± 70.2 | 1826.4 ± 16.6 | 1869.7 ± 9.1 | 2025.8 ± 3.2 |
| HalfCheetah-v2 | -201.5 ± 7.9 | -206.5 ± 6.0 | -73.4 ± 34.5 | -277.9 ± 5.7 | -243.9 ± 14.8 | -72.1 ± 31.8 |
| 2c_vs_64zg | 8.4 ± 0.2 | 9.3 ± 0.3 | 9.7 ± 0.7 | 11.8 ± 0.4 | 13.2 ± 1.5 | 15.0 ± 1.4 |
| 5m_vs_6m | 4.9 ± 1.3 | 6.3 ± 0.6 | 5.9 ± 0.4 | 6.6 ± 0.2 | 6.7 ± 0.1 | 7.2 ± 0.1 |
| 6h_vs_8z | 7.0 ± 0.1 | 7.4 ± 0.1 | 7.1 ± 0.2 | 7.4 ± 0.3 | 7.8 ± 0.1 | 8.5 ± 0.1 |
| corridor | 1.4 ± 0.1 | 1.5 ± 0.2 | 3.0 ± 0.6 | 3.0 ± 0.6 | 1.7 ± 0.0 | 4.7 ± 0.6 |
| Protoss 5_vs_5 | 9.8 ± 0.4 | 10.4 ± 0.4 | 10.2 ± 0.6 | 10.4 ± 0.5 | 11.5 ± 0.6 | 12.2 ± 0.7 |
| Protoss 10_vs_10 | 9.4 ± 0.2 | 11.7 ± 0.8 | 10.1 ± 0.3 | 10.9 ± 0.5 | 11.8 ± 0.6 | 12.6 ± 0.6 |
| Protoss 10_vs_11 | 7.7 ± 0.4 | 9.4 ± 0.3 | 8.4 ± 0.5 | 9.1 ± 0.4 | 9.5 ± 0.4 | 10.6 ± 0.4 |
| Protoss 20_vs_20 | 10.5 ± 0.3 | 10.3 ± 0.4 | 9.9 ± 0.6 | 11.3 ± 0.3 | 11.4 ± 0.3 | 13.1 ± 0.4 |
| Protoss 20_vs_23 | 7.9 ± 0.3 | 8.3 ± 0.3 | 8.3 ± 0.2 | 9.4 ± 0.2 | 9.5 ± 0.4 | 10.4 ± 0.4 |
| Terran 5_vs_5 | 6.5 ± 0.5 | 7.2 ± 0.7 | 6.9 ± 0.6 | 7.6 ± 0.4 | 8.0 ± 0.7 | 8.8 ± 0.5 |
| Terran 10_vs_10 | 6.1 ± 0.5 | 7.1 ± 0.7 | 6.6 ± 0.6 | 7.3 ± 0.7 | 7.6 ± 0.3 | 8.6 ± 0.3 |
| Terran 10_vs_11 | 4.6 ± 0.2 | 5.4 ± 0.6 | 5.0 ± 0.2 | 5.5 ± 0.1 | 5.7 ± 0.2 | 6.2 ± 0.5 |
| Terran 20_vs_20 | 6.5 ± 0.5 | 7.3 ± 0.5 | 6.7 ± 0.5 | 7.8 ± 0.6 | 8.4 ± 0.5 | 9.1 ± 0.4 |
| Terran 20_vs_23 | 4.0 ± 0.3 | 5.0 ± 0.2 | 4.2 ± 0.3 | 5.1 ± 0.5 | 5.0 ± 0.1 | 5.5 ± 0.4 |
| Zerg 5_vs_5 | 5.5 ± 0.3 | 6.5 ± 0.3 | 5.6 ± 0.2 | 6.2 ± 0.5 | 6.6 ± 0.3 | 7.4 ± 0.5 |
| Zerg 10_vs_10 | 7.1 ± 0.2 | 8.2 ± 0.5 | 7.2 ± 0.5 | 8.0 ± 0.3 | 8.5 ± 0.8 | 10.0 ± 0.2 |
| Zerg 10_vs_11 | 6.5 ± 0.4 | 8.2 ± 0.3 | 6.8 ± 0.1 | 7.9 ± 0.3 | 8.4 ± 0.2 | 9.1 ± 0.6 |
| Zerg 20_vs_20 | 7.3 ± 0.2 | 9.0 ± 0.3 | 7.6 ± 0.2 | 8.1 ± 0.7 | 8.6 ± 0.4 | 10.0 ± 0.6 |
| Zerg 20_vs_23 | 7.0 ± 0.2 | 7.9 ± 0.3 | 6.9 ± 0.1 | 8.1 ± 0.2 | 8.8 ± 0.3 | 9.5 ± 0.4 |

Table 12: Ablation Study 4: Final average win rates for MisoDICE and baseline methods on SMACv1 and SMACv2 tasks. For MisoDICE, the expert dataset ($\mathcal{D}^E_{\text{rule}}$) was identified using a rule-based preference labeling approach in Phase 1.

| | BC | | | INDD | VDN | MisoDICE |
| | ($\beta = 0.0$) | ($\beta = 0.5$) | ($\beta = 1.0$) | | | (ours) |
|---|---|---|---|---|---|---|
| 2c_vs_64zg | 0.1 ± 0.1 | 0.5 ± 0.3 | 1.0 ± 1.0 | 2.3 ± 0.1 | 7.1 ± 4.6 | 8.4 ± 5.9 |
| 5m_vs_6m | 0.2 ± 0.4 | 0.7 ± 0.5 | 0.1 ± 0.1 | 0.1 ± 0.1 | 1.1 ± 0.8 | 1.3 ± 0.5 |
| 6h_vs_8z | 0.2 ± 0.2 | 0.0 ± 0.0 | 0.2 ± 0.3 | 0.1 ± 0.1 | 1.0 ± 0.6 | 1.1 ± 0.8 |
| corridor | 0.1 ± 0.1 | 0.6 ± 0.7 | 0.3 ± 0.4 | 0.1 ± 0.1 | 0.9 ± 0.6 | 1.4 ± 0.6 |
| Protoss 5_vs_5 | 15.8 ± 2.0 | 11.8 ± 1.7 | 13.7 ± 3.4 | 10.1 ± 1.6 | 14.3 ± 3.4 | 18.4 ± 1.3 |
| Protoss 10_vs_10 | 7.4 ± 3.3 | 9.4 ± 3.2 | 8.7 ± 2.9 | 7.2 ± 1.6 | 9.8 ± 2.1 | 12.2 ± 1.6 |
| Protoss 10_vs_11 | 1.4 ± 1.0 | 2.9 ± 1.5 | 1.7 ± 0.2 | 1.7 ± 0.2 | 2.8 ± 0.7 | 4.1 ± 1.1 |
| Protoss 20_vs_20 | 7.6 ± 1.7 | 2.9 ± 0.7 | 5.0 ± 2.4 | 3.9 ± 0.9 | 3.6 ± 1.4 | 8.7 ± 1.8 |
| Protoss 20_vs_23 | 1.4 ± 0.3 | 0.7 ± 0.5 | 1.7 ± 1.2 | 2.0 ± 0.6 | 1.6 ± 0.8 | 3.2 ± 0.8 |
| Terran 5_vs_5 | 8.7 ± 2.7 | 10.3 ± 2.6 | 10.4 ± 2.3 | 9.3 ± 1.6 | 8.8 ± 1.8 | 14.0 ± 3.5 |
| Terran 10_vs_10 | 7.0 ± 2.0 | 7.8 ± 3.1 | 7.8 ± 2.8 | 6.6 ± 1.0 | 5.9 ± 2.3 | 11.3 ± 1.8 |
| Terran 10_vs_11 | 1.5 ± 0.5 | 2.3 ± 0.8 | 2.0 ± 0.4 | 1.2 ± 0.8 | 2.2 ± 1.1 | 2.7 ± 1.2 |
| Terran 20_vs_20 | 3.5 ± 1.7 | 5.6 ± 0.7 | 3.5 ± 2.8 | 4.3 ± 1.2 | 5.5 ± 1.9 | 7.6 ± 0.7 |
| Terran 20_vs_23 | 0.5 ± 0.6 | 0.9 ± 0.3 | 0.5 ± 0.8 | 0.6 ± 0.6 | 0.9 ± 0.6 | 1.7 ± 1.3 |
| Zerg 5_vs_5 | 5.7 ± 0.4 | 5.2 ± 0.7 | 5.5 ± 1.4 | 5.4 ± 1.2 | 6.0 ± 2.1 | 6.6 ± 1.0 |
| Zerg 10_vs_10 | 3.7 ± 1.2 | 3.7 ± 0.9 | 3.7 ± 1.4 | 4.5 ± 1.1 | 4.3 ± 2.0 | 8.8 ± 1.7 |
| Zerg 10_vs_11 | 3.3 ± 1.0 | 4.1 ± 1.8 | 3.2 ± 1.4 | 3.6 ± 0.5 | 5.0 ± 1.4 | 6.7 ± 2.6 |
| Zerg 20_vs_20 | 0.3 ± 0.1 | 0.9 ± 0.5 | 0.9 ± 0.3 | 0.1 ± 0.1 | 1.1 ± 0.4 | 1.7 ± 0.4 |
| Zerg 20_vs_23 | 0.7 ± 0.4 | 0.6 ± 0.4 | 0.9 ± 0.4 | 0.6 ± 0.6 | 1.6 ± 0.4 | 2.4 ± 0.2 |

procedures themselves and the overall robustness of MisoDICE. We will present the performance of MisoDICE in terms of returns and win rates in Table 13 and 14 under these ground truth conditions.

Table 13: Ablation Study 5: Final average returns on SMACv1 & SMACv2 tasks for MisoDICE and baselines, using ground truth data labels for expert/suboptimal trajectory separation.

| | | BC | | INDD | VDN | MisoDICE |
| | ($\beta = 0.0$) | ($\beta = 0.5$) | ($\beta = 1.0$) | | | (ours) |
|---|---|---|---|---|---|---|
| 2c_vs_64zg | 11.7 ± 0.4 | 11.3 ± 0.1 | 13.1 ± 0.8 | 15.1 ± 1.3 | 15.9 ± 2.0 | 16.1 ± 1.8 |
| 5m_vs_6m | 6.1 ± 1.4 | 6.8 ± 1.4 | 7.2 ± 0.2 | 7.4 ± 0.2 | 7.4 ± 0.3 | 7.4 ± 0.1 |
| 6h_vs_8z | 8.4 ± 0.1 | 8.4 ± 0.2 | 8.3 ± 0.3 | 8.2 ± 0.3 | 8.5 ± 0.1 | 8.9 ± 0.1 |
| corridor | 2.1 ± 0.3 | 1.9 ± 0.3 | 5.7 ± 1.5 | 2.0 ± 0.2 | 5.8 ± 1.5 | 6.1 ± 1.6 |
| Protoss 5_vs_5 | 13.3 ± 0.1 | 12.6 ± 2.8 | 12.0 ± 1.2 | 11.9 ± 1.9 | 13.3 ± 0.5 | 13.7 ± 1.2 |
| Protoss 10_vs_10 | 13.1 ± 0.9 | 12.5 ± 1.4 | 12.8 ± 0.9 | 12.3 ± 0.8 | 13.2 ± 0.5 | 14.0 ± 1.2 |
| Protoss 10_vs_11 | 10.8 ± 1.3 | 10.6 ± 0.9 | 10.9 ± 0.8 | 10.1 ± 1.1 | 11.4 ± 0.5 | 12.0 ± 1.3 |
| Protoss 20_vs_20 | 12.4 ± 1.4 | 12.2 ± 0.5 | 13.0 ± 0.4 | 12.7 ± 0.6 | 13.3 ± 1.4 | 13.9 ± 1.6 |
| Protoss 20_vs_23 | 10.1 ± 1.5 | 9.5 ± 0.5 | 10.2 ± 0.8 | 10.1 ± 0.9 | 10.4 ± 0.8 | 10.5 ± 1.1 |
| Terran 5_vs_5 | 8.9 ± 0.9 | 8.4 ± 1.0 | 8.7 ± 0.9 | 8.9 ± 1.8 | 9.4 ± 2.3 | 9.4 ± 1.5 |
| Terran 10_vs_10 | 7.5 ± 1.0 | 8.2 ± 1.0 | 7.7 ± 1.2 | 7.9 ± 0.7 | 8.9 ± 1.3 | 9.0 ± 0.6 |
| Terran 10_vs_11 | 6.3 ± 0.4 | 5.9 ± 1.4 | 5.8 ± 0.4 | 5.9 ± 0.7 | 7.1 ± 1.2 | 7.3 ± 1.0 |
| Terran 20_vs_20 | 8.6 ± 1.2 | 8.3 ± 0.8 | 7.9 ± 0.2 | 8.3 ± 1.0 | 8.7 ± 1.0 | 9.3 ± 1.2 |
| Terran 20_vs_23 | 5.1 ± 0.2 | 5.5 ± 0.5 | 5.4 ± 0.3 | 4.8 ± 0.4 | 5.6 ± 0.5 | 5.9 ± 0.7 |
| Zerg 5_vs_5 | 7.8 ± 1.2 | 7.0 ± 0.9 | 7.1 ± 1.4 | 6.8 ± 0.3 | 7.8 ± 0.8 | 8.0 ± 0.6 |
| Zerg 10_vs_10 | 8.7 ± 1.2 | 8.2 ± 0.5 | 9.2 ± 0.8 | 9.6 ± 1.2 | 9.6 ± 1.7 | 9.9 ± 0.5 |
| Zerg 10_vs_11 | 9.6 ± 0.4 | 9.3 ± 1.5 | 8.8 ± 1.2 | 9.3 ± 1.0 | 9.8 ± 0.9 | 9.9 ± 0.9 |
| Zerg 20_vs_20 | 9.2 ± 0.8 | 9.3 ± 0.6 | 9.4 ± 1.1 | 9.2 ± 0.7 | 9.5 ± 0.8 | 11.1 ± 1.3 |
| Zerg 20_vs_23 | 9.2 ± 1.1 | 8.9 ± 0.5 | 8.2 ± 0.6 | 9.3 ± 1.0 | 9.5 ± 0.3 | 9.5 ± 0.1 |

Table 14: Ablation Study 5: Final average win rates on SMACv1 & SMACv2 tasks for MisoDICE and baselines, using ground truth data labels for expert/suboptimal trajectory separation.

| | | BC | | INDD | VDN | MisoDICE |
| | ($\beta = 0.0$) | ($\beta = 0.5$) | ($\beta = 1.0$) | | | (ours) |
|---|---|---|---|---|---|---|
| 2c_vs_64zg | 1.6 ± 1.6 | 0.0 ± 0.0 | 5.5 ± 2.6 | 9.4 ± 2.2 | 10.2 ± 7.1 | 11.7 ± 8.1 |
| 5m_vs_6m | 0.8 ± 1.4 | 1.6 ± 1.6 | 0.8 ± 1.4 | 0.8 ± 1.4 | 1.6 ± 1.6 | 1.6 ± 1.6 |
| 6h_vs_8z | 0.0 ± 0.0 | 0.8 ± 1.4 | 0.0 ± 0.0 | 0.0 ± 0.0 | 1.6 ± 1.6 | 2.3 ± 2.6 |
| corridor | 0.8 ± 1.4 | 0.8 ± 1.4 | 0.8 ± 1.4 | 0.8 ± 1.4 | 1.6 ± 2.7 | 2.3 ± 2.6 |
| Protoss 5_vs_5 | 13.3 ± 3.4 | 17.2 ± 8.4 | 20.3 ± 4.7 | 17.2 ± 3.5 | 20.3 ± 6.8 | 21.1 ± 11.6 |
| Protoss 10_vs_10 | 10.2 ± 3.4 | 10.2 ± 3.4 | 8.6 ± 6.0 | 6.2 ± 2.2 | 10.2 ± 2.6 | 15.6 ± 4.4 |
| Protoss 10_vs_11 | 1.6 ± 1.6 | 3.9 ± 1.4 | 3.1 ± 3.1 | 4.7 ± 3.5 | 4.7 ± 1.6 | 6.2 ± 2.2 |
| Protoss 20_vs_20 | 8.6 ± 6.4 | 2.3 ± 2.6 | 9.4 ± 2.2 | 7.0 ± 7.8 | 9.4 ± 2.2 | 10.9 ± 5.2 |
| Protoss 20_vs_23 | 2.3 ± 2.6 | 1.6 ± 1.6 | 1.6 ± 1.6 | 2.3 ± 1.4 | 3.1 ± 3.8 | 3.9 ± 5.1 |
| Terran 5_vs_5 | 10.9 ± 3.5 | 7.0 ± 4.1 | 9.4 ± 3.8 | 11.7 ± 4.1 | 16.4 ± 7.5 | 17.2 ± 8.4 |
| Terran 10_vs_10 | 8.6 ± 2.6 | 7.0 ± 4.1 | 7.8 ± 3.5 | 5.5 ± 2.6 | 9.4 ± 3.8 | 10.9 ± 3.5 |
| Terran 10_vs_11 | 1.6 ± 1.6 | 1.6 ± 2.7 | 2.3 ± 1.4 | 1.6 ± 2.7 | 4.7 ± 3.5 | 5.5 ± 2.6 |
| Terran 20_vs_20 | 7.0 ± 2.6 | 6.2 ± 2.2 | 4.7 ± 3.5 | 4.7 ± 1.6 | 8.6 ± 5.1 | 9.4 ± 2.2 |
| Terran 20_vs_23 | 0.8 ± 1.4 | 0.0 ± 0.0 | 1.6 ± 2.7 | 0.8 ± 1.4 | 1.6 ± 1.6 | 2.3 ± 1.4 |
| Zerg 5_vs_5 | 6.2 ± 2.2 | 5.5 ± 1.4 | 3.9 ± 2.6 | 6.2 ± 3.8 | 7.0 ± 4.6 | 7.8 ± 4.7 |
| Zerg 10_vs_10 | 7.8 ± 2.7 | 5.5 ± 3.4 | 5.5 ± 4.6 | 3.1 ± 2.2 | 8.6 ± 6.8 | 8.6 ± 8.1 |
| Zerg 10_vs_11 | 3.9 ± 1.4 | 4.7 ± 3.5 | 0.8 ± 1.4 | 3.9 ± 2.6 | 8.6 ± 3.4 | 8.6 ± 4.6 |
| Zerg 20_vs_20 | 1.6 ± 2.7 | 1.6 ± 1.6 | 0.8 ± 1.4 | 0.0 ± 0.0 | 3.9 ± 2.6 | 2.3 ± 2.6 |
| Zerg 20_vs_23 | 0.8 ± 1.4 | 1.6 ± 1.6 | 1.6 ± 1.6 | 0.0 ± 0.0 | 1.6 ± 2.7 | 1.6 ± 2.7 |

## C.7 Ablation Study 6: Sensitivity to Noisy Preference Labels

We conduct an ablation study to analyze the framework's sensitivity to the quality of the preference feedback. This study investigates the impact of explicitly noisy preference labels in Phase 1 on the final performance of MisoDICE in Phase 2.

To simulate this, we first obtained the complete set of preference labels from GPT-4o. Then, we introduced synthetic noise by randomly "flipping" the preference outcomes for a certain percentage

of the trajectory pairs. We tested this at noise levels of 5%, 10%, 20%, and 30% of the samples. The performance of MisoDICE trained on these progressively noisier preference datasets is reported in Table 15.

The results clearly show a gradual degradation in MisoDICE's performance as the level of noise increases from 0% (no noise) to 30%. For example, in the '2c_vs_64zg' scenario, performance drops from $16.4 \pm 1.3$ to $11.5 \pm 0.5$. This trend is consistent across most scenarios, supporting the hypothesis that high-quality, consistent preference feedback in Phase 1 is critical for effective trajectory labeling and optimal downstream policy learning in Phase 2.

Table 15: Ablation Study 6: Sensitivity to Noisy Preference Labels. We report the final average returns for MisoDICE after introducing noise (randomly flipping labels) at different rates (5%, 10%, 20%, 30%) to the preference data generated by GPT-4o in Phase 1.

| | Scenario | MisoDICE (0% Noise) | MisoDICE (5% Noise) | MisoDICE (10% Noise) | MisoDICE (20% Noise) | MisoDICE (30% Noise) |
|---|---|---|---|---|---|---|
| | 2c_vs_64zg | $16.4 \pm 1.3$ | $14.7 \pm 1.0$ | $13.0 \pm 0.6$ | $11.6 \pm 0.7$ | $11.5 \pm 0.5$ |
| | 5m_vs_6m | $7.3 \pm 0.1$ | $6.5 \pm 1.0$ | $6.1 \pm 1.9$ | $6.2 \pm 1.3$ | $6.0 \pm 1.7$ |
| | 6h_vs_8z | $8.7 \pm 0.2$ | $8.2 \pm 0.1$ | $8.0 \pm 0.1$ | $8.1 \pm 0.1$ | $7.8 \pm 0.2$ |
| | corridor | $5.8 \pm 0.8$ | $2.9 \pm 0.6$ | $2.6 \pm 0.5$ | $2.0 \pm 0.3$ | $2.2 \pm 0.6$ |
| Protoss | 5_vs_5 | $12.4 \pm 0.5$ | $11.8 \pm 1.9$ | $11.5 \pm 0.8$ | $10.9 \pm 0.1$ | $10.0 \pm 0.5$ |
| | 10_vs_10 | $12.9 \pm 0.2$ | $11.9 \pm 0.9$ | $12.5 \pm 1.8$ | $11.6 \pm 0.3$ | $11.2 \pm 1.2$ |
| | 10_vs_11 | $10.7 \pm 0.4$ | $10.4 \pm 0.4$ | $10.3 \pm 0.8$ | $10.2 \pm 0.6$ | $9.7 \pm 0.7$ |
| | 20_vs_20 | $13.5 \pm 0.5$ | $12.9 \pm 0.8$ | $12.6 \pm 1.0$ | $12.4 \pm 1.0$ | $12.2 \pm 1.3$ |
| | 20_vs_23 | $10.6 \pm 0.2$ | $10.1 \pm 0.7$ | $9.9 \pm 1.1$ | $9.2 \pm 0.9$ | $9.4 \pm 1.0$ |
| Terran | 5_vs_5 | $9.1 \pm 0.3$ | $8.4 \pm 1.6$ | $6.8 \pm 0.5$ | $9.0 \pm 1.8$ | $8.1 \pm 1.4$ |
| | 10_vs_10 | $9.1 \pm 1.3$ | $8.3 \pm 1.9$ | $7.6 \pm 0.8$ | $7.1 \pm 0.8$ | $7.2 \pm 1.1$ |
| | 10_vs_11 | $6.4 \pm 0.5$ | $6.6 \pm 0.6$ | $6.2 \pm 0.9$ | $6.0 \pm 0.7$ | $6.0 \pm 1.2$ |
| | 20_vs_20 | $9.2 \pm 0.6$ | $8.5 \pm 0.7$ | $8.4 \pm 0.7$ | $8.1 \pm 1.1$ | $7.8 \pm 0.9$ |
| | 20_vs_23 | $5.6 \pm 0.4$ | $5.5 \pm 0.4$ | $5.2 \pm 0.6$ | $5.0 \pm 0.5$ | $4.9 \pm 0.5$ |
| Zerg | 5_vs_5 | $7.5 \pm 0.1$ | $7.4 \pm 0.9$ | $7.2 \pm 1.2$ | $6.3 \pm 0.4$ | $6.5 \pm 1.6$ |
| | 10_vs_10 | $10.2 \pm 0.6$ | $9.3 \pm 0.5$ | $8.2 \pm 0.5$ | $8.7 \pm 1.3$ | $8.0 \pm 1.4$ |
| | 10_vs_11 | $9.4 \pm 0.3$ | $9.5 \pm 1.3$ | $8.8 \pm 0.3$ | $8.4 \pm 0.8$ | $7.5 \pm 0.5$ |
| | 20_vs_20 | $10.2 \pm 0.6$ | $9.5 \pm 0.3$ | $9.2 \pm 0.7$ | $9.0 \pm 0.6$ | $8.5 \pm 0.9$ |
| | 20_vs_23 | $9.5 \pm 0.2$ | $9.4 \pm 0.9$ | $8.9 \pm 0.7$ | $8.5 \pm 0.3$ | $8.1 \pm 1.1$ |

## C.8   Ablation Study 7: Alignment of Phase 1 Labeling with Ground Truth

We conduct an experiment to measure the alignment of our Phase 1 labeling pipeline with ground-truth preferences. This study compares the preferences generated by our LLM-guided O-MAPL process against ground-truth preferences derived from environment rewards.

For this experiment, we sampled 2,000 trajectory pairs per task. The "learned" preferences were derived from the reward function recovered by O-MAPL (which was trained on LLM-generated feedback). The "ground-truth" preferences were determined by comparing the cumulative environment rewards for the two trajectories in each pair. We then calculated the accuracy as the percentage of pairs where the learned preference (expert vs. suboptimal) matched the ground-truth preference.

The results are reported in Table 16. The pipeline achieved a high overall accuracy (90.5% average across all tasks), demonstrating the strong reliability of the LLM feedback and O-MAPL refinement process in Phase 1. This highlights the effectiveness of our approach in filtering expert-like trajectories that closely align with the ground-truth reward-based preferences.

## C.9   Ablation Study 8: Importance of Phase 1 Trajectory Labeling

We assess the importance of the entire Phase 1 labeling pipeline by running an ablation study where this phase is skipped entirely. In this "no labeling" variant, all trajectories in the unlabeled union dataset $\mathcal{D}^U$ are treated as optimal demonstrations for the Phase 2 imitation learning algorithm (MisoDICE).

We compare this variant, "MisoDICE (no labeling)," against our full MisoDICE framework (which includes Phase 1 labeling) and the standard "BC (no labeling)" baseline (which corresponds to BC with $\beta = 0.0$ from our main experiments, i.e., standard BC on the entire mixed dataset).

The results in Table 17 show that the "MisoDICE (no labeling)" variant performs significantly worse than the full MisoDICE framework across all tasks. While it still slightly outperforms the simple BC baseline in most cases, this result strongly highlights the importance of Phase 1's labeling strategy. Simply applying the Phase 2 algorithm to the entire mixed-quality dataset as if it were all expert data is suboptimal. The labeling pipeline is crucial for distinguishing high-quality demonstrations, which in turn enables the Phase 2 algorithm to learn effectively from noisy, unlabeled data.

Table 16: Ablation Study 7: Accuracy of Phase 1 Preference Labeling vs. Ground Truth. This table shows the accuracy of the O-MAPL-recovered reward function (trained on LLM feedback) in correctly ranking trajectory pairs compared to ground-truth rankings based on environment rewards.

| Scenario | Accuracy | Scenario | Accuracy |
|---|---|---|---|
| *SMACv1* | | *SMACv2 - Terran* | |
| 2c_vs_64zg | 79.2% | terran_5_vs_5 | 90.4% |
| 5m_vs_6m | 87.0% | terran_10_vs_10 | 81.5% |
| 6h_vs_8z | 89.9% | terran_10_vs_11 | 78.3% |
| corridor | 87.7% | terran_20_vs_20 | 90.4% |
| | | terran_20_vs_23 | 96.8% |
| *SMACv2 - Protoss* | | *SMACv2 - Zerg* | |
| protoss_5_vs_5 | 89.2% | zerg_5_vs_5 | 98.8% |
| protoss_10_vs_10 | 91.1% | zerg_10_vs_10 | 85.3% |
| protoss_10_vs_11 | 90.5% | zerg_10_vs_11 | 95.6% |
| protoss_20_vs_20 | 92.0% | zerg_20_vs_20 | 92.4% |
| protoss_20_vs_23 | 93.8% | zerg_20_vs_23 | 97.7% |

Table 17: Ablation Study 8: Importance of Phase 1 Labeling. We compare the final average returns of the full MisoDICE (with Phase 1 labeling) against a variant where Phase 1 is skipped ("MisoDICE (no labeling)") and all data is treated as optimal. "BC (no labeling)" is the baseline BC trained on all data.

| Scenario | BC (no labeling) | MisoDICE (with Phase 1) | MisoDICE (no labeling) |
|---|---|---|---|
| 2c_vs_64zg | $8.5 \pm 0.1$ | $16.4 \pm 1.3$ | $10.9 \pm 0.3$ |
| 5m_vs_6m | $5.0 \pm 1.1$ | $7.3 \pm 0.1$ | $5.2 \pm 0.1$ |
| 6h_vs_8z | $7.0 \pm 0.0$ | $8.7 \pm 0.2$ | $7.2 \pm 0.0$ |
| corridor | $1.5 \pm 0.1$ | $5.8 \pm 0.8$ | $1.8 \pm 0.2$ |
| Protoss 5_vs_5 | $9.2 \pm 0.1$ | $12.4 \pm 0.5$ | $9.9 \pm 1.9$ |
| Protoss 10_vs_10 | $10.3 \pm 0.6$ | $12.9 \pm 0.2$ | $10.5 \pm 0.7$ |
| Protoss 10_vs_11 | $8.2 \pm 0.4$ | $10.7 \pm 0.4$ | $9.3 \pm 0.3$ |
| Protoss 20_vs_20 | $10.1 \pm 0.2$ | $13.5 \pm 0.5$ | $11.7 \pm 0.9$ |
| Protoss 20_vs_23 | $8.1 \pm 0.2$ | $10.6 \pm 0.2$ | $8.5 \pm 0.5$ |
| Terran 5_vs_5 | $6.5 \pm 0.8$ | $9.1 \pm 0.3$ | $6.8 \pm 0.8$ |
| Terran 10_vs_10 | $6.6 \pm 0.3$ | $9.1 \pm 1.3$ | $5.5 \pm 1.3$ |
| Terran 10_vs_11 | $4.7 \pm 0.2$ | $6.4 \pm 0.5$ | $4.2 \pm 0.6$ |
| Terran 20_vs_20 | $6.9 \pm 0.4$ | $9.2 \pm 0.6$ | $7.0 \pm 0.4$ |
| Terran 20_vs_23 | $4.0 \pm 0.3$ | $5.6 \pm 0.4$ | $4.5 \pm 0.7$ |
| Zerg 5_vs_5 | $5.7 \pm 0.5$ | $7.5 \pm 0.1$ | $5.7 \pm 1.1$ |
| Zerg 10_vs_10 | $7.3 \pm 0.1$ | $10.2 \pm 0.6$ | $7.6 \pm 0.8$ |
| Zerg 10_vs_11 | $7.3 \pm 0.2$ | $9.4 \pm 0.3$ | $7.7 \pm 0.5$ |
| Zerg 20_vs_20 | $7.4 \pm 0.6$ | $10.2 \pm 0.6$ | $7.8 \pm 1.0$ |
| Zerg 20_vs_23 | $7.1 \pm 0.3$ | $9.5 \pm 0.2$ | $7.3 \pm 0.5$ |

