# OpenReview forum: "MisoDICE: Multi-Agent Imitation from Mixed-Quality Demonstrations"
_NeurIPS.cc/2025/Conference — NeurIPS 2025 poster_

### Official Review · Reviewer_YWXt · 2025-07-01

**Clarity:** 2
**Significance:** 3
**Originality:** 3
**Rating:** 4
**Confidence:** 2

**Summary:**

This paper addresses the problem of imitation learning in cooperative multi-agent settings from a dataset of unlabeled, mixed-quality trajectories. The proposed method, MisoDICE, is a two-stage framework. The first stage employs a pipeline combining Large Language Models (LLMs) and preference-based reinforcement learning to identify and label expert-quality trajectories from the mixed data. The second stage introduces MisoDICE, a novel multi-agent imitation learning algorithm that leverages these labels. By extending the single-agent DICE framework with a new value decomposition architecture, the method derives a convex optimization objective and theoretically ensures consistency between the global joint policy and local agent policies. The effectiveness of the proposed method is evaluated on standard benchmarks such as the StarCraft Multi-Agent Challenge (SMAC).

**Questions:**

-Clarification on BC Performance: In Figure 1, the performance of Behavioral Cloning (BC) appears to increase over the course of training. Since BC is typically a supervised learning method whose performance on a fixed offline dataset should not change with training epochs, could you clarify what "training" entails for the BC baseline and why its performance curve shows an upward trend?

-Justification of the Two-Stage Framework: The paper's main theoretical contribution appears to be the MisoDICE algorithm (Stage 2), which is elegant and well-motivated. However, it is tightly coupled with the LLM-based labeling pipeline (Stage 1), which seems less general and brittle, as evidenced by its failure on MAMuJoCo. Could the authors elaborate on the necessity of this tight coupling? Would the paper's contribution be clearer and more robust if Stage 2 (MisoDICE) were presented as the core method, with the LLM-based approach being just one possible (and domain-dependent) method for generating the required labeled data?

- Positioning of Imitation Learning: The paper begins by stating, "Imitation Learning (IL) is a key subfield of Reinforcement Learning (RL)." This is a strong claim that may not be universally accepted, as many researchers treat IL and RL as related but distinct paradigms. It would be beneficial to elaborate on the positioning of IL relative to RL. A more nuanced discussion of their relationship and how this work fuses concepts from both would strengthen the paper's framing, especially considering existing studies that describe the theoretical fusion of RL and IL (e.g., [Kinose+ 2020] in robot learning).

Kinose, A., & Taniguchi, T. (2020). Integration of imitation learning using GAIL and reinforcement learning using task-achievement rewards via probabilistic graphical model. Advanced Robotics, 34(16), 1055-1067.

**Ethical Concerns:**

["NO or VERY MINOR ethics concerns only"]

**Final Justification:**

I appreciate the authors’ efforts to clarify and refine their contributions, but the concerns regarding the limited generality and robustness of the LLM-based component remain unresolved, so my overall evaluation is unchanged.

**Limitations:**

Yes, the authors have a section discussing limitations. However, the limitation regarding the LLM-based labeling stage could be addressed more deeply. The paper acknowledges that the LLM approach struggles in environments like MAMuJoCo, but it doesn't fully discuss the implications this has for the overall framework's claimed generality. The LLM labeling is presented as a key part of the contribution in Section 4, but the experiments suggest it is not universally applicable. I would encourage the authors to be more upfront about this limitation and discuss how the MisoDICE algorithm (Stage 2) can be used with other, potentially simpler or more general, methods for data labeling. This would strengthen the paper by focusing on its most robust and generalizable contribution.

**Quality:**

3

**Strengths And Weaknesses:**

*Strengths
- Originality and Significance: The paper tackles a practical and important problem: offline multi-agent imitation learning where demonstration quality is heterogeneous and unknown. This is a significant and relatively under-explored area, and the proposed problem setting is a valuable contribution.
- Theoretical Quality: The extension of the DICE framework to the multi-agent setting is theoretically sound. A key contribution is the use of a linear mixing network to maintain the convexity of the learning objective. This is a crucial choice that ensures stability and efficient optimization. The use of Lagrange multipliers for decomposition in this context is also an interesting approach.
- Empirical Results: While the validation might be considered minimal in scope, the paper demonstrates that MisoDICE outperforms baselines on standard, challenging benchmarks, providing evidence for the method's effectiveness.

*Weaknesses
- Dependence on LLMs and Generality of the Framework: The paper's core contribution feels split between the LLM-based labeling pipeline (Stage 1) and the MisoDICE algorithm (Stage 2). The first stage's reliance on LLMs for preference generation seems to lack generality. The authors themselves note that this approach fails on tasks like MAMuJoCo, where semantic context is poor, forcing them to use a rule-based oracle. This undermines the claim that LLM-based labeling is a core, generalizable part of the contribution and raises questions about the framework's applicability to common imitation learning domains like robotic manipulation. The two stages feel disconnected, and the paper might be stronger if it focused on the more robust contribution of the MisoDICE algorithm itself.
- Clarity of Positioning and Presentation:
The paper's positioning of imitation learning (IL) as a subfield of reinforcement learning (RL) could be more nuanced. While related, they are often treated as distinct paradigms. A more detailed discussion of their relationship would clarify the paper's contribution.
- The presentation of the baselines is potentially misleading. Although methods like "VDN" in Figure 1 look like baseline methods, thye are clearly ablation studies of the proposed method. They should be explicitly framed as such to avoid confusion and clearly delineate the contributions of each component of the proposed architecture.

---

> ### Author Rebuttal · Authors · 2025-07-30
>
> We thank the reviewer for the insightful comments. Below, we provide our detailed responses to each of your questions and concerns.
>
> > Dependence on LLMs and Generality of the Framework: ... This undermines the claim that LLM-based labeling is a core, generalizable part of the contribution and raises questions about the framework's applicability to common imitation learning domains like robotic manipulation. The two stages feel disconnected, and the paper might be stronger if it focused on the more robust contribution of the MisoDICE algorithm itself.
>
> We appreciate the reviewer’s insightful observation regarding the broader applicability of our algorithmic contributions. We agree that the LLM-based component is currently more suited to domains with interpretable features. Nonetheless, we believe that both components of our framework represent meaningful advancements. Our use of LLMs to identify expert-like demonstrations from fully unlabeled datasets is, to the best of our knowledge, a novel and impactful contribution that opens promising avenues for preference-based learning. In particular, it paves the way for future work on leveraging LLMs to identify expert behavior in more challenging and less interpretable environments, such as MAMuJoCo.
>
> Importantly, our core algorithmic design in Phase 2 is general and agnostic to how expert demonstrations are obtained, making it broadly applicable across various settings. In the revised version of the paper, we will refine our contribution statement to more clearly articulate the generality and standalone value of this component.
>
> > Clarity of Positioning and Presentation: The paper's positioning of imitation learning (IL) as a subfield of reinforcement learning (RL) could be more nuanced....
>
> We thank the reviewer for the insightful feedback. We recognize the importance of clearly distinguishing between IL and RL, especially given the nuanced differences in their problem settings, learning objectives, and assumptions. While RL focuses on learning optimal behavior through interaction with an environment and explicit reward signals, IL seeks to learn policies by mimicking expert behavior, often in the absence of reward functions. In the revised version of the paper, we will revise the presentation to explicitly highlight this distinction—both in the introduction and background sections.
>
> > ... Although methods like "VDN" in Figure 1 look like baseline methods, they are clearly ablation studies of the proposed method. ...
>
> We thank the reviewer for the insightful observation. You are correct that VDN, as used in our experiments, is a simplified adaptation of our main algorithm, MisoDICE, rather than a standalone existing baseline. We will revise the manuscript to clarify this distinction and avoid any potential confusion.
>
> > Clarification on BC Performance: In Figure 1, the performance of Behavioral Cloning (BC) appears to increase over the course of training. ...
>
> We thank the reviewer for the insightful question. The performance curves shown in our experiments reflect evaluation during the policy update process. Specifically, we use batch samples from the offline dataset to update the policy using BC, and after a certain number of epochs, we evaluate the policy’s performance. We will clarify this evaluation procedure more explicitly in the experimental section.
>
> > Justification of the Two-Stage Framework: The paper's main theoretical contribution appears to be the MisoDICE algorithm (Stage 2), which is elegant and well-motivated. However, it is tightly coupled with the LLM-based labeling pipeline (Stage 1), which seems less general and brittle, as evidenced by its failure on MAMuJoCo. Could the authors elaborate ...? Would the paper's contribution be clearer and more robust if Stage 2 (MisoDICE) were presented as the core method, with the LLM-based approach being just one possible (and domain-dependent) method for generating the required labeled data?
>
> We thank the reviewer for the insightful comments and thoughtful question, with which we fully agree. The close coupling between Phase 1 and Phase 2 in our framework stems from the original motivation of the work—*tackling imitation learning in the challenging setting where the dataset is entirely unlabeled*. In this context, Phase 1 plays a critical role, as effective imitation learning cannot proceed without first identifying expert-like demonstrations.
>
> While we agree that Phase 2 is more general and may have broader standalone applicability—making it particularly relevant to some readers—we believe that Phase 1 also offers important contributions. To the best of our knowledge, it is the first approach that leverages LLMs to identify expert-like trajectories from noisy, unlabeled data. This opens promising future directions, including extending such techniques to less interpretable domains like MuJoCo, potentially leading to broader impact in preference-based and offline IL.
>
> ---
>
> *We hope the above responses satisfactorily address your concerns. We will update the paper to include the discussion described above. If you have any further comments or questions, we would be happy to address them.*

---

### Official Review · Reviewer_7ov1 · 2025-07-02

**Clarity:** 3
**Significance:** 3
**Originality:** 3
**Rating:** 4
**Confidence:** 3

**Summary:**

The authors propose a novel framework for cooperative multi-agent offline imitation learning from unlabeled mixed-quality demonstrations. It is structured in two stages: trajectory labeling and multi-agent imitation learning. By extending the popular single-agent DICE framework to multi-agent settings with a new value decomposition and mixing architecture, their method yields a convex policy optimization objective and ensures consistency between global and local policies.

**Questions:**

see the weakness

**Ethical Concerns:**

["NO or VERY MINOR ethics concerns only"]

**Limitations:**

yes

**Quality:**

3

**Strengths And Weaknesses:**

Strengths：
1、The paper tackles the underexplored challenge of multi-agent IL with unlabeled mixed-quality demonstrations. The two-stage design is an innovative solution, effectively bridging preference learning and occupancy matching.
2、The technical core is strong. The convexity analysis of the objective under linear mixing is a crucial insight ensuring stable optimization. The proofs of global-local policy consistency and the connection between local policies and value functions provide theoretical grounding for the CTDE approach and decentralized execution.

Weakness：
1、The labeling phase relies on domain-specific semantic features (e.g., unit health in SMAC) for LLM feedback. For domains without interpretable features (e.g., MaMuJoCo), the proposed "rule-based oracle" lacks details. How was the rule-based oracle designed for MaMuJoCo? Provide algorithmic details and its performance vs. LLM in SMAC. If it requires domain-specific heuristics, does this undermine the framework’s generality?
2、How sensitive is the labeling phase to the quality and specificity of the LLM prompts? Could low-quality or overly generic LLM outputs significantly degrade O-MAPL refinement and subsequent IL performance?
3、 The author claims to have solved the computational complexity of the large-scale joint state-action space, but lacks relevant theoretical or experimental proof.
4、Proposition 5.5 assumes that behavioral strategies can be decomposed into independent local strategies, but the behaviors of agents in cooperative tasks are usually correlated. Please demonstrate the rationality of this assumption.
s

---

> ### Author Rebuttal · Authors · 2025-07-30
>
> We thank the reviewer for the insightful comments. Below, we provide our detailed responses to each of your questions and concerns.
> > The labeling phase relies on domain-specific semantic features ... How was the rule-based oracle designed for MaMuJoCo? ...If it requires domain-specific heuristics, does this undermine the framework’s generality?
>
> We thank the reviewer for the insightful question. First, to clarify, the "rule-based oracle" used for the MAMuJoCo tasks is derived from the ground-truth reward function. In this domain, we assume access to expert feedback to generate preference labels, as LLMs are not well-suited for interpreting raw low-level state representations typical of MuJoCo environments.
>
> In our framework, Phase 2 is more general, as it can be applied to settings where expert and suboptimal demonstrations are available, regardless of how the expert data is identified—either manually or via LLMs. In contrast, Phase 1 is somewhat less general, as it relies on the availability of contextual and interpretable features that can be understood by LLMs. However, we believe Phase 1 still offers independent value: to the best of our knowledge, this is the first approach that leverages LLMs to identify expert-like demonstrations from fully unlabeled and potentially noisy datasets. This opens promising directions for future work, particularly in exploring how to adapt LLM-based preference labeling to less interpretable domains like MuJoCo through better feature engineering or learned state abstractions.
>
> > How sensitive is the labeling phase to the quality and specificity of the LLM prompts? Could low-quality or overly generic LLM outputs significantly degrade O-MAPL refinement and subsequent IL performance?
>
> We thank the reviewer for the insightful question. As noted in the appendix, we have already conducted an ablation study to analyze the sensitivity of MisoDICE to the quality of preference feedback in Phase 1. Specifically, in **Section C.4** of the appendix, we compare two versions of the preference labeling pipeline: one using **GPT-4o** and the other using **GPT-4o-mini**. Since **GPT-4o-mini** is a smaller model with limited reasoning and contextual understanding capabilities, it tends to generate lower-quality preference signals. Our results show that this degradation in feedback quality leads to less accurate labeling and significantly reduces the overall performance of MisoDICE in Phase 2.
>
> We have also conducted an additional ablation study to analyze the impact of noisy preference feedback in Phase 1 on the overall performance of MisoDICE. In this experiment, we started with preference data generated by GPT-4o and randomly flipped a portion of the preference labels at varying levels (5%, 10%, 20%, etc.). The results, reported in the table below, clearly illustrate that increasing the amount of noise in the preference data leads to a noticeable decline in MisoDICE's performance, highlighting the framework’s sensitivity to label quality in the preference-based learning phase.
>
> | Scenario| MisoDICE (0%)| MisoDICE (5%)| MisoDICE (10%)| MisoDICE (20%)| MisoDICE (30%)|
> |--------------------|-----------------|------------------|------------------|------------------|------------------|
> | 2c_vs_64zg | 16.4 ± 1.3 | 14.7 ± 1.0| 13.0 ± 0.6| 11.6 ± 0.7| 11.5 ± 0.5|
> | 5m_vs_6m| 7.3 ± 0.1| 6.5 ± 1.0| 6.1 ± 1.9| 6.2 ± 1.3| 6.0 ± 1.7|
> | 6h_vs_8z| 8.7 ± 0.2| 8.2 ± 0.1| 8.0 ± 0.1| 8.1 ± 0.1| 7.8 ± 0.2|
> | corridor| 5.8 ± 0.8| 2.9 ± 0.6| 2.6 ± 0.5| 2.0 ± 0.3| 2.2 ± 0.6|
> | protoss_5_vs_5| 12.4 ± 0.5 | 11.8 ± 1.9| 11.5 ± 0.8| 10.9 ± 0.1| 10.0 ± 0.5|
> | protoss_10_vs_10| 12.9 ± 0.2 | 11.9 ± 0.9| 12.5 ± 1.8| 11.6 ± 0.3| 11.2 ± 1.2|
> | protoss_10_vs_11| 10.7 ± 0.4 | 10.4 ± 0.4| 10.3 ± 0.8| 10.2 ± 0.6| 9.7 ± 0.7|
> | protoss_20_vs_20| 13.5 ± 0.5 | 12.9 ± 0.8| 12.6 ± 1.0| 12.4 ± 1.0| 12.2 ± 1.3|
> | protoss_20_vs_23| 10.6 ± 0.2 | 10.1 ± 0.7| 9.9 ± 1.1| 9.2 ± 0.9| 9.4 ± 1.0|
> | terran_5_vs_5 | 9.1 ± 0.3| 8.4 ± 1.6| 6.8 ± 0.5| 9.0 ± 1.8| 8.1 ± 1.4|
> | terran_10_vs_10 | 9.1 ± 1.3| 8.3 ± 1.9| 7.6 ± 0.8| 7.1 ± 0.8| 7.2 ± 1.1|
> | terran_10_vs_11 | 6.4 ± 0.5| 6.6 ± 0.6| 6.2 ± 0.9| 6.0 ± 0.7| 6.0 ± 1.2|
> | terran_20_vs_20 | 9.2 ± 0.6| 8.5 ± 0.7| 8.4 ± 0.7| 8.1 ± 1.1| 7.8 ± 0.9|
> | terran_20_vs_23 | 5.6 ± 0.4| 5.5 ± 0.4| 5.2 ± 0.6| 5.0 ± 0.5| 4.9 ± 0.5|
> | zerg_5_vs_5| 7.5 ± 0.1| 7.4 ± 0.9| 7.2 ± 1.2| 6.3 ± 0.4| 6.5 ± 1.6|
> | zerg_10_vs_10 | 10.2 ± 0.6 | 9.3 ± 0.5| 8.2 ± 0.5| 8.7 ± 1.3| 8.0 ± 1.4|
> | zerg_10_vs_11 | 9.4 ± 0.3| 9.5 ± 1.3| 8.8 ± 0.3| 8.4 ± 0.8| 7.5 ± 0.5|
> | zerg_20_vs_20 | 10.2 ± 0.6 | 9.5 ± 0.3| 9.2 ± 0.7| 9.0 ± 0.6| 8.5 ± 0.9|
> | zerg_20_vs_23 | 9.5 ± 0.2| 9.4 ± 0.9| 8.9 ± 0.7| 8.5 ± 0.3| 8.1 ± 1.1|
>
> > The author claims to have solved the computational complexity of the large-scale joint state-action space, but lacks relevant theoretical or experimental proof.
>
> We thank the reviewer for the insightful question. Our design choices—including value factorization and local network construction—are specifically aimed at addressing the computational complexity inherent in multi-agent settings, particularly when dealing with large-scale joint state-action spaces. This approach has been adopted in several recent state-of-the-art MARL methods and has been shown to significantly improve computational scalability and efficiency. We will clarify this point more explicitly in the revised version of the paper.
>
>
> > Proposition 5.5 assumes that behavioral strategies can be decomposed into independent local strategies, but the behaviors of agents in cooperative tasks are usually correlated. Please demonstrate the rationality of this assumption.
>
> We thank the reviewer for the insightful question. This is actually a common assumption in the existing MARL literature. To clarify, we do not assume that the global policy is decomposable into separate independent local policies (The local policies are not necessarily independent.). Rather, our framework relies on the assumption of consistency between global and local policies—specifically, that the global policy can be expressed as the product of local policies. This formulation captures a synergistic relationship, where coordinated local behaviors collectively give rise to the global strategy, rather than treating each agent’s policy in isolation.
>
> ---
>
> *We hope the above responses satisfactorily address your concerns. We will update the paper to include the additional experiments and discussion described above. If you have any further comments or questions, we would be happy to address them*

---

### Official Review · Reviewer_eERK · 2025-07-02

**Clarity:** 3
**Significance:** 3
**Originality:** 3
**Rating:** 5
**Confidence:** 3

**Summary:**

The paper proposes multi-agent imitation learning in a cooperative setting containing mixed quality demonstrations - expert & non-expert. The authors propose a two-step approach where LLMs and preference based RL is used to label the demonstrations and then these are used likewise to train an offline imitation learning policy. The problem setup is novel and really important.

**Questions:**

1. What is the intuition behind R~Q-\gamma V in section 4? Is this advantage estimation that you are doing?
2. Does the rankings of the trajectory done in Phase-1 align with ground truth (actual rankings) ? This would help to understand how good is O-MAPL in recovering an appropriate reward function from noisy preference data.
3. How often are the 2 mixing networks trained in this framework? How often are the local networks trained? What objective function  is used to train the 2 mixing networks?
4. Though the evaluation returns are better for MIsoDice, the training plot seems not that statistically significant. I am curious as to why MisoDICE is almost similar in training performance in smaller configurations (Fig 1; 5v5; 10Vs 10) than bigger ones?
5. I could not find MisoDICE-VDN as mentioned in Baselines in the Experimental plots.
6. I would have liked to see an ablation study without phase-1; i.e: getting demonstrations and using them all as optimal demonstrations along with Phase-2 to understand how important the pretraining process is in this framework.

**Ethical Concerns:**

["NO or VERY MINOR ethics concerns only"]

**Final Justification:**

I thank the authors who clearly addressed my concerns and performed additional experiments/ablation to validate some of my concerns. I would like to increase my score post the rebuttal because this is a nice idea and has sufficient experimental evaluations /ablations to support the claims.

**Limitations:**

Yes

**Quality:**

3

**Strengths And Weaknesses:**

Strength:

1. Problem setting is novel, learning multi-agent IL policies in settings where demonstrations are of mixed quality is an interesting and important problem; this also occurs very often due to scarcity of expert demonstrations
2. Related work is well written; positioning the work well in the light of prior work
3. Experimental results are well conducted on standard MARL benchmarks of increasing difficulty, showing reasonable advantage of the proposed approach over baselines.

Weakness:

1. Though the paper uses techniques from prior work (like O-MAPL); they are not mentioned in the Background section making it difficult to follow specific details.
2. The first phase of this work is summarized briefly, without providing sufficient details. If LLM provides noisy preference data, I am not sure how O-MAPL learns a soft Q-function? Can it filter out the noisy preferences?
3. The authors use local network to learn local value functions; but I am not sure how well this would scale up for large number of agents and the no. of local value functions would be large.
4. I do not understand the intuition behind discriminator based classification problem (looks like cross-entropy loss) as in Eqn 9? What is the intuition behind posing the estimation problem as a supervised classification problem?
5. Though the problem statement is interesting, I think it is too computationally intensive to be able to scale up to large number of agents. There are 3 different local networks per agent and 2 different mixing networks making the overall framework extremely complicated and intensive.
6. There is a trade off between the amount of gain achieved in terms of performance versus the computational overhead of the proposed approach; thus I am skeptical about it’s utility for furthering research. IT looks like VDN also learns pretty well in this setting.

---

> ### Author Rebuttal · Authors · 2025-07-30
>
> We thank the reviewer for the insightful comments. Below, we provide our detailed responses to each of your questions and concerns.
> >Though the paper uses techniques from prior work (like O-MAPL); they are not mentioned in the Background section…
>
> We will revise the Background section to better highlight and clarify the importance of preference-based methods (such as O-MAPL) in our framework.
> >The first phase of this work is summarized briefly, without providing sufficient details.
>
> We thank the reviewer for the comment. We agree that the details of Phase 1 could be more thoroughly explained in the main paper. However, due to space constraints, we chose to move some of the descriptions to the appendix, in order to allocate more space in the main text to Phase 2, which involves more technical depth.
> > If LLM provides noisy preference data, I am not sure how O-MAPL learns a soft Q-function? …
>
> When the feedback is noisy or less accurate, the ability of O-MAPL to correctly distinguish expert-like trajectories deteriorates, which in turn negatively affects the downstream performance in Phase 2. To empirically validate this, we already conducted an ablation study in **Section C.4** comparing **GPT-4o** and **GPT-4o-mini** for preference labeling. Due to its limited reasoning ability, GPT-4o-mini produced noisier feedback, leading to lower labeling accuracy and degraded Phase 2 performance.
>
> Additionally, we ran an additional ablation where we introduced synthetic noise by randomly flipping GPT-4o-generated preferences at varying rates. The results, shown below, demonstrate a clear decline in MisoDICE’s performance as noise increases.
>
> | Scenario| MisoDICE (0%)| MisoDICE (5%)| MisoDICE (10%)| MisoDICE (20%)| MisoDICE (30%)|
> |--------------------|-----------------|------------------|------------------|------------------|------------------|
> | 2c_vs_64zg | 16.4 ± 1.3 | 14.7 ± 1.0| 13.0 ± 0.6| 11.6 ± 0.7| 11.5 ± 0.5|
> | 5m_vs_6m| 7.3 ± 0.1| 6.5 ± 1.0| 6.1 ± 1.9| 6.2 ± 1.3| 6.0 ± 1.7|
> | 6h_vs_8z| 8.7 ± 0.2| 8.2 ± 0.1| 8.0 ± 0.1| 8.1 ± 0.1| 7.8 ± 0.2|
> | corridor| 5.8 ± 0.8| 2.9 ± 0.6| 2.6 ± 0.5| 2.0 ± 0.3| 2.2 ± 0.6|
> | protoss_5_vs_5| 12.4 ± 0.5 | 11.8 ± 1.9| 11.5 ± 0.8| 10.9 ± 0.1| 10.0 ± 0.5|
> | protoss_10_vs_10| 12.9 ± 0.2 | 11.9 ± 0.9| 12.5 ± 1.8| 11.6 ± 0.3| 11.2 ± 1.2|
> | protoss_10_vs_11| 10.7 ± 0.4 | 10.4 ± 0.4| 10.3 ± 0.8| 10.2 ± 0.6| 9.7 ± 0.7|
> | protoss_20_vs_20| 13.5 ± 0.5 | 12.9 ± 0.8| 12.6 ± 1.0| 12.4 ± 1.0| 12.2 ± 1.3|
> | protoss_20_vs_23| 10.6 ± 0.2 | 10.1 ± 0.7| 9.9 ± 1.1| 9.2 ± 0.9| 9.4 ± 1.0|
> | terran_5_vs_5 | 9.1 ± 0.3| 8.4 ± 1.6| 6.8 ± 0.5| 9.0 ± 1.8| 8.1 ± 1.4|
> | terran_10_vs_10 | 9.1 ± 1.3| 8.3 ± 1.9| 7.6 ± 0.8| 7.1 ± 0.8| 7.2 ± 1.1|
> | terran_10_vs_11 | 6.4 ± 0.5| 6.6 ± 0.6| 6.2 ± 0.9| 6.0 ± 0.7| 6.0 ± 1.2|
> | terran_20_vs_20 | 9.2 ± 0.6| 8.5 ± 0.7| 8.4 ± 0.7| 8.1 ± 1.1| 7.8 ± 0.9|
> | terran_20_vs_23 | 5.6 ± 0.4| 5.5 ± 0.4| 5.2 ± 0.6| 5.0 ± 0.5| 4.9 ± 0.5|
> | zerg_5_vs_5| 7.5 ± 0.1| 7.4 ± 0.9| 7.2 ± 1.2| 6.3 ± 0.4| 6.5 ± 1.6|
> | zerg_10_vs_10 | 10.2 ± 0.6 | 9.3 ± 0.5| 8.2 ± 0.5| 8.7 ± 1.3| 8.0 ± 1.4|
> | zerg_10_vs_11 | 9.4 ± 0.3| 9.5 ± 1.3| 8.8 ± 0.3| 8.4 ± 0.8| 7.5 ± 0.5|
> | zerg_20_vs_20 | 10.2 ± 0.6 | 9.5 ± 0.3| 9.2 ± 0.7| 9.0 ± 0.6| 8.5 ± 0.9|
> | zerg_20_vs_23 | 9.5 ± 0.2| 9.4 ± 0.9| 8.9 ± 0.7| 8.5 ± 0.3| 8.1 ± 1.1|
>
> > There are 3 different local networks per agent and 2 different mixing networks making the overall framework extremely complicated and intensive…
>
> We thank the reviewer for the insightful comment. In fact, our use of local networks with mixing architectures under the CTDE framework is explicitly designed to enhance scalability and efficiency in multi-agent learning—a strategy also adopted by recent SOTA MARL and IL methods. Our experiments confirm that MisoDICE scales well, even on the most challenging SMAC_v2 scenarios.
>
> To further support our argument regarding the computational efficiency, we will include the following table summarizing the total training time for each benchmark environment. The results clearly demonstrate that MisoDICE requires a reasonable amount of training time.
>
> | Scenario| Time |
> |--------------------|-----------|
> | SMACv1| 2.30h|
> | SMACv2 (protoss)| 4.19h|
> | SMACv2 (terran) | 3.50h|
> | SMACv2 (zerg) | 2.95h|
>
> > I do not understand the intuition behind discriminator based classification problem (looks like cross-entropy loss) as in Eqn 9? …
>
> We thank the reviewer for the question. The purpose of the discriminator in Eq. 9 is to estimate the occupancy ratio, which is a critical component of our learning objective. This classifier-based approach can be shown to recover the occupancy ratio between expert and unlabeled datasets. Similar techniques have been widely adopted in prior IL algorithms, such as GAIL, where the discriminator serves a comparable role in distinguishing between expert and non-expert trajectories.
>
> > There is a trade off between the amount of gain achieved in terms of performance versus the computational overhead…  VDN also learns pretty well in this setting.
>
> We thank the reviewer for the comment. We agree that there is an inherent trade-off between performance gains and computational efficiency, which is a common phenomenon in ML. In our context, the VDN variant serves as a simplified version of MisoDICE. As expected, its performance is slightly lower than MisoDICE, but better than other baselines.
>
> > What is the intuition behind R~Q-\gamma V in section 4?...
>
> This approach is adapted from the O-MAPL paper on preference-based learning, where the expression $Q−\gamma V$ is interpreted as a reward function defined in the Q-value space. By learning this reward function implicitly through the Q and V functions, we enable a more consistent and integrated way of learning both the reward and policy.
> > Does the rankings of the trajectory done in Phase-1 align with ground truth?...
>
> We thank the reviewer for the insightful question. In response, we conducted an additional experiment in which we sampled 2,000 trajectory pairs per task and generated preference rewards using LLM feedback and O-MAPL. These were then compared to ground-truth preferences derived from environment rewards. The results, reported below, indicate an overall accuracy of 90.5%, demonstrating the high reliability of the LLM feedback in Phase 1. This highlights the effectiveness of our approach in Phase 1 in filtering expert trajectories that closely align with the ground-truth preferences.
>
> | Scenario| Accuracy |
> |---------------------|-----------|
> | 2c_vs_64zg | 79.2% |
> | 5m_vs_6m| 87.0% |
> | 6h_vs_8z| 89.9% |
> | corridor| 87.7% |
> | protoss_5_vs_5  | 89.2% |
> | protoss_10_vs_10| 91.1% |
> | protoss_10_vs_11| 90.5% |
> | protoss_20_vs_20| 92.0% |
> | protoss_20_vs_23| 93.8% |
> | terran_5_vs_5| 90.4% |
> | terran_10_vs_10 | 81.5% |
> | terran_10_vs_11 | 78.3% |
> | terran_20_vs_20 | 90.4% |
> | terran_20_vs_23 | 96.8% |
> | zerg_5_vs_5| 98.8% |
> | zerg_10_vs_10| 85.3% |
> | zerg_10_vs_11| 95.6% |
> | zerg_20_vs_20| 92.4% |
> | zerg_20_vs_23| 97.7% |
> > How often are the 2 mixing networks trained in this framework?….
>
> We thank the reviewer for the question. The mixing network and local networks are trained simultaneously via gradient backpropagation. The main training objective for jointly updating both components is provided in **Eq.6**. A detailed description of the training procedure can be found in **Section B.5** of the appendix, along with a visual illustration in **Figure 4**.
>
> > I am curious as to why MisoDICE is almost similar in training performance in smaller configurations (Fig 1; 5v5; 10Vs 10) than bigger ones?
>
> We thank the reviewer for the question. On small-scale tasks, MisoDICE and other baselines show similar performance due to low task complexity. While training plots may appear noisy, closer inspection shows that MisoDICE consistently outperforms the baselines.
>
> > I could not find MisoDICE-VDN as mentioned in Baselines in the Experimental plots.
>
> MisoDICE-VDN is actually VDN reported in our main experiments. We included it as a lightweight variant to highlight the modularity of our approach. We will clarify this in the revised version of the paper.
>
> > I would have liked to see an ablation study without phase-1; …
>
> We thank the reviewer for the suggestion. In response, we ran an ablation where all unlabeled data were treated as optimal for Phase 2. Results show that MisoDICE without labeling performs worse than the full version but slightly better than BC, highlighting the importance of Phase 1's labeling strategy for learning from noisy, unlabeled data.
>
> | Scenario| BC (no labeling)| MisoDICE (with Phase 1) | MisoDICE (no labeling) |
> |--------------------|----------------|------------------|--------------------------|
> | 2c_vs_64zg | 8.5 ± 0.1 | 16.4 ± 1.3| 10.9 ± 0.3|
> | 5m_vs_6m | 5.0 ± 1.1 | 7.3 ± 0.1| 5.2 ± 0.1 |
> | 6h_vs_8z | 7.0 ± 0.0 | 8.7 ± 0.2| 7.2 ± 0.0 |
> | corridor | 1.5 ± 0.1 | 5.8 ± 0.8| 1.8 ± 0.2 |
> | protoss_5_vs_5 | 9.2 ± 0.1 | 12.4 ± 0.5| 9.9 ± 1.9 |
> | protoss_10_vs_10 | 10.3 ± 0.6| 12.9 ± 0.2| 10.5 ± 0.7|
> | protoss_10_vs_11 | 8.2 ± 0.4 | 10.7 ± 0.4| 9.3 ± 0.3 |
> | protoss_20_vs_20 | 10.1 ± 0.2| 13.5 ± 0.5| 11.7 ± 0.9|
> | protoss_20_vs_23 | 8.1 ± 0.2 | 10.6 ± 0.2| 8.5 ± 0.5 |
> | terran_5_vs_5 | 6.5 ± 0.8 | 9.1 ± 0.3| 6.8 ± 0.8 |
> | terran_10_vs_10 | 6.6 ± 0.3 | 9.1 ± 1.3| 5.5 ± 1.3 |
> | terran_10_vs_11 | 4.7 ± 0.2 | 6.4 ± 0.5| 4.2 ± 0.6 |
> | terran_20_vs_20 | 6.9 ± 0.4 | 9.2 ± 0.6| 7.0 ± 0.4 |
> | terran_20_vs_23 | 4.0 ± 0.3 | 5.6 ± 0.4| 4.5 ± 0.7 |
> | zerg_5_vs_5 | 5.7 ± 0.5 | 7.5 ± 0.1| 5.7 ± 1.1 |
> | zerg_10_vs_10 | 7.3 ± 0.1 | 10.2 ± 0.6| 7.6 ± 0.8 |
> | zerg_10_vs_11 | 7.3 ± 0.2 | 9.4 ± 0.3| 7.7 ± 0.5 |
> | zerg_20_vs_20 | 7.4 ± 0.6 | 10.2 ± 0.6| 7.8 ± 1.0 |
> | zerg_20_vs_23 | 7.1 ± 0.3 | 9.5 ± 0.2| 7.3 ± 0.5 |
>
> ---
>
> *We hope the above responses satisfactorily address your concerns. We will update the paper to include the additional experiments and discussion described above. If you have any further comments or questions, we would be happy to address.*

---

> > ### Comment · Reviewer_eERK · 2025-08-07
> >
> > I thank the authors who clearly addressed my concerns and performed additional experiments/ablation to validate some of my concerns. I would like to increase my score post the rebuttal because this is a nice idea and has sufficient experimental evaluations /ablations to support the claims.

---

> > > ### Author Response · Authors · 2025-08-08
> > >
> > > We sincerely thank the reviewer for carefully reading our paper, providing a thoughtful response, and offering valuable feedback, as well as for maintaining a positive view of our work. We will certainly incorporate all the suggested discussions and additional experiments into the final version of the paper.

---

### Official Review · Reviewer_VDx6 · 2025-07-03

**Clarity:** 3
**Significance:** 3
**Originality:** 3
**Rating:** 4
**Confidence:** 4

**Summary:**

This paper introduces MisoDICE, a novel framework for offline multi-agent imitation learning (MAIL) that addresses the challenge of unlabeled, mixed-quality demonstrations. The authors propose a two-stage solution: first, a multi-step labeling pipeline utilizes LLMs and preference-based reinforcement learning to identify expert trajectories. Second, MisoDICE, a multi-agent IL algorithm, leverages these labels and extends the single-agent DICE framework with a value decomposition and mixing architecture to learn robust policies. The research demonstrates MisoDICE's superior performance on standard MARL benchmarks and discusses its theoretical guarantees, including convex policy optimization and global-local policy consistency.

**Questions:**

Could preference noise lead to misleading labels that propagate into Phase 2?

Have you tested how the performance of MisoDICE degrades when preference labels are noisy or contradictory?

**Ethical Concerns:**

["NO or VERY MINOR ethics concerns only"]

**Final Justification:**

Thank you for the detailed responses. I strongly encourage the authors to include the additional results in the final version of the paper. I find my original score reasonable and will keep it.

**Limitations:**

yes

**Paper Formatting Concerns:**

no concern

**Quality:**

3

**Strengths And Weaknesses:**

### Strength
+ MisoDICE tackles the highly challenging problem of offline MAIL using only unlabeled demonstrations of mixed quality. The proposed two stage solution is interesting.
+ MisoDICE addresses the combinatorial explosion of joint state-action spaces in multi-agent environments by building upon the CTDE paradigm leveraging the value decomposition and a linear mixing architecture.
+ The framework provides a theoretical analysis establishing local-global optimality consistency.

### Weakness
- MisoDICE performance heavily rely on the quality of the external preference model. Given different LLMs trained on different domains or context, the performance could be different. Though this point has been mentioned in the limitation, it is important as it being the basis of the success of this work.
- The two stage pipeline requires multiple model components which introduces additional implementation complexity.
- Though linear mixing provide convexity, the model might benefit from more expressive mixing strategies.

---

> ### Author Rebuttal · Authors · 2025-07-30
>
> We thank the reviewer for the insightful comments. Below, we provide our detailed responses to each of your questions and concerns.
> > MisoDICE performance heavily rely on the quality of the external preference model...
>
> We acknowledge that our method relies on a learned preference model guided by LLM-generated feedback to identify expert-like demonstrations. This design represents a significant departure from prior work, which typically assumes access to costly, high-quality expert-labeled trajectories. By leveraging LLMs as a scalable and cost-efficient supervision source, our framework enables IL from fully unlabeled datasets. This not only reduces reliance on human annotation but also broadens the applicability of imitation learning to more practical, real-world settings where labeled expert data is unavailable or difficult to collect.
>
> While we agree that performance heavily depends on the quality of feedback provided by the LLM, this is a common and reasonable phenomenon in the community. In fact, all existing IL algorithms similarly rely on the quality of expert demonstrations in the dataset.
>
> > The two stage pipeline requires multiple model components which introduces additional implementation complexity.
>
> We thank the reviewer for the insightful comment. We would like to highlight that, as demonstrated through extensive ablation studies in the appendix, our two-stage framework works effectively in tandem to deliver robust training outcomes across a range of settings. Moreover, while the stages are designed to complement each other, each stage also holds independent value. Specifically, the first stage can be broadly applied to extract expert-like demonstrations from unlabeled data and is compatible with a variety of existing imitation learning algorithms. The second stage, in turn, can operate with any dataset that includes both expert and suboptimal demonstrations, regardless of how the data is labeled. Therefore, whether used jointly or independently, both stages of our framework offer meaningful contributions to the imitation learning literature.
>
> > Though linear mixing provide convexity, the model might benefit from more expressive mixing strategies.
>
> We agree that more expressive mixing strategies—such as two-layer nonlinear mixing networks—can, in principle, enhance representational capacity. Our algorithm is designed to be general and can indeed accommodate such architectures. That said, we intentionally adopt a linear mixing network for two key reasons. First, as shown in Proposition 5.2, the linear form guarantees convexity of the training objective, which promotes more stable and reliable optimization. Second, our design is grounded in empirical evidence from recent work in MARL and multi-agent imitation learning, which consistently finds that nonlinear mixing networks are prone to overfitting and often degrade performance in offline regimes. Thus, our choice reflects a balance between expressiveness and stability, tailored for the offline learning setting
>
> > Could preference noise lead to misleading labels that propagate into Phase 2? Have you tested how the performance of MisoDICE degrades when preference labels are noisy or contradictory?
>
> Yes, noisy preference signals can lead to inaccurate trajectory labeling, which in turn can degrade the performance of the imitation learning algorithm in Phase 2. To investigate the sensitivity of our framework to the quality of preference feedback, we  conducted an ablation study presented in **Section C.4** of the appendix. In this experiment, we compared two versions of the preference labeling pipeline: one using **GPT-4o** and the other using **GPT-4o-mini**. While both models are capable of generating high-level semantic judgments, GPT-4o-mini—being a lighter variant—is significantly more limited in its reasoning and contextual understanding, especially in the complex, high-dimensional environments we consider.
>
> Our results, reported in **Table 10**, reveal a clear performance gap between the two. Specifically, MisoDICE trained with GPT-4o-based labels consistently outperformed the version using GPT-4o-mini, both in terms of average returns and learning stability. This supports the hypothesis that high-quality, consistent preference feedback is critical for effective trajectory labeling and downstream policy learning. These findings also underscore the importance of choosing capable LLMs when deploying preference-based learning pipelines in practical applications.
>
> To further address your question, we conducted an additional ablation study in which we explicitly introduced noise into the preference labeling process in Phase 1. Specifically, we obtained preference labels from GPT-4o and randomly flipped the preference outcomes for a certain percentage of trajectory pairs—at  5%, 10%, 20%, and 30% of the  samples. The results, reported below, clearly show a gradual degradation in MisoDICE’s performance as the level of noise increases,
>
> | Scenario           | MisoDICE (0%)   | MisoDICE (5%)   | MisoDICE (10%)  | MisoDICE (20%)  | MisoDICE (30%)  |
> |--------------------|-----------------|------------------|------------------|------------------|------------------|
> | 2c_vs_64zg         | 16.4 ± 1.3      | 14.7 ± 1.0       | 13.0 ± 0.6       | 11.6 ± 0.7       | 11.5 ± 0.5       |
> | 5m_vs_6m           | 7.3 ± 0.1       | 6.5 ± 1.0        | 6.1 ± 1.9        | 6.2 ± 1.3        | 6.0 ± 1.7        |
> | 6h_vs_8z           | 8.7 ± 0.2       | 8.2 ± 0.1        | 8.0 ± 0.1        | 8.1 ± 0.1        | 7.8 ± 0.2        |
> | corridor           | 5.8 ± 0.8       | 2.9 ± 0.6        | 2.6 ± 0.5        | 2.0 ± 0.3        | 2.2 ± 0.6        |
> | protoss_5_vs_5     | 12.4 ± 0.5      | 11.8 ± 1.9       | 11.5 ± 0.8       | 10.9 ± 0.1       | 10.0 ± 0.5       |
> | protoss_10_vs_10   | 12.9 ± 0.2      | 11.9 ± 0.9       | 12.5 ± 1.8       | 11.6 ± 0.3       | 11.2 ± 1.2       |
> | protoss_10_vs_11   | 10.7 ± 0.4      | 10.4 ± 0.4       | 10.3 ± 0.8       | 10.2 ± 0.6       | 9.7 ± 0.7        |
> | protoss_20_vs_20   | 13.5 ± 0.5      | 12.9 ± 0.8       | 12.6 ± 1.0       | 12.4 ± 1.0       | 12.2 ± 1.3       |
> | protoss_20_vs_23   | 10.6 ± 0.2      | 10.1 ± 0.7       | 9.9 ± 1.1        | 9.2 ± 0.9        | 9.4 ± 1.0        |
> | terran_5_vs_5      | 9.1 ± 0.3       | 8.4 ± 1.6        | 6.8 ± 0.5        | 9.0 ± 1.8        | 8.1 ± 1.4        |
> | terran_10_vs_10    | 9.1 ± 1.3       | 8.3 ± 1.9        | 7.6 ± 0.8        | 7.1 ± 0.8        | 7.2 ± 1.1        |
> | terran_10_vs_11    | 6.4 ± 0.5       | 6.6 ± 0.6        | 6.2 ± 0.9        | 6.0 ± 0.7        | 6.0 ± 1.2        |
> | terran_20_vs_20    | 9.2 ± 0.6       | 8.5 ± 0.7        | 8.4 ± 0.7        | 8.1 ± 1.1        | 7.8 ± 0.9        |
> | terran_20_vs_23    | 5.6 ± 0.4       | 5.5 ± 0.4        | 5.2 ± 0.6        | 5.0 ± 0.5        | 4.9 ± 0.5        |
> | zerg_5_vs_5        | 7.5 ± 0.1       | 7.4 ± 0.9        | 7.2 ± 1.2        | 6.3 ± 0.4        | 6.5 ± 1.6        |
> | zerg_10_vs_10      | 10.2 ± 0.6      | 9.3 ± 0.5        | 8.2 ± 0.5        | 8.7 ± 1.3        | 8.0 ± 1.4        |
> | zerg_10_vs_11      | 9.4 ± 0.3       | 9.5 ± 1.3        | 8.8 ± 0.3        | 8.4 ± 0.8        | 7.5 ± 0.5        |
> | zerg_20_vs_20      | 10.2 ± 0.6      | 9.5 ± 0.3        | 9.2 ± 0.7        | 9.0 ± 0.6        | 8.5 ± 0.9        |
> | zerg_20_vs_23      | 9.5 ± 0.2       | 9.4 ± 0.9        | 8.9 ± 0.7        | 8.5 ± 0.3        | 8.1 ± 1.1        |
>
> ---
>
> *We hope the above responses satisfactorily address your concerns. We will update the paper to include the additional experiments  and discussion described above. If you have any further comments or questions, we would be happy to address them.*

---

> > ### Comment · Reviewer_VDx6 · 2025-08-05
> >
> > Thank you for the detailed responses. I strongly encourage the authors to include the additional results in the final version of the paper. I find my original score reasonable and will keep it.

---

> > > ### Author Response · Authors · 2025-08-05
> > >
> > > We thank the reviewer for the suggestion. We will definitely include these valuable additional experiments in the revised manuscript.

---

### Decision · Program_Chairs · 2025-09-17

**Decision:**

Accept (poster)

**Comment:**

The paper introduces a framework for multi-agent imitation learning in cooperative settings with mixed-quality demonstrations. The proposed two-step approach uses large language models and preference-based reinforcement learning to label demonstrations, which are then applied to train an offline imitation learning policy. The method, MisoDICE, is evaluated on standard MARL benchmarks, showing improved performance and providing theoretical guarantees such as convex policy optimization and global–local policy consistency.

After discussion, most reviewers concerns are addressed and all recommended acceptance.